# Confining charge-transfer complex in a metal-organic framework for photocatalytic CO$_2$ reduction in water

Sanchita Karmakar [1], Soumitra Barman [1], Faruk Ahamed Rahimi[1], Darsi Rambabu[1], Sukhendu Nath [2] & Tapas Kumar Maji [1]✉

In the quest for renewable fuel production, the selective conversion of CO$_2$ to CH$_4$ under visible light in water is a leading-edge challenge considering the involvement of kinetically sluggish multiple elementary steps. Herein, 1-pyrenebutyric acid is post-synthetically grafted in a defect-engineered Zr-based metal organic framework by replacing exchangeable formate. Then, methyl viologen is incorporated in the confined space of post-modified MOF to achieve donor-acceptor complex, which acts as an antenna to harvest visible light, and regulates electron transfer to the catalytic center (Zr-oxo cluster) to enable visible-light-driven CO$_2$ reduction reaction. The proximal presence of the charge transfer complex enhances charge transfer kinetics as realized from transient absorption spectroscopy, and the facile electron transfer helps to produce CH$_4$ from CO$_2$. The reported material produces 7.3 mmol g$^{-1}$ of CH$_4$ under light irradiation in aqueous medium using sacrificial agents. Mechanistic information gleans from electron paramagnetic resonance, in situ diffuse reflectance FT-IR and density functional theory calculation.

The current scenario of the global energy crisis due to uncontrolled fossil fuel consumption and continuous elevation of greenhouse gases inspires us to utilize solar light as a renewable energy source for CO$_2$ reduction reaction (CO$_2$RR). Unfortunately, photoreduction of the CO$_2$ molecule is arduous due to the high C=O dissociation energy of ~ 750 kJ mol$^{-1}$[1]. CO$_2$RR involves proton-coupled multi-electron reduction process in producing CO, CH$_4$, and higher hydrocarbons. Considering the C1-based reduced product from CO$_2$, the formation of CH$_4$ is thermodynamically more favorable ($E^0_{CO_2/CH_4}$ = −0.24 V vs. NHE at pH 7) compared to other C1 feedstock ($E^0_{CO_2/CO}$ = −0.53 V; $E^0_{CO_2/HCHO}$ = −0.48 V; $E^0_{CO_2/CH_3OH}$ = −0.38 V, $E^0_{CO_2/HCOOH}$ = −0.61 V vs. NHE at pH 7)[1,2]. In contrast, from a kinetic point of view, the formation of CH$_4$ is challenging due to the involvement of the eight-electron transfer process, which requires precise electrons drive to carry forward the overall reaction for selective

product formation[3]. Additionally, the kinetically sluggish nature of the multielectron transfer process from photosensitizer to the catalytic site makes the process even harder. Apart from these hurdles, the production of CH$_4$ as a CO$_2$-reduced product is highly beneficial, as it is the main component of natural gas that provides a significant environmental benefit, producing more energy by mass than other hydrocarbons[4,5]. Bearing these aspects in mind, a significant effort has been directed to design robust photocatalysts that can capture and reduce CO$_2$ to CH$_4$ selectively and efficiently[1,6–9].

The introduction of supramolecular donor–acceptor (D–A) complex by post-synthetic modification within the confined nano-space of a metal-organic framework (MOF) can eventually alter the chemical environment and overall optoelectronic properties[10]. Through a rational supramolecular design approach with structural regulation at a molecular level, a catalytically active coordination nanospace can be

[1]Molecular Materials Laboratory, Chemistry and Physics of Material Unit (CPMU), School of Advance Material (SAMat), Jawaharlal Nehru Centre for Advanced Scientific Research, Jakkur, Bangalore 560064, India. [2]Ultrafast Spectroscopy Section, Radiation & Photochemistry Division, Bhabha Atomic Research Centre, Mumbai 400 085, India. ✉e-mail: tmaji@jncasr.ac.in

engineered in a highly stable nanoscale MOF. However, a molecular design approach with remarkable precision and exquisite control needs to be implemented to achieve such a catalytically active site in a confined nanospace. The versatile and highly amenable structural tunability of MOFs by post-synthetic modification (PSM) allows tailoring inherent properties, including semiconducting and optoelectronic properties, compared to traditional inorganic semiconductors. By virtue, the PSM of a nanospace of MOF using an electron donor moiety would allow the introduction of a suitable acceptor moiety to attain a charge transfer (CT) complex without altering the integrity of the framework. Besides, it would be an excellent choice to establish a functional photosensitizer to galvanize the electronic and optical properties. Furthermore, the D–A interaction can be exploited for channeling the electron flow based on the push-pull effect to the catalytic center to carry out catalytic reactions. Additionally, the charge transfer complex would also result in a low energy absorption band to harvest visible light to generate photo-modulated electron-hole pair with low exciton binding energy[9,11]. Recently, MOFs have been extensively studied as a photocatalyst for $CO_2RR$ due to their promising $CO_2$ capture ability, high surface area, and textural properties[12–18]. However, there are a handful of reports for MOF-based photocatalysts, which are studied in an aqueous medium for visible-light-driven $CO_2RR$[19–21].

Previous studies on $Zr^{IV}$-based MOFs showed that the Zr-oxo cluster in PCN222 ($Zr^{IV}$-porphyrin MOF) and PCN136 ($Zr^{IV}$-hexakis(4-carboxyphenyl)hexabenzocoronene MOF) acts as an active catalytic site, where integrated organic linkers perform as an antenna for light-harvesting to activate the metal cluster on photoirradiation[22,23]. However, the reduction ability is mostly limited to a two-electron reduction process to produce CO or HCOOH. Herein, we have envisioned for post-synthetic fabrication of a donor-acceptor CT complex as a visible light-harvesting unit inside the nanospace of MOF, which would enhance the electron transfer kinetics to the catalytic center for $CO_2$ reduction in an aqueous medium. Our design principle lies with the following consideration: the low energy CT band will help in visible light-harvesting; secondly, close confinement inside the coordination nanospace will provide a high excited-state lifetime of the photogenerated electrons by decreasing the nonradiative electron–hole recombination pathway; and thirdly, the proximity of catalytic center and the light-harvesting unit will enhance the kinetics of electrons transfer which is of paramount importance for highly reduced and selective $CO_2$ reduction product[19,24–28].

In this context, we envisaged mesoporous MOF-808 (Zr) as a suitable platform for PSM for its well-accessible formate to exchange with 1-pyrenebutyric acid (PBA) to produce a luminescent MOF (MOF-808-PBA) with a high excited-state lifetime[19,29]. Moreover, the prudent choice of defect-regulated mesoporous MOF-808 (Zr) can be justified by its large pore size, presence of hierarchical meso and microporosity, very high water/chemical stability combined with available unsaturated $Zr^{IV}$ metal site[29–31]. Further, a supramolecular D-A assembly was introduced using noncovalent grafting of methyl viologen (MV; an electron acceptor) inside the pore of MOF-808-PBA. Integrating the D–A module inside the pore surface can create an artificial "special-pair" like the system to facilitate fast charge transfer kinetics by driving the charge-separation process to reduce $CO_2$ beyond the two-electron reduction process[32]. Hence, rapid electron transfer process from PBA → MV → catalytic site (Zr-oxo cluster) and suitable band position overcome the required electron injection for $CO_2RR$ to produce a highly reduced product in aqueous medium (Fig. 1). MOF-808-PBA-MV produced 7.3 mmol g[-1] of $CH_4$ with >99% selectivity suppressing $H_2$ evolution in aqueous medium using BNAH (1-benzyl-1,4-dihydro-nicotinamide) and TEA (triethylamine) as a sacrificial agent. Furthermore, the reaction mechanism was established by in situ *diffuse reflectance FT-IR* (DRIFT), electron paramagnetic resonance (EPR) studies and transient absorption (TA) spectroscopy, which were well supported by density functional theory (DFT) calculation.

## Results

### Synthesis and characterizations of MOF-808-PBA and MOF-808-PBA-MV

MOF-808 was synthesized by a slight modification of the reported procedure and characterized using powder X-ray diffraction (PXRD), FT-IR, NMR spectroscopy and thermogravimetric analysis (Fig. 2a and Supplementary Figs. 1–3)[30,33]. TEM images of as-synthesized MOF-808 showed octahedral morphology with particle sizes in the range of 300-400 nm (Fig. 2b)[1]. The porosity of the MOF-808 was confirmed by performing $N_2$ adsorption measurements at 77 K, and it showed type IV adsorption isotherm according to IUPAC classification, which suggested the presence of meso and micropore (Fig. 2c)[34]. The Brunauer−Emmett−Teller (BET) surface area was calculated to be 889 m[2] g[-1] in the range between $P/P_0$ 0.05 and 0.3. The pore size distribution using the NLDFT method showed that the micropore centered at 1.6 nm, whereas the mesopores were distributed in

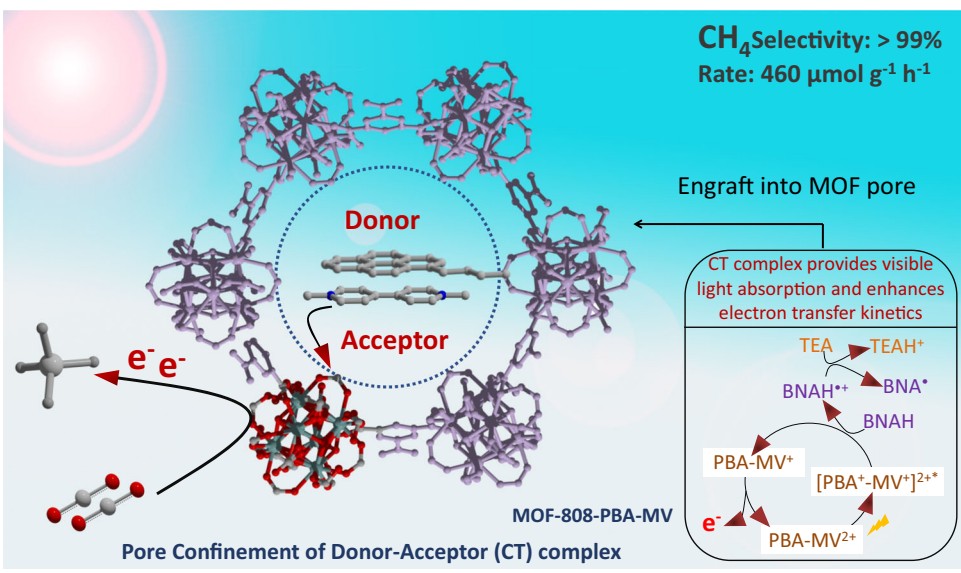

**Fig. 1 | Schematic illustration to engraft a donor−acceptor (PBA-MV) complex into MOF-808 (Zr) pore for visible-light-driven $CO_2$ reduction to selective $CH_4$ production.** Here, the D–A complex acts as a light harvester to boost the electron flow near the catalytic site in the presence of light.

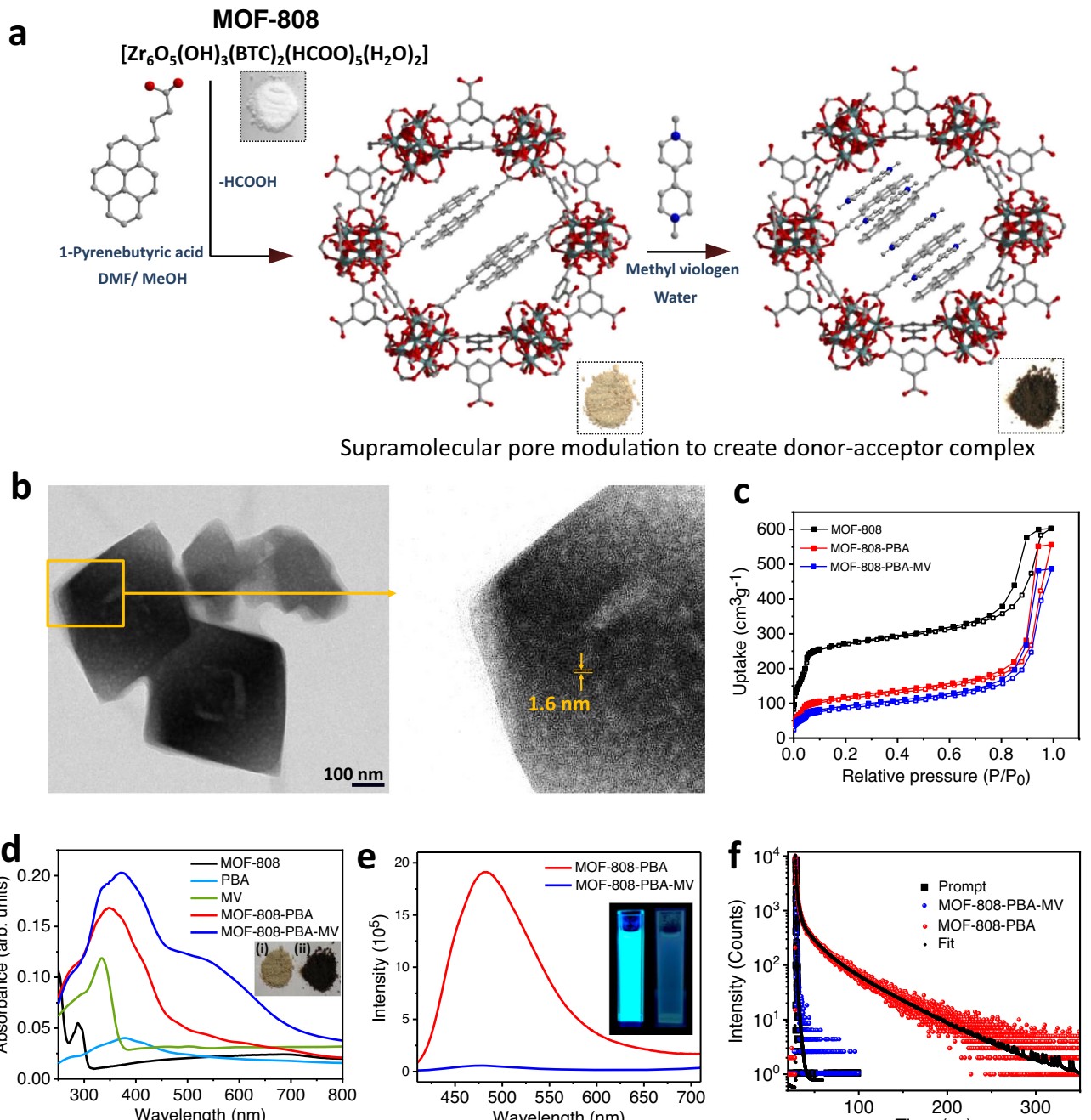

**Fig. 2 | Characterization and illustration of MOF-808, MOF-808-PBA, and MOF-808-PBA-MV. a** Synthetic scheme: Construction of MOF-808-PBA as a donor module via post-synthetic linker exchange and then introducing methyl viologen (MV) as an electron acceptor to prepare MOF-808-PBA-MV. **b** HRTEM images of MOF-808. Magnified image shows the presence of micro and mesopore. **c** N₂ adsorption isotherm of MOF-808, MOF-808-PBA, and MOF-808-PBA-MV at 77 K. **d** Solid-state UV–vis spectra of PBA, MV, MOF-808, MOF-808-PBA, and MOF-808-PBA-MV; inset showing the photograph of MOF-808-PBA (i) and MOF-808-PBA-MV (ii). **e** Photoluminescence spectra of MOF-808-PBA and MOF-808-PBA-MV. The inset shows a photograph of MOF-808-PBA and MOF-808-PBA-MV dispersed in MeOH under UV light. **f** Time-resolved luminescence decay of MOF-808-PBA and MOF-808-PBA-MV ($\lambda_{ex}$ = 330 nm, $\lambda_{col}$ = 480 nm).

between 3-10 nm (Supplementary Fig. 4). Accordingly, the micro and mesopore volume was evaluated to be 0.3 and 0.78 cm³ g⁻¹, respectively, with $V_{micro}/V_{meso}$ ratio of 0.39. The origin of the mesopore can be attributed to the missing linker defect, as reported earlier[30,31]. The hierarchical porosity is also endorsed by HRTEM analysis (Fig. 2b). As-synthesized MOF-808 showed sponge-like morphology with a clear contrast that can be attributed to the defects-based mesopore, which was randomly distributed throughout each particle[35]. HRTEM images also exhibited lattice fringes with an interplanar distance of 1.66 nm

which can be credited to the inherent microporosity of the framework, which is also supported by the pore size distribution analysis[19].

At first, the internal pore of mesoporous MOF-808 was covalently modified with 1-pyrenebutyric acid (PBA) via solvent-assisted linker exchange (SALE) method by submerging MOF powder in PBA solution (MeOH/DMF) and the resulting luminescent MOF named as MOF-808-PBA (Supplementary Figs. 5–9)[29,36]. ¹H-NMR spectra of digested MOF-808-PBA revealed that the formate peak integration diminished to one from five along with the presence of an additional peak related to the

PBA molecule, indicating four formates of $Zr_6$ cluster substituted by the PBA molecule (Supplementary Figs. 7 and 8)[29]. FT-IR spectra of MOF-808-PBA exhibited bands at~ 3000 and 846 cm$^{-1}$ which corresponds to $\nu$(C–H) symmetric and asymmetric stretching frequency of the PBA molecule, further confirming the successful encapsulation of PBA into the MOF pore (Supplementary Fig. 9)[37]. $N_2$ adsorption measurement of MOF-808-PBA revealed a decrease in surface area from 889 to 413 m$^2$ g$^{-1}$ with a significant decrease of micropore volume to 0.148 cm$^3$ g$^{-1}$, whereas mesopore volume remained intact with $V_{micro}$/ $V_{meso}$ ratio of 0.191. This result suggested the micropore is occupied by the PBA during post-synthetic linker exchange (Fig. 2c and Supplementary Fig. 4). Here, mesoporosity helped the diffusion of the PBA molecules into the micropore during the SALE process. Pristine MOF-808 possessed an absorption band in the UV region (250–310 nm) due to the $\pi \rightarrow \pi^*$ transition of the BTC linker[19]. Solid-state UV–vis spectra of MOF-808-PBA exhibited an additional broad absorption band from 310 to 500 nm with a maximum at 350 nm, along with a shoulder band at 420 nm, which is utterly different from vibronic monomeric spectral features of PBA (Fig. 2d and Supplementary Fig 10)[38]. Moreover, the solid-state UV–vis spectrum of PBA showed a broad absorption band from 250 to 700 nm (Fig. 2d). This result indicated ground-state aggregation of PBA molecules in the confined MOF pore, presumably through aromatic $\pi$–$\pi$ stacking interactions. Afterward, photoluminescence (PL) measurements were carried out to obtain further insight into the optical properties of MOF-808-PBA. The PL spectra of MOF-808-PBA showed a red-shifted, broad, structureless excimer emission band at 480 nm ($\lambda_{ex} = 370$ nm), which can be attributed to the close proximity of PBA molecules in the confined nanospace of MOF, hence enabling excimer formation (Fig. 2e, Supplementary Figs. 11, 12)[38,39]. An exceptionally long lifetime of the PBA-confined MOF-808 ($\tau_{av} = 23.2$ ns) over PBA monomer ($\tau_{av} = 3.01$ ns) further corroborates the formation of PBA excimer. (Fig. 2f, Supplementary Fig. 13 and

Supplementary Table 1)[40,41]. Moreover, the confinement effect of MOF pore contributes to the restriction of the nonradiative pathway, which resulted in an unexpectedly higher quantum yield of 23.88% (solid-state) and a long lifetime of 23.2 ns compared to solid PBA ligand (QY% = 9.3%, $\tau_{av} = 14.51$ ns) (Fig. 2f, Supplementary Fig. 14 and Supplementary Table 1)[42].

Methyl viologen ($MV^{2+}$) molecules were introduced as an acceptor in MOF-808-PBA to form a donor−acceptor supramolecular CT complex inside the confined nanospace. The UV−vis spectrum of the resulting dark brown color MOF-808-PBA-MV compound showed a distinct CT band from 450 to 700 nm with an absorption maximum at 540 nm (Fig. 2d and Supplementary Fig. 5). MOF-808-PBA-MV was digested with KOH/$D_2O$/DMSO-$d_6$ and subjected to $^1H$ NMR measurement which suggested the precise ratio of PBA and $MV^{2+}$ is of 4:3 per formula unit of MOF (Supplementary Fig. 15). Time-dependent density functional theory (TDDFT) calculation suggested the CT band between pyrenebutyric acid (PBA) and methyl viologen (MV) appears at 596.01 nm, which is in agreement with the experimental one (Fig. 3a). Detailed analysis of the results obtained from TDDFT calculation revealed that the band at 596.01 nm could be attributed to the transition from HOMO to LUMO + 1 (49.5% contribution), which is found to be the charge transfer transition from PBA to MV as obtained from the molecular orbital analysis (Supplementary Table 2). Next, the PL and the excited-state lifetime of the PBA in MOF-808-PBA-MV were studied to understand the charge transfer properties. As shown in Fig. 2e and Supplementary Fig. 16, the excimer emission of MOF-808-PBA was significantly quenched, reflecting the exciton quenching due to CT complex formation[43]. Quenching of the emission intensity of PBA excimer in MOF-808-PBA-MV was also observed under UV illumination (Inset of Fig. 2e). The generation of CT complex displayed a drastic decline of PBA lifetime from 23.2 to 1.2 ns ($\lambda_{ex} = 370$ nm, $\lambda_{col} = 480$ nm) (Fig. 2f, Supplementary Table 1). Further, EPR studies showed a single

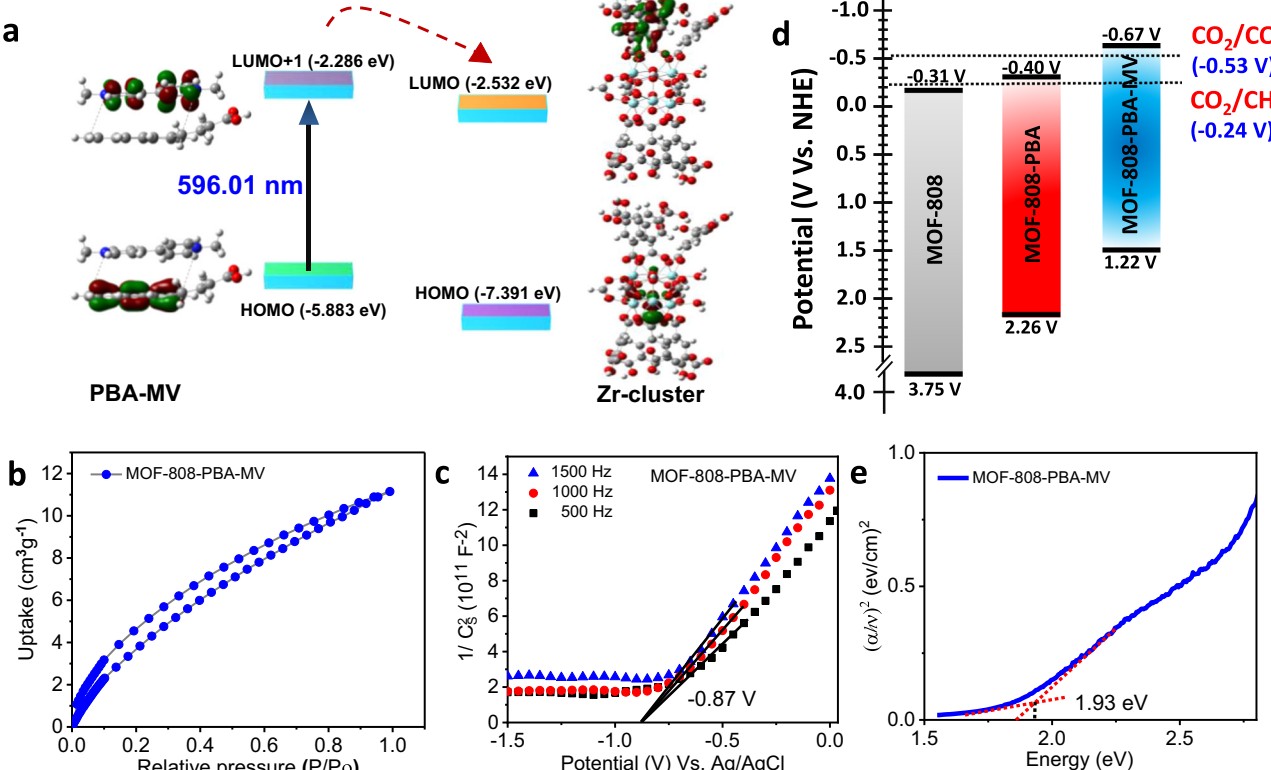

**Fig. 3 | Feasibility towards $CO_2$ reduction reaction. a** Electron transfer feasibility from PBA-MV to Zr-cluster using DFT. **b** $CO_2$ adsorption of MOF-808-PBA-MV at 298 K. **c** Mott−Schottky (MS) plot for MOF-808-PBA-MV in 0.2 M $Na_2SO_4$ aqueous solution. **d** HOMO−LUMO band position diagram for MOFs (MOF-808, MOF-808-PBA, MOF-808-PBA-MV) obtained from Mott−Schottky (MS) and DRS. **e** Tauc plot of MOF-808-PBA-MV evaluating the optical bandgap.

resonance peak at $g = 1.98$ of viologen radical, confirming the formation of charge-separated species (PBA$^{\bullet+}$-MV$^{\bullet+}$) inside the MOF pore (Supplementary Figs. 17, 18)[44]. Importantly, to understand the nature of the CT complex we also prepared the MOF-808-PBA-MV under completely dark condition designated as MOF-808-PBA-MV_d, which showed no recognizable EPR signal in that state, which implies daylight act as a stimulus to form the charge-separated species (details have been provided in the Supplementary Information; Supplementary Fig. 19). Apart from that, the redox signature of MV$^{2+}$ from the cyclic voltammetry exhibited two reversible one-electron waves at $E^1_{1/2} = -0.38$ V and $E^2_{1/2} = -0.78$ V vs. Ag/AgCl corresponding to the MV$^{2+}$/MV$^{\bullet+}$, and MV$^{\bullet+}$/MV redox transitions, respectively (Supplementary Fig. 20)[45]. The cyclic voltammetry of MOF-808-PBA-MV exhibited both the cathodic and anodic peaks corresponding to the generation of MV$^{2+} \rightleftharpoons$ MV$^{\bullet+}$ and MV$^{\bullet+} \rightleftharpoons$ MV (Supplementary Fig. 21). To further clarify the decrease of the anodic peak corresponding to MV$^{\bullet+} \rightarrow$ MV$^{2+}$ formation, three consecutive CV cycles were scanned, where we noticed after 1st segment, the peak current corresponding to MV$^{2+} \rightleftharpoons$ MV$^{\bullet+}$ was significantly decreased due to the formation of stable PBA$^{\bullet+}$-MV$^{\bullet+}$ CT complex inside the MOF pore. The PXRD and HRTEM analysis of MOF-808-PBA-MV confirmed that the structural integrity and morphology of pristine MOF remained intact after PSM (Supplementary Figs. 1 and 22–24). In addition, MOF-808-PBA-MV possessed BET surface area of 357 m$^2$ g$^{-1}$ with a significant contraction of micropore volume to 0.09 cm$^3$ g$^{-1}$, whereas mesopore volume remained almost constant with $V_{micro}/V_{meso}$ ratio of 0.125 (Fig. 2c and Supplementary Fig. 4). All these experiments suggest the formation of supramolecular donor–acceptor CT complex in the confined nanospace of MOF. Next, CO$_2$ adsorption measurement of MOF-808-PBA-MV was carried out at 298 K and showed an uptake of 12 cm$^3$ g$^{-1}$ at saturation, suggesting that CO$_2$ can diffuse easily to the catalytic centre during photocatalytic CO$_2$ reduction (Fig. 3b).

A low energy CT band appeared due to the formation of donor–acceptor CT complex inside the MOF pore, which can render a convenient path to regulate the electron flow based on visible light harvesting toward the Zr-oxo cluster, an active catalytic site for successive CO$_2$ reduction. Apart from that, for efficient CO$_2$ reduction, the photocatalyst is required to have suitable conduction and valence band. The band energy levels of MOF-808, MOF-808-PBA, and MOF-808-PBA-MV were examined using flat band potential from Mott-Schottky (MS) measurement (Fig. 3c and Supplementary Figs. 25a, 26a). All the MOFs exhibited positive slopes from C$^{-2}$ values (vs. applied potential), which are consistent with the typical n-type nature of the semiconductor. Figure 3d summarized the band energy level of pristine MOF-808 as well as post-synthetically modified MOFs. From the MS measurement, LUMO of pristine MOF-808 was calculated, which was estimated to be −0.51 V vs. Ag/AgCl (i.e. −0.31 V vs. NHE at pH 7) (Supplementary Fig. 25a). Based on the Tauc-plot, the optical band gap of MOF-808 was calculated to be 4.06 eV (Supplementary Fig. 25b). The disfavored band alignment of pristine MOF clearly signifies the inability to reduce CO$_2$. Upon successive grafting of PBA-MV into the MOF pore, the position of LUMO changed dramatically from −0.31 V (MOF-808) to −0.40 V (MOF-808-PBA) to −0.67 V (MOF-808-PBA-MV) vs. NHE at pH 7 (Fig. 3c and Supplementary Fig. 25a, 26a). Next, the optical band gap was calculated from the Tauc plot and found to be 2.75 and 1.93 eV for MOF-808-PBA, MOF-808-PBA-MV, respectively (Fig. 3e and Supplementary Fig. 26b). Accordingly, the valence band (HOMO) of MOF-808-PBA and MOF-808-PBA-MV was calculated to be 2.35 and 1.26 V (vs. NHE at pH 7), respectively. Hence, it can be concluded the introduction of supramolecular donor–acceptor CT complex in the confined nanospace of MOF-808 resulted in an energetically favorable LUMO to carry out CO$_2$ reduction[46]. In addition, their energy level was further investigated by ultraviolet photoelectron spectroscopy (UPS) (Supplementary Fig. 27, Supplementary Table 3) which showed a high accordance with the MS

measurement[47,48]. As aforementioned, all the optical and electronic features are hugely advantageous for visible-light-driven photocatalytic applications. To understand the importance of the homogeneous distribution of the CT complex in MOF-808-PBA-MV, a physical mixture of MOF-808, PBA, and MV was prepared (MOF-808(PBA + MV)-Phy mix.) (Supplementary Figs. 28, 29). This physical mixture can be considered a heterogeneous (supramolecular CT + MOF) system. It is worth mentioning that MOF-808(PBA + MV)-Phy mix. showed minimal CO$_2$ reduction activity compared to MOF-808-PBA-MV (Supplementary Table 7). This poor activity can be attributed to the unstable CT complex in MOF-808(PBA + MV)-Phy mix. which disintegrates in the catalytic condition as realized in UV−vis spectra of the catalytic solution (Supplementary Fig. 29).

## Photocatalytic CO$_2$ reduction reaction

The catalytic activity of MOF-808-PBA-MV was assessed in a CO$_2$-saturated aqueous medium under visible light irradiation (300 W Xenon Arc lamp, $\lambda > 400$ nm) in the presence of 1-benzyl-1,4-dihydro-nicotinamide (BNAH) and triethylamine (TEA) as a sacrificial agent. Remarkably, MOF-808-PBA-MV showed impressive activity for CO$_2$RR towards CH$_4$ formation with >99% selectivity, with efficient suppression of H$_2$ evolution in the aqueous medium. The gaseous products were monitored by gas chromatography and represented in Fig. 4a. The CH$_4$ production was continuously increased with irradiation time, and a total of 7.3 mmol g$^{-1}$ was produced in 16 h (Fig. 4a and Supplementary Figs. 30, 31). The maximum production rate was calculated to be 460 μmol g$^{-1}$ h$^{-1}$ with a maximum TON of 28 for CH$_4$ formation, which is exceptionally high for photocatalytic CO$_2$ reduction in the water medium (Supplementary Fig. 32, Supplementary Tables 4, 5). In this process, 0.369 mmol g$^{-1}$ of H$_2$ was also generated. Apparent quantum efficiency tests (AQE) were also performed with different monochromatic wavelengths. As illustrated in Supplementary Fig. 33, the highest AQE of 1.4% was achieved at a wavelength of 550 nm, which strongly suggested the charge transfer-assisted photocatalytic activity of MOF-808-PBA-MV (Supplementary Table 6, see Supplementary Information). To confirm the generated CH$_4$ originated from the photocatalytic reduction of dissolved CO$_2$, we performed the reaction using isotopic $^{13}$CO$_2$. A peak at $m/z = 17$ clearly demonstrated that produced $^{13}$CH$_4$ was generated from the photoreduction of dissolved CO$_2$ in a water medium (Fig. 4b and Supplementary Fig. 34). To unveil the role of CT complex inside the MOF pore in prompting photocatalytic CO$_2$ reduction, several control experiments were conducted under a similar reaction condition and discussed in the Supplementary information demonstrating the important role of electron flow in a confined nanospace (Fig. 4c, Supplementary Figs. 35, 36 and Supplementary Table 7). Furthermore, no detectable amount of product was obtained when the reaction was carried out without a catalyst or in the dark or in an Ar atmosphere (Fig. 4d, Supplementary Table 7). In CO$_2$-saturated aqueous solution in the presence of only TEA, MOF-808-PBA-MV exhibited 1.2 mmol g$^{-1}$ of CH$_4$ along with 0.456 mmol g$^{-1}$ of CO and 0.291 mmol g$^{-1}$ of H$_2$ in 16 h irradiation time (Fig. 4d, Supplementary Table 7). Similarly, the employment of BNAH as a sole sacrificial electron donor in water medium, MOF-808-PBA-MV produced 3.5 mmol g$^{-1}$ of CH$_4$ along with 0.230 mmol g$^{-1}$ of CO and 0.265 mmol g$^{-1}$ of H$_2$ in 16 h irradiation time (Fig. 4d, Supplementary Table 7). Importantly, the combination of BNAH and TEA played a significant role in the process of CO$_2$ reduction reaction to CH$_4$ formation over MOF-808-PBA-MV[49]. Here, BNAH acts as an efficient hole scavenger to enhance the catalytic activity. However, TEA was required for efficient CO$_2$RR as it suppressed back electron transfer from the reduced photosensitizer to BNAH$^{\bullet+}$[50]. Finally, the BNA radical (BNA$^{\bullet}$) produced by the deprotonation of BNAH$^+$ was further dimerized to produce BNA$_2$, which was confirmed from the LCMS

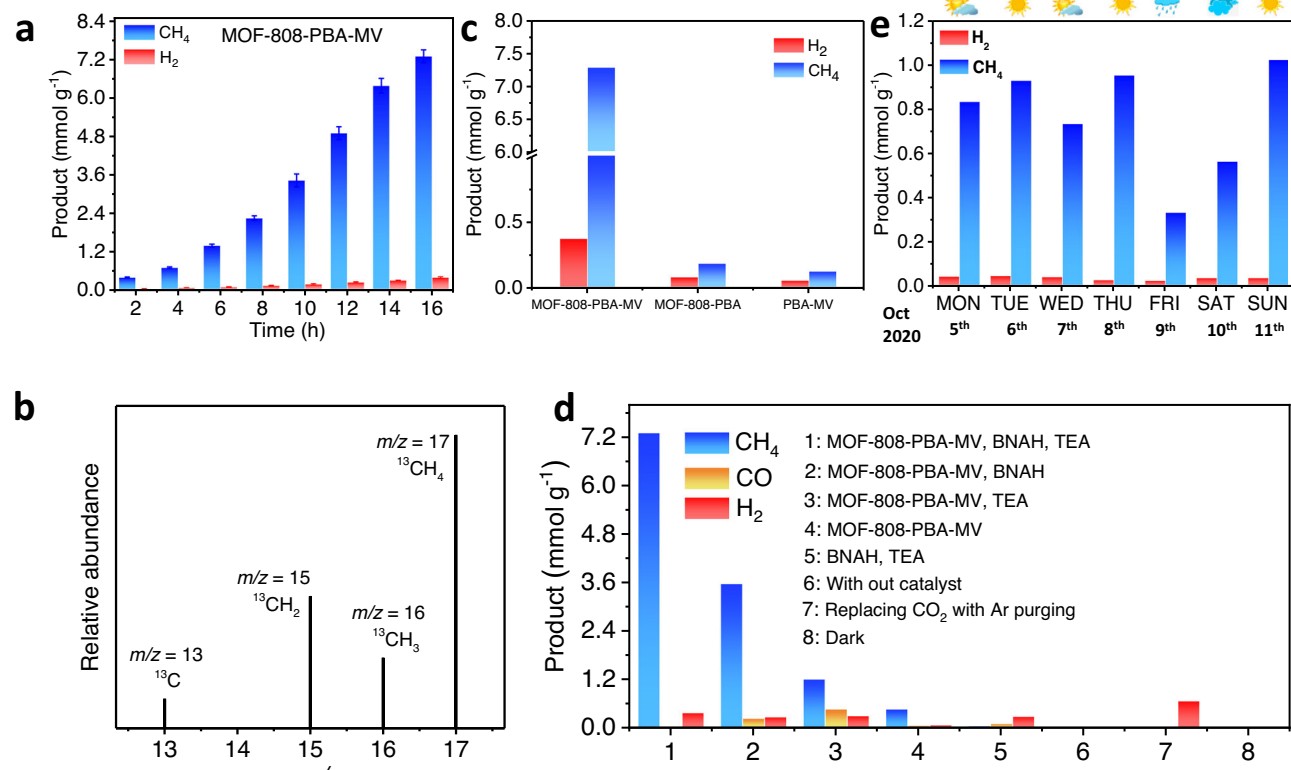

**Fig. 4 | Photocatalytic CO₂RR performance. a** The amount of CH₄ and H₂ evolution by MOF-808-PBA-MV as a function of time under visible light irradiation in water medium using BNAH and TEA as a sacrificial electron donor; error bars mean ± standard deviations calculated from three independent measurements. **b** Mass spectrum of produced ¹³CH₄ ($m/z = 17$) via isotopic ¹³CO₂ reduction under visible light over MOF-808-PBA-MV. **c** Comparison of photocatalytic product generation by using MOF-808-PBA-MV along with MOF-808-PBA and homogeneous PBA-MV in a water medium. **d** Control experiments of photocatalytic CO₂ reduction reaction over MOF-808-PBA-MV under different conditions. **e** The amount of CH₄ and H₂ evolution by MOF-808-PBA-MV under sunlight irradiation in a water medium using BNAH and TEA as a sacrificial electron donor after 6 h.

data (Supplementary Figs. 37, 38). Moreover, the reduction power of BNA₂ ($E^0 = 0.26$ V vs. SCE) is stronger than BNAH ($E^0 = 0.57$ V vs. SCE) and, therefore, efficiently quenched the PBA⁺ photosensitizer[51,52]. Further, to understand the role of the defect in mesoporous MOF-808, we have also prepared a microporous analog of the catalyst (details have been provided in the Supplementary Information; Supplementary Figs. 39–43). A four-fold enhanced catalytic activity was observed in mesoporous MOF-808-PBA-MV (CH₄: 7.3 mmol g⁻¹) as compared to microporous MOF-808-PBA-MV (CH₄: 1.8 mmol g⁻¹), which can be attributed to the increased availability of coordinatively unsaturated Zr^IV sites in the Zr-oxo cluster (Supplementary Fig. 44)[31]. Catalytic efficiency of MOF-808-PBA-MV was evaluated under direct sunlight irradiation on our laboratory's rooftop from 5 to 11 October (2020). The catalytic efficiency was weather dependent as expected, and we achieved the highest CH₄ production of 1.020 mmol g⁻¹ on a sunny day, 11th Oct, after 6 h of sunlight irradiation (Fig. 4e, Supplementary Fig. 45). ¹H NMR and high-performance liquid chromatography (HPLC) were carried out to analyze the liquid product. ¹H NMR spectra suggested the presence of a trace amount of formate, which was formed as an intermediate of CH₄ formation during the catalysis (Supplementary Fig. 46). No other liquid products were detected, further confirmed by HPLC analysis (Supplementary Fig. 47). In addition, MOF-808-PBA-MV was easily recovered from the reaction medium by centrifugation for post-catalytic analysis and again used for catalytic activity. Examining the stability of MOF-808-PBA-MV in the recycling experiment, it showed a negligible drop in CH₄ production even after six consecutive cycles with a total of 66 h of light irradiation. (Supplementary Fig. 48). Additional

experiment was performed using CO to verify the catalytic performance towards CO reduction. After the photocatalytic reaction, CH₄ was found to be a major product with a production of 10.4 mmol g⁻¹ in 16 h (Supplementary Fig. 49). In this case, the CH₄ production is higher compared to photocatalytic CO₂RR, which is obvious due to the conversion of CO₂ to CO is a more thermodynamically uphill process. Post-catalytic analysis of MOF-808-PBA-MV manifested the structural integrity and retention of optoelectronic properties (Supplementary Figs. 50–54).

The enhanced photocatalytic CO₂ reduction to CH₄ over MOF-808-PBA-MV as compared to its donor module MOF-808-PBA can be justified by transient photocurrent measurement and impedance studies. The photo-responsive behavior was dramatically enhanced due to the formation of the CT complex (Fig. 5a). The measured photocurrent of MOF-808-PBA-MV was 11.24 µA cm⁻² whereas; MOF-808-PBA exhibited a photocurrent of 1.18 µA cm⁻², indicating a tenfold increase in photocurrent after the formation of the CT complex. Additionally, this experiment was also measured in the presence of a sacrificial electron donor and found a slight enhancement of current density (13.26 µA cm⁻²) was observed (Supplementary Fig. 55). The enhanced photo-induced current upholds the superior catalytic activity of MOF-808-PBA-MV. This assertion was also supported by the electrochemical impedance spectroscopy (Supplementary Fig. 56). MOF-808-PBA-MV showed a much smaller radius of semi-circle as compared to MOF-808-PBA, indicating a small charge transfer resistance. Moreover, after irradiation with visible light, the semicircle of MOF-808-PBA-MV got further decreased, suggesting a fast electron transfer kinetics.

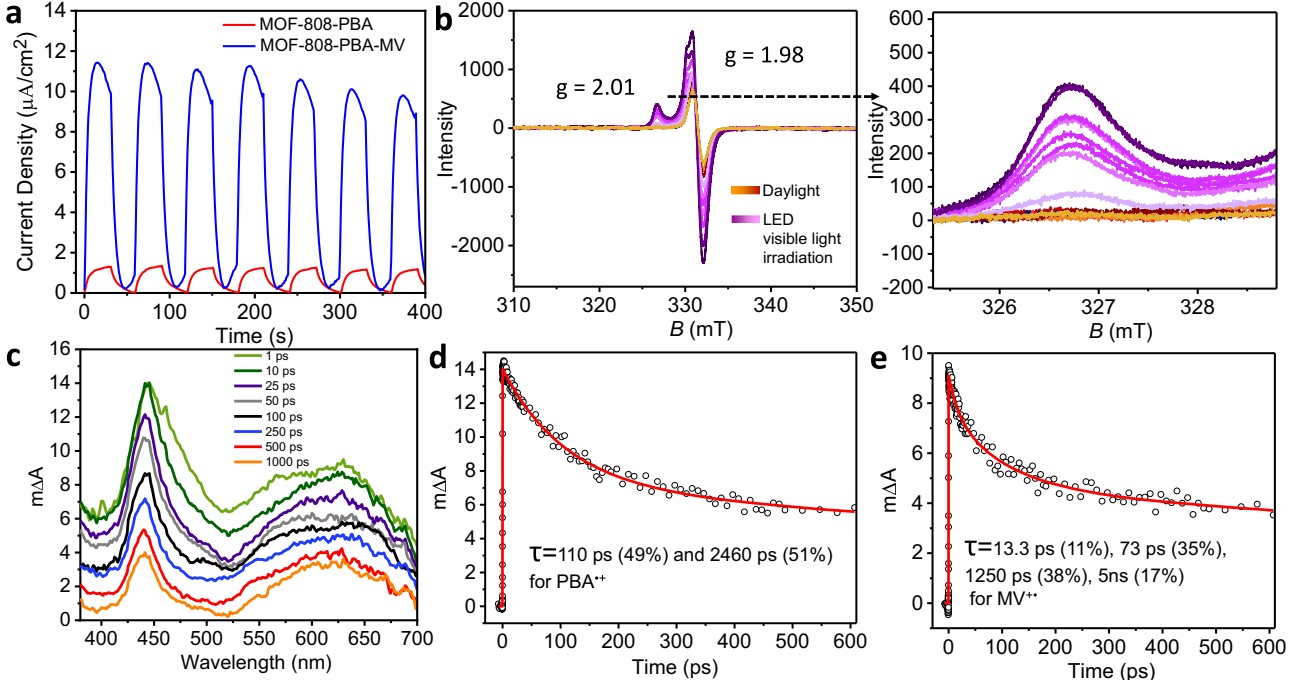

**Fig. 5 | CT interaction into the confined space towards the formation of Zr$^{III}$ species. a** Transient photocurrent responses of MOF-808-PBA and MOF-808-PBA-MV in 0.2 M Na$_2$SO$_4$ aqueous solution under visible-light irradiation. **b** EPR spectra of MOF-808-PBA-MV in dark and under visible light irradiation. Magnified EPR spectra of MOF-808-PBA-MV showing the formation of Zr$^{III}$ species. **c** Transient absorption spectra of MOF-808-PBA-MV at different time delays. **d**, **e** The transient decays at 440 and 650 nm. Solid lines in panels **d** and **e** are fitted data.

## Exploration of the charge transfer pathway

To uncover the mechanism behind photocatalytic activity and gain deeper insight into the electron transfer process in MOF-808-PB-MV, EPR spectra were collected with light irradiation (Fig. 5b). As expected, the intensity of the signal at a g value of 1.98 increased significantly due to the increased concentration of charge-separated species [PBA$^{+\bullet}$-MV$^{+\bullet}$] into the MOF pore. Most importantly, a characteristic EPR signal at $g = 2.01$ was detected due to the formation of Zr$^{III}$ species, which was revealed only upon light irradiation (Fig. 5b)[53]. Hence, the excited state electron generated from the CT complex is injected into the Zr-oxo cluster to reduce CO$_2$ to CH$_4$[22,54]. To further support this statement, time-resolved photoluminescence (TR-PL) spectroscopy of MOF-808-PBA-MV was performed both in Ar and CO$_2$ atmosphere (Supplementary Fig. 57 and Supplementary Table 8), which exhibited an average lifetime of 1.74 and 0.92 ns, respectively, suggesting that CO$_2$ binding with the Zr-oxo cluster promoted fast excited state electron transfer kinetics from PBA-MV to the catalytic centre[55].

The charge transfer interaction between PBA and MV was further investigated using femtosecond (fs) transient absorption (TA) techniques. The TA spectra (Fig. 5c) of MOF-808-PBA-MV showed a narrow excited state absorption (ESA) band at 440 nm due to the cationic pyrene moiety, PBA$^{+\bullet}$ and a very broad ESA band around 550–700 nm due to MV$^{+\bullet}$[56]. The instantaneous appearance of the PBA$^{+\bullet}$ and MV$^{+\bullet}$ indicates very efficient electron transfer which takes place within the time resolution of our TA instrument (120 fs). The absence of the electron transfer process in a homogeneous solution of pyrene and MV$^{2+}$ (Supplementary Fig. 58) suggests the importance of confinement in the electron transfer process. The decay kinetics of PBA$^{+\bullet}$ (440 nm) and MV$^{+\bullet}$ (650 nm) was analyzed by multiexponential decay function, and the fitted parameters are 110 ps (49%) and 2460 ps (51%) for PBA$^{+\bullet}$ and 13.3 ps (11%), 73 ps (35%), 1250 ps (38%), 5 ns (17%) for MV$^{+\bullet}$ (Fig. 5d, e). The 5 ns decay component at 650 nm is due to the

pyrene excimer (see supplementary information)[57]. Distinct decay time constants for PBA$^{+\bullet}$ and MV$^{+\bullet}$ suggested the absence of the energy-wasting back electron transfer process in MOF-808-PBA-MV. Significantly shorter lifetime of MV$^{+\bullet}$ (average lifetime ~507 ps) in MOF-808-PBA-MV as compared to microsecond lifetime in other Zr-based MOF system[58] also suggested enhanced transfer of electrons from MV$^{+\bullet}$ to the catalytic center of MOF-808. Moreover, the frontier orbitals (FMOs) of PBA-MV and Zr-cluster of the MOF are aligned based on TDDFT analysis. LUMO of Zr-cluster (−2.532 eV) was found to be lower in energy than LUMO + 1 of PBA-MV (−2.286 eV). This indicates the feasibility of the photoexcited electron transfer from LUMO + 1 of PBA-MV photosensitizer to the LUMO of the Zr-cluster (Fig. 3a).

## Insight into the CO$_2$ reduction mechanism

Subsequently, in situ DRIFT measurement was performed to reveal the reaction intermediate during the CO$_2$ reduction process (Fig. 6, Supplementary Fig. 59). The MOF-808-PBA-MV photocatalyst was first degassed at 120 °C to remove all the adsorbed gases. Before the introduction of CO$_2$ and H$_2$O vapor into the reactor, an FT-IR spectrum of the catalyst was recorded and selected as background to suppress catalyst peaks. A peak at 1377 cm$^{-1}$ appeared after introducing CO$_2$, and water vapor into the reactor could be assigned as monodentate carbonate (m-CO$_3^{2-}$)[59]. The peak at 1436 cm$^{-1}$ could be assigned to symmetric stretching of HCO$_3^{-1}$. After light irradiation, several new peaks appeared. A peak that emerged at 1242 cm$^{-1}$ can be assigned to the CO$_2^{-}$ stretching[59]. Most importantly, a peak at 1619 cm$^{-1}$ was detected, which can be assigned as COOH* species[1]. COOH* is a very crucial intermediate during CO$_2$ reduction to CH$_4$ via the formate pathway. In addition, a strong band at 1723 cm$^{-1}$ appeared, which could be attributed to the bending vibration of C = O[60]. In the meantime, a small band appeared at 2101 cm$^{-1}$, assigned to CO coordinated with metal[61]. Importantly, the absorption band at 1855, 1766, and 1111 cm$^{-1}$ belongs to CHO*, CH$_2$O*, and

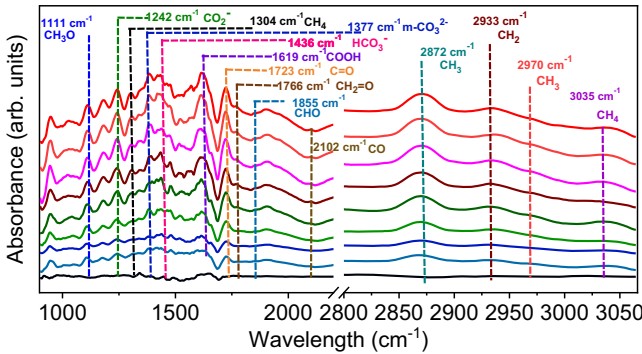

**Fig. 6 | Detection of intermediate.** Time-dependent in situ FT-IR spectra of photocatalytic $CO_2RR$ in a mixture of $CO_2$ and $H_2O$ vapor over MOF-808-PBA-MV in the dark and under visible light irradiation. Irradiation times 10, 20, 30, 40, 50, 60, 70, 80 min, respectively.

$CH_3O^*$ species, respectively, which are pivotal intermediates for $CH_4$ formation[1,62,63]. Besides, the band at 2872, 2933, 2970, 3035, and 1304 $cm^{-1}$ were assigned to C−H stretching vibrations, which clearly suggests the formation of $CH_4$ (see the supplementary information for more details)[60]. Furthermore, we have also performed an in-situ DRIFT experiment in the presence of BNAH and TEA, which followed a similar pathway as mentioned above (details have been provided in the Supplementary Information; Supplementary Fig. 60). In addition, the in-situ DRIFT study using isotopic $CO_2$ ($^{13}CO_2$) provides an insightful conclusion, where we observed red shifted stretching bands corresponds to the crucial intermediate (Supplementary Fig. 61)[64]. The mechanistic aspect was further clarified by using CO as a feeding gas instead of $CO_2$ (Supplementary Fig. 62).

Based on the DRIFTS study and DFT calculations, we have proposed a possible catalytic cycle for the photocatalytic $CO_2RR$ by MOF-808-PBA-MV (Fig. 7a, b, Supplementary Tables 9–36). For theoretical calculations, we have designated the MOF-808 catalyst, having the formula $[Zr_6(\mu_3-O)_5(\mu_3-OH)_3(HCOO)_5(BTC)_2(H_2O)_2]$ as f-$[Zr^{IV}(BTC)_2(H_2O)_2]$ (f = $Zr_5(\mu_3-O)_5(\mu_3-OH)_3(HCOO)_5$, Zr-cluster fragment). Upon photoirradiation, the photogenerated hole of [PBA⁺-MV⁺]²⁺* will get reductively quenched by sacrificial electron donor BNAH (Supplementary Fig. 63), and the photoexcited electron will readily get transferred to the BTC ligand of Zr-cluster, and thus the catalytic $CO_2$ reduction will be initiated on Zr-cluster as reported earlier[22–24,54,65]. As compared to microporous MOF-808, missing linker (formate) defect-regulated mesoporous MOF contains open metal sites (Zr^IV) which are considered to be the active catalytic centers[31]. One electron reduction of f-$[Zr^{IV}(BTC)_2(H_2O)_2]$ (1) generates f-$[Zr^{IV}(BTC^-)(BTC)(H_2O)]^-$ (2) with the removal of one water molecule ($\Delta G = -0.48$ eV). The generation of transient state in f-$[Zr^{IV}(BTC^-)(BTC)(H_2O)]^-$ with BTC⁻ (2) is verified by the spin density distribution plot where the spin density of the extra electron is clearly seen on BTC ligand (Supplementary Table 12). Moreover, f-$[Zr^{IV}(BTC^-)(BTC)(H_2O)]^-$ (2) (Supplementary Table 12) is the active species for the catalytic cycle. Next, either reduction of f-$[Zr^{IV}(BTC^-)(BTC)(H_2O)]^-$ (2) followed by oxidative addition of $CO_2$ to f-$[Zr^{III}(BTC^-)(BTC)(H_2O)]^{2-}$ (3) (Fig. 7a and Supplementary Table 13) or first, oxidative addition of $CO_2$ to f-$[Zr^{IV}(BTC^-)(BTC)(H_2O)]^-$ (2) followed by reduction of f-$[Zr^V(BTC)(BTC)(H_2O)(CO_2{}^{2-})]^-$ (4′) (Supplementary Fig. 64, Supplementary Table 14) may occur. The binding of $CO_2$ with f-$[Zr^{III}(BTC^-)(BTC)(H_2O)]^{2-}$ (3) is an exergonic process ($\Delta G = -0.72$ eV) with a lower activation barrier ($\Delta G^{\ddagger} = 0.55$ eV) to generate $CO_2$ bound species whereas the $CO_2$ binding pathway with f-$[Zr^{IV}(BTC^-)(BTC)(H_2O)]^-$ (2) is an endergonic ($\Delta G = +0.36$ eV) with a higher activation barrier ($\Delta G^{\ddagger} = 0.60$ eV) (Fig. 7b and Supplementary Tables 16, 17). This inferred that the $CO_2$ binding to f-$[Zr^{III}(BTC^-)(BTC)(H_2O)]^{2-}$ (3) is the major pathway under the

experimental conditions. This is quite expected because a lower oxidation state on Zr prior to the binding of $CO_2$ will facilitate the oxidative addition of $CO_2$. The next step in the photocatalytic cycle involves the protonation of $CO_2$-bound intermediate f-$[Zr^{IV}(BTC)(BTC)(H_2O)(CO_2{}^{2-})]^{2-}$ (4) (Supplementary Table 15) leading to f-$[Zr^{IV}(BTC)(BTC)(H_2O)(COOH)]^-$ (5) ($\Delta G = -2.07$ eV) (Supplementary Table 18) which after subsequent protonation and water elimination will afford f-$[Zr^{IV}(BTC)(BTC)(H_2O)(CO)]$ (6) ($\Delta G = -1.54$ eV) (Supplementary Table 19). After that, one-electron reduction of f-$[Zr^{IV}(BTC)(BTC)(H_2O)(CO)]$ (6) will lead to f-$[Zr^{IV}(BTC^-)(BTC)(H_2O)(CO)]^-$ (7) ($\Delta G = -0.77$ eV) (Supplementary Table 20) which can either release CO molecule to regenerate f-$[Zr^{IV}(BTC^-)(BTC)(H_2O)]^-$ (2) or undergo proton-coupled reduction to form f-$[Zr^{IV}(BTC^-)(BTC)(H_2O)(CHO)]^-$ (8) (Supplementary Table 21) and enter the $CH_4$ formation pathway. However, proton-coupled reduction of f-$[Zr^{IV}(BTC^-)(BTC)(H_2O)(CO)]^-$ (7) to f-$[Zr^{IV}(BTC^-)(BTC)(H_2O)(CHO)]^-$ (8) is highly exergonic ($\Delta G = -2.45$ eV) compared to the release of CO from f-$[Zr^{IV}(BTC^-)(BTC)(H_2O)(CO)]^-$ (7) ($\Delta G = -0.34$ eV) which indicates preferential higher-order reduction up to $CH_4$ rather than the generation of CO. In the following steps of the photocatalytic cycle, the successive proton couple reduction of f-$[Zr^{IV}(BTC^-)(BTC)(H_2O)(CHO)]^-$ (8) will lead to f-$[Zr^{IV}(BTC^-)(BTC)(H_2O)(CH_3OH)]^-$ (11), where all the intermediate steps are highly exergonic (Fig. 7a, b and Supplementary Tables 21–24). In the next step, proton-coupled reduction and subsequent water elimination from f-$[Zr^{IV}(BTC^-)(BTC)(H_2O)(CH_3OH)]^-$ (11) intermediate will lead to the formation of f-$[Zr^{IV}(BTC^-)(BTC)(H_2O)(CH_3)]^-$ (12) (Supplementary Table 25) which is found to be highly exergonic ($\Delta G = -2.47$ eV). Finally, f-$[Zr^{IV}(BTC^-)(BTC)(H_2O)(CH_3)]^-$ (12) intermediate will readily undergo proton-coupled reduction to result in the release of $CH_4$ ($\Delta G = -2.71$ eV), and thus the active species f-$[Zr^{IV}(BTC^-)(BTC)(H_2O)]^-$ (2) will get regenerated to re-enter into the catalytic cycle. The highly exergonic reduction pathway of f-$[Zr^{IV}(BTC^-)(BTC)(H_2O)(CH_3OH)]^-$ (11) ($\Delta G = -2.47$ eV) suppressed the endergonic desorption of $CH_3OH$ from f-$[Zr^{IV}(BTC^-)(BTC)(H_2O)(CH_3OH)]^-$ (11) ($\Delta G = +0.28$ eV) which explained the non-production of $CH_3OH$ (Fig. 7b). Furthermore, the role of proximal $H_2O$ was also investigated by DFT calculation, which revealed that the proximal $H_2O$ facilitates the desorption of hydrophobic $CH_4$ ($\Delta G = -2.71$ eV) (Supplementary Fig. 65). This result suggests the positive impact of the existing proximal $H_2O$ in releasing off the hydrophobic $CH_4$ product to enhance desorption.

## Discussion

In summary, we have created a simple yet highly efficient integrated system by pore modulation of MOF-808 for visible-light-driven $CO_2$ reduction to $CH_4$ selectively and efficiently with excellent yield in an aqueous medium. The embedded CT complex comprising of PBA and MV was exploited as a backbone for the electron transfer process to provide photoexcited electrons to the Zr^IV catalytic center to facilitate challenging multielectron reduction. The CT complex-induced electron delivery for the utilization of $CO_2RR$ is the first of its kind to the best of our knowledge. Encouragingly, this strategy is beneficial to achieve $CH_4$ formation with >99% selectivity with excellent yield by suppressing $H_2$ and other carbonaceous products. The mechanism was well explored through in situ DRIFTS by identifying reaction intermediates. Apart from that, EPR was utilized to identify Zr^III as an active catalytic site for $CO_2$ photoreduction, and the rapid electron transfer kinetics was recognized from TA spectroscopy. Additionally, a catalytic cycle was constructed along with Gibbs's free energy pathway to visualize detailed mechanistic insight and energetics of the reaction intermediates with the help of DFT calculation. This work provides the utilization of a D−A complex inside a confined nanospace of MOF that can lead to a design protocol to construct a highly efficient heterogeneous catalyst towards selective photoreduction of $CO_2$ to a highly reduced product.

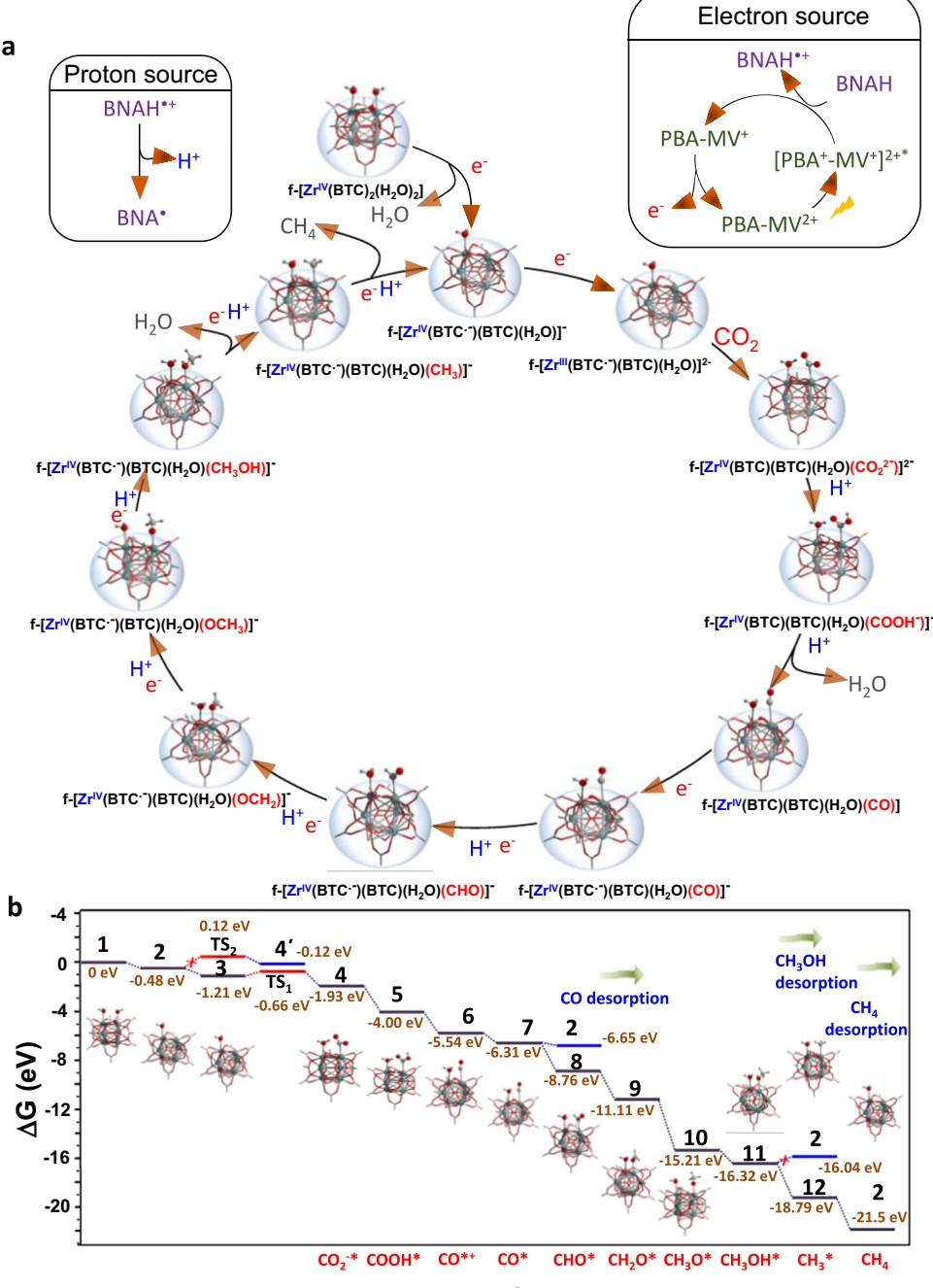

**Fig. 7 | Proposed CO₂ reduction catalytic mechanism for MOF-808-PBA-MV.**
**a** Schematic illustration of the proposed reaction mechanism for the CO₂-to-CH₄ photo-conversion on MOF-808-PBA-MV in water (f = Zr-cluster fragment). **b** Free
energy diagram for the photocatalytic CO₂ reduction to CH₄ promoted by MOF-808-PBA-MV.

## Methods

### Reagents

All the reagents were commercially available and used as such without further purification. Zirconium tetrachloride ($ZrCl_4·8H_2O$, purity ≥ 98%), Zirconium oxychloride octahydrate ($ZrOCl_2·8H_2O$, purity ≥ 99.5%) 1,3,5-Benzenetricarboxylic acid (BTC), 1-Pyrenebutyric acid (PBA), and Methyl viologen (MV) were purchased from Sigma-Aldrich Chemical Co. Ltd. Dimethylformamide (DMF), formic acid (purity > 98%), anhydrous acetone and anhydrous methanol were purchased from Spectrochem Pvt. Ltd. All the photophysical studies were carried

out using HPLC grade solvents. ¹³CO₂ cylinder was purchased from Sigma-Aldrich Chemical Co. Ltd.

### Preparation of mesoporous MOF-808

Mesoporous MOF-808 was synthesized by following a slightly modified reported procedure[33]. Benzene-1,3,5-tricarboxylic acid ($H_3BTC$) (0.282 g, 1.3 mmol) and $ZrCl_4$ (0.932 g, 3.9 mmol) were dissolved in 20 mL of formic acid and 40 mL of DMF and placed in a screw-capped glass vial and heated at 130 °C for one day. Afterward, the mixture was cooled to room temperature in the oven, and the precipitate was

filtered by vacuum filtration and washed with DMF (2 × 40 mL), acetone (2 × 40 mL), and methanol (3 × 40 mL). The resulting precipitate was dried in the air followed by activating at 150 °C for 24 h under a vacuum to yield the desired compound. $^1$H solution NMR spectra of the digested MOF-808 (400 MHz, $D_2O$, ppm) δ: 8.1 (s, BTC), 8.2 (s, HCOOH), peak area ratio (BTC: HCOOH) = 6.0:5.0. Isolated yield 80% based on Zr.

### Preparation of MOF-808-PBA

In a 15 mL glass vial, 5 mL 0.1 M DMF/MeOH (1:2) solution of pyrenebutyric acid (PBA) molecule was added to 0.100 g of activated MOF-808. The reaction mixture was placed in a 60 °C oven for 15 days. Notably, during this period, the PBA molecule solution was replaced after 3 days. After 15 days, a pale yellow color MOF was collected by filtration and washed several times with methanol to remove the unreacted PBA molecule. The exchanged product was dried in an oven at 60 °C under a vacuum. $^1$H-NMR spectra of digested MOF-808-PBA (400 MHz, $D_2O$, ppm) δ: 8.1 (s, BTC), 8.21 (s, HCOOH), peak area ratio (BTC:HCOOH) = 6.0:2.0.

### Preparation of MOF-808-PBA-MV

PBA exchanged MOF, MOF-808-PBA (70 mg) was added to a solution containing Methyl viologen (5 mg, 23.8 mmol). A dark brown color compound was formed immediately after adding MV. The suspension was kept in a 60 °C oven for one day. The resulting powder sample was filtered and washed with water to remove any unreacted methyl viologen, affording MOF-808-PBA-MV.

### Physical measurement

Elemental analysis was performed in Thermo Scientific Flash 2000 CHN analyzer. Bruker FT-IR spectrometer was used to record infrared spectra. Samples were prepared as pellets using KBr for IR measurement. $^1$H NMR spectra were recorded on Bruker AVANCE-400 spectrometer (at 400 MHz) with chemical shifts recorded as ppm. Mettler Toledo-TGA 850 instrument was used to measure the thermal stability in the $N_2$ atmosphere within the temperature range 30–800 °C at a heating rate of 5 °C/min. Powder X-ray diffraction patterns for all the MOF samples were recorded in Bruker D8 discover instrument using Cu-Kα radiation. Morphological studies have been carried out using Bruker Leica-S440I field emission scanning electron microscope (FE-SEM) by placing samples on a silicon wafer under a high vacuum with an accelerating voltage of 100 kV. Transmission electron microscopy (TEM) analysis has been performed using JEOL JEM-3010 with an accelerating voltage at 300 kV. $N_2$ adsorption measurements were carried out at 77 K in a QUANTA-CHROME QUADRASORD-SI analyzer. Samples were degassed at 150 °C and $1 \times 10^{-1}$ Pa vacuum for 12 h before performing sorption isotherm measurement. UV–vis studies were performed in the solid state using Perkin Elmer Model Lambda 900 spectrophotometer instrument. The photoluminescence properties were performed in Fluorolog 3.21 spectrofluorimeter (Horiba Jobin-Yvon) instrument. The photoluminescence (PL) quantum yield was determined by an absolute method using the same fluorescence spectrometer equipped with a 120 mm integrating sphere with a BENFLEC-coated inner face (Horiba Instruments). Fluorescence decay profiles were recorded using a Horiba Delta Flex time-correlated single-photon-counting (TCSPC) instrument. A 405 nm laser diode with a pulse repetition rate of 1 MHz was used as the light source. The instrument response function (IRF) was collected using a scatterer (Ludox AS40 colloidal silica, Sigma-Aldrich).

### Photocatalytic reaction

The stock solution was prepared by dispersing 7 mg of finely powdered sample in 30 mL water and sonicated for 30 min. In the photocatalytic reduction of $CO_2$ for each experiment, 4 mL stock solution was taken, and 2 mL water was added. The experiments were carried out in a 30 mL stopper vessel with magnetic stirring containing 6 mL of

catalyst solution along with 9.3 µM of BNAH. Highly pure $CO_2$ (99.99%) gas was passed into the reactor containing a well-dispersed catalyst solution for 40 min. Then, 1 mL TEA was added to the reaction mixture as a base and sealed the setup making a total volume of 7 mL of the reaction mixture and 23 mL of void space. The reaction mixture was irradiated with a 300 W Xenon lamp (Newport) through a visible bandpass filter (400–800 nm) with constant stirring. We measured the intensity of the light source using a power meter (Newport-843-R). The measured intensity during the catalytic reaction ($\lambda = 400–800$) is 100 mW/cm$^2$ (Note: The distance between the light source and the reactor is 8 cm.). The amount of CO and $CH_4$ evolved was analyzed using gas chromatography–mass spectrometry (SHIMAZU GC-2010 PLUS). CO and $CH_4$ were quantified by using RT® Molecular sieve 5 A column (45 m, 0.32 mm ID, 30 µm df) with a mass detector. The calibration was done by a standard gas mixture of $H_2$, CO, and $CH_4$ of different concentrations at ppm-level. Importantly, GC–MS has a detection limit of 1.0 ppm for $H_2$, CO, and $CH_4$. After the photocatalysis, the reaction mixture was filtered to remove the residual solid, and the solution was further analyzed to determine the amount of liquid product. The liquid product was detected using a RID detector in High-performance liquid chromatography (Agilent Infinity1260) with Hi-Plex H column (300 × 7.7 mm) and 5 mM $H_2SO_4$ aqueous solution as eluent. The liquid product was also verified with the NMR technique using the solvent suppression method. For $^{13}CO_2$ and CO reduction reactions, we purged the reaction mixture with $^{13}CO_2$ and CO, respectively. Ultraviolet photoelectron spectroscopy (UPS) was performed using a thermo-scientific K-alpha X-ray photoelectron spectrometer (Source He(I)−21.22 eV).

### Computational details

The density functional theoretical (DFT) calculations are performed using the Gaussian 16 package of programs[66]. The plausible catalytic cycle and the corresponding free energy diagram for the photocatalytic $CO_2$ reduction to $CH_4$ by π-stacked PBA-MV moiety integrated inside the MOF pore as a photosensitizer and the MOF, f-[$Zr^{IV}(BTC)_2(H_2O)_2$] itself as catalyst (where f = $Zr_5(\mu_3-O)_5(\mu_3-OH)_3(HCOO)_5$) are established from the theoretical study. The optimizations are performed utilizing B3LYP[67–72] exchange-correlation functional along with 6–31G(d) basis set for all atoms except Zr. LANL2DZ is used as a basis set as well as ECP for Zr[73,74]. The polarizable continuum model (PCM) is utilized to account for the solvation effect of water[75]. Grimme's d3 dispersion is also used to tackle weak interactions[76]. The harmonic vibrational frequency analysis of the optimized geometries is performed to confirm the nature of stationary points. The absence of any imaginary vibrational mode in the frequency calculations of all the optimized intermediates suggest the optimized structures to be the minima on the potential energy surface (PES). The electronic absorption spectra are calculated using the time-dependent DFT (TDDFT) method. For this, B3LYP exchange-correlation functional along with 6-31+G ($d,p$) basis set is utilized, whereas the solvent effect (water) and dispersive interactions are tackled by PCM and Grimme's d3 methods, respectively. Molecular orbital (MO) pictures and the spin density distribution plots are generated using GaussView 6.0.16[77].

## Data availability

The data that support the findings of this study are available from the corresponding author upon reasonable request. Source data are provided with this paper.

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

## Acknowledgements

The authors are grateful to Prof. S. Sampath, IISC, for UPS measurements. The authors are grateful for the support and the resources provided by the "PARAM Yukti Facility" under the National Supercomputing Mission, Government of India at the JNCASR, Bangalore. S.K. and F.A.R. thanks CSIR (Government of India) for the fellowship. S.B. thanks JNCASR for the fellowship. TKM acknowledges the Department of Science and Technology (DST, project no. CRG/2019/005951; SPR/2021/000592) RAKCAM (from UAE), SSL, ICMS, SAMat research facility, and JNCASR for financial support. All authors also acknowledge Life Sciences Research, Education, and Training at JNCASR (project no. DBT/JNCASR/D0004/2018/ 00159:– 4547)

## Author contributions

S.K. and T.K.M. designed the concept of this work. S.K. and S.B. performed major experiments. S.K., S.B., and T.K.M. analyzed the experimental data and wrote the manuscript. D.R. assisted during the synthesis. F.A.R. performed the computational study. S.N. performed the TA measurement. All the authors discussed the results and commented on the manuscript.

## Competing interests

The authors declare no competing interests.
