## [Peer Review File · Nature Communications]

Confining Charge-Transfer Complex in MOF for Modulating Electron Transfer Kinetics towards Photocatalytic Selective CO₂ Reduction in WaterREVIEWER COMMENTS

Reviewer #1 (Remarks to the Author):

CO₂RR for the Selective formation of C₁ feedstock is of significant importance, and the conversion from CO₂ to CH₄ highly depends on consecutive multiple electron and proton transfer processes, which makes CH₄ formation a dynamically challenging task although CH₄ is actually more thermodynamically favored than the formation routes of other CO₂RR products. To solve this issue, the authors established a heterogeneous photocatalytic system using MOF-808-PBA-MV which is obtained from the postsynthetic modification of Zr-MOF 808 and pyrene-bearing carboxylic acid and the successive charge transfer complexation with methyl viologen. The electron transfer from charge-transfer complex pyrene(PBA)...methyl viologen(MV²⁺) to the CO₂RR active site of Zr defect was remarkably hastened, which gave rise to the enhanced CO₂RR to CH₄ in aqueous system, and also depressed the competitive HER process. The related photochemical/physical processes were well experimentally characterized and theoretically rationalized by DFT. Comprehensively speaking, this work is very interesting and worthy of publication in Nat. Commun. after solving several potential issues and further illustrating several key points. Thus, I suggest to give a minor revision before the final decision.

The detailed comments are listed as below:

- (1) what is the precise ratio between pyrene(PBA)...methyl viologen(MV²⁺) in the CT complex? And the precise amount of encapsulated MV²⁺? And the digested NMR sample was not clear enough to show those information.
- (2) The carboxylic free defect Zr-O cluster works as the CO₂ accessible active site, thus the higher density of such kind of Zr defect sites will favor the CO₂RR efficiency. On the other side, more efficient electron injection from the dangling PBA...MV²⁺ CT complex requires higher density of uploaded PBA-carboxylic acid to occupy such active Zr defect site in the postsynthetic procedure. These two requirements are contradictive to each other, so how to balance the ratio between the uploaded PBA and the residual Zr defect sites?
- (3) Regarding the VIP claim of photogenerated Zr(III) EPR signal ($g = 2.01$) of Fig. 4b, it is doubtful that the transition metal signal would exhibit this narrow EPR peak, why? Compared with the conventional understanding of formation of Zr(III), the photosensitizing ligand to defect node electron transfer might not definitely reduce the Zr(IV). The formation of oxygen vacancy with strong reducing ability is usually encountered in energy catalysis, which exhibits radical-like narrow EPR signal. Thus, it would be meaningful to argue that whether this signal in Fig. 4b is an oxygen vacancy that working as an in situ generated true active site.
- (4) Regarding the carboxylic free defect Zr-O cluster, the “real” CO₂RR active Zr site is proximal to H₂O molecule that coordinated to neighbouring Zr. What is the role of this proximal H₂O? To facilitate the proton transfer as a proton shuttle? To bind with hydrophilic intermediates with hydrogen bonds to allow for further e⁻/proton injection? To kick off the hydrophobic CH₄ product to enhance desorption? Those are some possible hints. Thus, deeper discussions and experimental/theoretical investigations are needed.
- (5) The consecutive multiple electron transfer is important and well demonstrated in the manuscript. In comparison, the consecutive multiple proton transfer is also vital but much less mentioned.

(6) Regarding the key claim of CT complex, methyl viologen(MV²⁺) interacted with PBA in a noncovalent pi...pi stacking. As demonstrated by the author, after donating one electron, the proton/electron mediator BNAH tended to form dimer BNA₂ which reducing ability is even stronger than the monomer BNAH, thus the dimer BNA₂ might form the analogues of methyl viologen after further donating electrons/protons, and the resulting benzyl viologen analogue generated from BNA₂ also has the possibility to form CT complex with pyrene in MOF. The exchange between MV²⁺ and the much more excess amount of (BNA₂)₂⁺ is quite possible. This is also what we would like to discuss with the author. Since it is hard to believe that MV²⁺ will be just retained after photocatalysis. If the photocatalyst can be reused without re-loading the MV²⁺, there should be something replacing MV²⁺ and work exactly in the same role.

(7) The author claimed the possible Zr(V) intermediate, the ref. will be needed.

(8) Supplementary Fig. 17, the peak of MV⁺/MV²⁺ in the anode oxidation half curve disappeared, please clarify the relationship of this phenomenon with CT complex formation.

Reviewer #2 (Remarks to the Author):

This referee found the paper to be well written with a logical material characterization process and material design rationale. The scope of analysis methods is warranted for the claims made and are coherently presented. While this specific MOF/material has been reported several times for photocatalytic CO₂ reduction recently (e.g., 10.1038/s41929-021-00665-3, 10.1021/jacs.1c03283 and 10.1039/d0ee03643a), the presented donor-acceptor charge separated state rationale and selective CO₂-to-CH₄ reduction is interesting and may warrant publication in Nature Communications

The material synthesis and characterization is well-documented and meets state-of-the-art criteria for the field. The Supporting Information is extensive and, in most cases (see technical comments below), should be detailed further to warrant reproducibility.

However, some key mechanistic and technical points are not made clear enough in the manuscript's current form and strongly impact the authors' conclusions. Accordingly, this referee concludes the manuscript may be reconsidered for publication in Nature Communications, providing that the following key points can be addressed satisfactorily:

1. The charge transfer state and electron cascade as currently explained are unclear and somehow confusing. For the D-A charge transfer system, further characterization of this species and/or clarification should be provided on its nature.

From the text and EPR experiments in S16, it is suggested that a radical species is formed immediately after introduction of MV in the MOF in the dark under no stimulus, with this new species displaying an absorption at around 590 nm. This is very surprising and, if confirmed, remarkable. First, this referee wishes to know if the MV-loading experiments and the resulting materials handling was strictly done in

the dark? The referee suspects that a photoinduced charge separation resulting in a long-lived radical ion pair would make sense to explain the following observed behavior?

If the above behavior is confirmed, the authors ascribe this species to “the formation of charge-separated species (PBA+•-MV+•)”, as a radical ion pair, with reduced MV radical species being observed in S16 and PBA being the electron donor. However, for the rest of the MS discussion, it seems that the authors consider PBA/MV in their ground state rather than as a radical ion pair. For instance, the authors ascribed the absorption feature by TDDFT to “the charge transfer transition from PBA to MV”. Do they mean PBA+• to MV+•? As MV+• is known for its strong absorption at around 600 nm, this could rather explain the broad absorption feature. Similarly, the suggested mechanism depicts (PBA-MV)²⁺ → (PBA+•-MV+•)^{2+*} as the first step under irradiation (as for instance in fig 1). This would contradict the EPR S16 which would rather imply a (PBA+•-MV+•)²⁺ → (PBA+•-MV+•)^{2+*}. The same uncertainty arises at the TA spectroscopy experiments. Clarifications should be provided with clear differentiation between the different states and feature allocation.

2. The in-situ DRIFTS measurements raise several key fundamental mechanism questions which should be addressed in full. As these experiments are performed with dry powders, CO₂ and water vapor but without an electron source (BNAH or TEA), the occurrence of bands assigned by the authors to HCO₃⁻, CO₂⁻, COOH*, CO, CHO*, CH₂O*, and CH₃O* during irradiation is puzzling. Where do the electrons supposedly come from? Particularly as these intermediates require multi-electron reductions to be reached. How comparable are the drawn conclusions to colloidal conditions with stirring, solvent, and BNAH/TEA present?

3. Closely related to this point, Supplementary Table 6 entries 7 and 8 also raise doubts concerning the active species and mechanism. To this referee’s understanding of the table, the authors state that unloaded MOF-808 in photocatalytic media produces a mixture of H₂ and CH₄, and – more importantly – blank photocatalytic media produces significant amounts of H₂, CO and CH₄ (more than the blank MOF)? This implies that BNAH/TEA alone are active in CO₂ reduction?

4. In addition to the above points, the authors should consider performing photocatalysis control experiments and further DRIFTS analysis using CO (ideally labelled ¹³C carbon monoxide) instead of CO₂ to further support their mechanism and the origin of CH₄.

5. Following from here, more experimental data and literature support for the dual role of BNAH/TEA is needed (the experiments described in the SI, Suppl. Figure 30 and 31 are a valid starting point), as it seems the specific combination has a considerable impact on selectivity (page 10, Figure 3d).

6. Further analysis and a critical comment from the authors is warranted on mass transport limitations during photocatalysis. With such a heavily reduced BET surface area (down to 357 m²g⁻¹) for the loaded MOF samples, and substantial expected pore blocking from the D-A complex, how well can BNAH and TEA still diffuse to active sites? Particularly as a dual role is claimed. Is this the rate limiting step?

7. The impact of the photocurrent experiments is currently limited and should be further explained as

basic experimental details are missing. What is the origin of these Faradaic photocurrents in absence of electron donor? Were they any potential applied during the experiment or only open circuit potential?

8. This referee recommends considering refrain from using Tauc plot analysis for the functionalized material or performed further deconvolution to calculate the MOF-808-based band gap. As it is best used to describe (bare) semiconductor behavior, it does not apply well for modified MOF systems including molecular (non-band behaving) species such as PBA and MV. Please refer to 10.1021/acs.jpcclett.8b02892 for further detail.

9. Although not clearly stated, it seems that the excitation wavelength used for the experiment in Fig S49 is UV-based and unmatching the catalysis-effective light absorption in MOF-808-PBA-MV which is around 590 nm according to Fig 1d and S27. Additionally, the emission profile is unlikely to stem from the relevant PBA-MV species as occurring at lower wavelength than its absorption. Thus, this referee finds the experiment shown in Fig S49 unable to support the conclusions.

In addition, the following technical points should be considered:

10. The authors should clearly state in the abstract and manuscript that the CH₄ selectivity is on an electronic basis and does not refer to the absolute quantities of produced gas, as this is misleading.

11. The assignment of pore sizes to the enlarged HRTEM image in Figure 1b seems ambiguous based on the provided resolution and this reviewer suggests removing the orange assignments.

12. Why are not all PBA IR bands visible in the assembly? The authors show the inclusion of new bands at 2945, 1658 and 846 cm⁻¹ (Suppl. Figure 9), but why do the bands at e.g., 1200-1300 cm⁻¹ not appear?

13. The structural integrity of the sample post-catalysis is shown (Suppl. Figure 42-45), however an investigation on how much PBA/MV is retained would be interesting. Is the D-A complex completely stable under photocatalytic conditions? Does some leaching occur? This could be investigated by dissolving a post-catalysis assembly for ¹H NMR experiments (as done for the initial characterization).

14. Regarding the electrochemical experiments S17, could the authors comment on the redox peak at -0.38 V in MOF-808-PBA-MV. Why is it irreversible? Further cycling scan should be shown and it would be important to know to which scan the CV in S17 corresponds.

15. Further experimental details are required in the Supporting Information for photocurrent and Nyquist impedance measurements. For example, the solvent media and applied potential for photocurrent. In their current form the experiments cannot be reproduced and the value for the photocatalytic mechanism discussion is questionable.

16. Readability of the DRIFTS data could be improved by revising Figure 5 to be clearer (e.g., colors, region between 2800 and 2200 cm⁻¹ is irrelevant etc.).

17. Figure 6b features several “desorption” typos and many unexplained acronyms can be found in the MS.

18. Clarification over the “TON of 28” should be given. What is the reference here? The Zr-clusters?

Reviewer #3 (Remarks to the Author):

In this manuscript, the authors confine D-A supramolecular complex into MOF-808 for photocatalytic CO₂ reduction. The activity and selectivity of CO₂-to-CH₄ are impressive. Notably, the photocatalyst exhibits an around 1mmol/g CH₄ production rate under natural sunlight irradiation. Despite the exciting high photocatalytic results, the mechanism of CH₄ production is unclear. Overall, some key issues need to be addressed to support the authors' conclusion before reconsideration in Nature Communications.

1. The authors claim the Zr-oxo cluster is the catalytic site for CH₄ production. However, almost all previous reports have shown that Zr-oxo clusters in MOFs, behaving photocatalytic sites, catalyze the reduction of CO₂ to produce HCOOH or CO --- 2e products. Unexpectedly, in this work, CH₄ can be generated in almost all control experimental conditions – this is difficult to understand. This is the most important point and should be well explained with related evidences.
2. Page. 10, Line 257: In the presence of only TEA or BNAH, CO will be produced. Why coexistence of TEA and BNAH does not produce CO? The detailed explanations with evidences behind the results should be provided.
3. Page. 6, Line 137: The authors think the PBA molecules will occupy the micropore of MOF-808. In my opinion, PBA may occupy mesopores more easily due to the steric hindrance. Can the authors give some possible explanations?
4. Page. 6, Line 149: The authors ascribe the PL spectrum change of MOF-808-PBA to the close proximity of PBA molecules. Whether to lower the loading of PBA, the PL spectra will be similar to PBA monomer?
5. Page. 6, Line 156: How to get the quantum yield of 23.88%?
6. The unit the of x-axis in Supplementary Fig. 3. Should be °C.

Thank you for sending us the reviewer reports on our manuscript titled " **Confining Charge-Transfer Complex in MOF for Modulating Electron Transfer Kinetics towards Photocatalytic Selective CO₂ Reduction in Water**" bearing manuscript id: **NCOMMS-22-08405-T**. We have thoroughly revised our manuscript in accordance with the comments of reviewers, included experimental evidences and here we provide a point-by-point response to all the comments made by the reviewers.

REVIEWER REPORT(S):

Reviewer #1 (Remarks to the Author):

CO₂RR for the Selective formation of C1 feedstock is of significant importance, and the conversion from CO₂ to CH₄ highly depends on consecutive multiple electron and proton transfer processes, which makes CH₄ formation a dynamically challenging task although CH₄ is actually more thermodynamically favored than the formation routes of other CO₂RR products. To solve this issue, the authors established a heterogeneous photocatalytic system using MOF-808-PBA-MV which is obtained from the postsynthetic modification of Zr-MOF 808 and pyrene-bearing carboxylic acid and the successive charge transfer complexation with methyl viologen. The electron transfer from charge-transfer complex pyrene (PBA)...methyl viologen (MV²⁺) to the CO₂RR active site of Zr defect was remarkably hastened, which gave rise to the enhanced CO₂RR to CH₄ in aqueous system, and also depressed the competitive HER process. The related photochemical/physical processes were well experimentally characterized and theoretically rationalized by DFT. Comprehensively speaking, this work is very interesting and worthy of publication in Nat. Commun. after solving several potential issues and further illustrating several key points. Thus, I suggest to give a minor revision before the final decision. The detailed comments are listed as below:

We are thankful to the reviewer for appreciating our work and recommending for publication in Nat. Commun. after minor revision. The constructive comments are very helpful for improving the quality and value of this article.

(1) What is the precise ratio between pyrene (PBA)...methyl viologen (MV²⁺) in the CT complex? And the precise amount of encapsulated MV²⁺? And the digested NMR sample was not clear enough to show those information.

Response: We thank the reviewer for raising the concern. To understand the precise ratio of PBA and MV²⁺, we have again performed the ¹H NMR measurement in the JEOL-600 spectrometer (600 MHz). The ¹H NMR spectrum of digested **MOF-808-PBA-MV** in KOH/D₂O/DMSO-d₆ showed a prominent peak at 8.31 ppm attributed to the formate ligand, and integration of this peak suggested one formate unit remained in the framework after the post-synthetic modification (Supplementary Fig. 15). Therefore, it is strongly suggested four formate units were exchanged with PBA molecules. It is worth mentioning that the addition of DMSO-d₆ was required to dissolve the PBA molecule. Additionally, there are several overlapping peaks in the region of 7.85 to 8.33 ppm, which corresponds to the aromatic protons of PBA, MV²⁺ and BTC ligand. To further conclude the precise amount of MV²⁺ in the framework, we carefully looked to the aliphatic region for the CH₃ proton peak of MV²⁺, which appeared at 4.49 ppm and integration of this particular peak suggested that 18 protons are present in the digested NMR. Hence, it implies the precise ratio of PBA and MV²⁺ is 4:3 per formula unit of **MOF-808-PBA-MV**. The new NMR data has been included in the revised manuscript (Supplementary Fig. 15) and also provided below.

Supplementary Fig. 15 ^1H NMR spectra for digested **MOF-808-PBA-MV** in $\text{KOH}/\text{D}_2\text{O}/\text{DMSO-d}_6$.

(2) The carboxylic free defect Zr-O cluster works as the CO_2 accessible active site, thus the higher density of such kind of Zr defect sites will favor the CO_2RR efficiency. On the other side, more efficient electron injection from the dangling PBA... MV^{2+} CT complex requires higher density of uploaded PBA-carboxylic acid to occupy such active Zr defect site in the postsynthetic procedure. These two requirements are contradictive to each other, so how to balance the ratio between the uploaded PBA and the residual Zr defect sites?

Responses: We are obliged to address and clarify the reviewer's point of concern. In this work, we prepared a mesoporous Zr-MOF-808, where one formate is missing from each Zr_6 cluster present per formula unit of MOF-808. So, five formates are accessible for the ligand exchange in the SBU, and the exchangeable sites are fixed. In our experimental condition, we are able to exchange four formates with PBA ligand, and that is the maximum possibility for exchanging due to steric hindrance inside the MOF pore. A similar observation was

previously reported by other groups, where formates were exchanged by different ligands (*Nat. Commun.* 2018, **9**, 187; *Nat. Catal.*, 2021, **4**, 719-729; *Energy Environ. Sci.*, 2021, **14**, 2429-2440). Afterwards, we introduce the MV²⁺ to form the CT complex, and from the ¹H NMR, we found the precise ratio of PBA and MV²⁺ is 4:3 (Supplementary fig. 15). We think this is the maximum possible CT complexation to achieve high CO₂RR efficiency. Also, each Zr₆-cluster has two fixed water molecules, and water bound Zr⁴⁺ sites acts as active catalytic site as observed by DFT calculation. During the CO₂RR catalysis water molecule is replaced by CO₂ molecule.

Additionally, we have performed DFT calculations to properly address this problem. First of all, the formula of formate-defected based Zr-MOF-808 can be represented as [Zr₆(μ₃-OH)₃(μ₃-O)₅(HCOO)₅(BTC)₂(H₂O)₂], where each missing formate ligand is replaced by two water molecules. Now the question is, in the course of the ligand exchange, whether the two water molecules will be replaced or one of the remaining formate ligands will be exchanged. During the formate exchange by PBA (PBA-H), the anionic formate ligand will abstract proton from PBA (PBA-H) and will be released as neutral formic acid (HCOOH), whereas the resulting PBA anionic ligand will occupy the sites vacated by formate anionic ligand.

(To show the neutral and anionic form, neutral PBA acid is represented as PBA-H)

Whereas, if two water molecules are replaced by PBA (PBA-H), then one of the water molecules will abstract proton from PBA (PBA-H) to make it anionic, which will occupy the vacated sites.

From DFT calculations, it can be seen that though both types of exchanges are endergonic, the exchange of water molecules from the defected sites is the highly endergonic one. This suggests a high difficulty level of the exchange of water molecules from the defected sites by PBA acid. Thus, from all these, we can reasonably exclude the possibility of water exchange

from the defected sites, which automatically rules out the occurrence of the aforementioned contradiction mentioned by the reviewer.

(3) Regarding the VIP claim of photogenerated Zr(III) EPR signal ($g=2.01$) of Fig. 4b, it is doubtful that the transition metal signal would exhibit this narrow EPR peak, why? Compared with the conventional understanding of formation of Zr(III), the photosensitizing ligand to defect node electron transfer might not definitely reduce the Zr(IV). The formation of oxygen vacancy with strong reducing ability is usually encountered in energy catalysis, which exhibits radical-like narrow EPR signal. Thus, it would be meaningful to argue that whether this signal in Fig. 4b is an oxygen vacancy that working as an in situ generated true active site.

Response: We understand the reviewer's concern regarding the formation of oxygen vacancy and the corresponding EPR signal. To gain a deeper understanding, we have again performed EPR experiment with pristine **MOF-808** as well as post-modified **MOF-808-PBA-MV**, and the EPR spectrum of the sample was monitored *in situ* under daylight conditions and irradiated to an external visible light source ($\lambda=380-780$ nm) (Fig R1). Under daylight conditions, **MOF-808-PBA-MV** showed a weak signal of CT complex at $g=1.98$, which can be attributed to the formation of $MV^{+\bullet}$ from MV^{2+} (Fig. 4b)(*Chem. Sci.*, 2015, **6**, 2922-2927, *Chem. Commun.*, 2014, 50, 13544; *Angew. Chem., Int. Ed.*, 2007, 46, 3249). Interestingly, when the sample was irradiated to external visible light, the intensity of the signal corresponding to the CT band increased drastically. At the same time, a new EPR signal appeared at $g=2.01$, which can be accredited to the formation of Zr^{3+} ion. This can be attributed to the electron transfer from the CT complex to the Zr_6 -oxo cluster (Fig. 4b). This phenomenon is well documented in the previous literatures (*J. Am. Chem. Soc.* 2015, **137**, 13440-13443; *J. Mater. Chem. A*, 2016, **4**, 2657–2662; *Phys. Chem. Chem. Phys.*, 2000, **2**, 2635-2639; *Chem. Commun.*, 2012, **48**, 11656–11658). Next, the pristine **MOF-808** was investigated under daylight and light irradiated conditions, where it showed no EPR signal (Fig. R2), which further confirms the origin of the EPR signal due to the presence of CT complex in **MOF-808-PBA-MV**. If the active center is the oxygen vacancy, in that case, there should be a distinct EPR signal in both pre and post-modified **MOF-808**, and there should be no increment in signal intensity with light irradiation as there will be no change in the number of oxygen vacancies.

Fig. R1 Images of the set-up utilized for EPR measurement. Condition i: Under normal daylight. Condition ii: Irradiated with visible light source (Leica KL1600 LED).

Fig. 4b EPR spectra of MOF-808-PBA-MV under daylight (Condition i) and irradiated with the LED light source (Condition ii).

Fig. R2 EPR spectra of **MOF-808** under daylight (Condition i) and irradiated with the LED light source (Condition ii).

(4) Regarding the carboxylic free defect Zr-O cluster, the "real" CO₂RR active Zr site is proximal to H₂O molecule that coordinated to neighbouring Zr. What is the role of this proximal H₂O? To facilitate the proton transfer as a proton shuttle? To bind with hydrophilic intermediates with hydrogen bonds to allow for further e/proton injection? To kick off the hydrophobic CH₄ product to enhance desorption? Those are some possible hints. Thus, deeper discussions and experimental/theoretical investigations are needed.

Response: We really appreciate the reviewer for the insightful comment. According to the suggestion of the reviewer, we have performed DFT calculation to understand the role of H₂O. First of all, since it is a formate defect which is a bidentate ligand, upon generation of each formate defect, there will be two vacant sites on two adjacent Zr-centers in the SBU simultaneously, which will be occupied by two solvent water molecules. Either one of these water molecules can be vacated at a time to carry out the catalysis, and the remaining one can be called the proximal H₂O.

From the hints provided by the reviewer, we have tried to find out the role of the proximal H₂O in the catalysis by DFT calculation.

(a) Checking the role of proximal H₂O as a proton shuttle:

We have checked all the intermediate steps involving proton transfer whether or not the hydrogen of the proximal water molecule and the proton abstracting atom of the bound

intermediate are sufficiently closer to transfer the proton. We have found out that the aforementioned distance for the intermediates going to be protonated is as follows:

From the above results, it can be seen that for none of the above intermediates, the proximal H₂O is not sufficiently closer to the proton abstracting atom to act as a proton shuttle.

(b) Checking whether the proximal H₂O acts to bind with hydrophilic intermediates with hydrogen bonds to allow for further e/proton injection:

From the optimized intermediates obtained from DFT calculation, few intermediates are found to be having close proximity between the hydrogen of proximal H₂O and the bound hydrophilic intermediates to form hydrogen bonds such as follows:

Thus the proximal H₂O obviously plays a role in stabilizing some of the intermediates by binding with hydrophilic intermediates with hydrogen bonds which will allow further e/proton injection

(c) Checking whether the proximal H₂O acts to kick off the hydrophobic CH₄ product to enhance desorption:

From DFT calculations, it is found that in the presence of the proximal H₂O, desorption of hydrophobic CH₄ becomes slightly more feasible ($\Delta G = -2.71 \text{ eV}$), which reveals the positive

impact of the existing proximal H₂O in releasing off the hydrophobic CH₄ product to enhance desorption. The last part has been included in the revised manuscript (Supplementary Fig. 56).

(5) The consecutive multiple electron transfer is important and well demonstrated in the manuscript. In comparison, the consecutive multiple proton transfer is also vital but much less mentioned.

Response: We understand the reviewer's concern. Ideally, there are several competitive proton sources, as mentioned below:

Fig. R3 Proton donating process from different proton sources with corresponding Gibbs free energy values (ΔG).

However, as we can see from the DFT calculation, BNAH is found to be having much better proton donating capability compared to its other counterparts (Fig R3). In this aspect, though water is present in bulk, it is very unlikely to be acting as a proton source due to its high endergonic ($\Delta G=+4.47$ eV) proton donation process. Furthermore, since BNAH is providing the electron to the catalyst, it will be in close proximity to the catalytic centre. Thus the subsequent proton donation will be much easier from proximal BNAH⁺ despite the presence of the bulk amount of water as solvent.

(6) Regarding the key claim of CT complex, methyl viologen (MV²⁺) interacted with PBA in a noncovalent π - π stacking. As demonstrated by the author, after donating one electron, the proton/electron mediator BNAH tended to form dimer BNA₂ which reducing ability is even stronger than the monomer BNAH, thus the dimer BNA₂ might form the analogues of methyl viologen after further donating electrons/protons, and the resulting benzyl viologen analogue generated from BNA₂ also has the possibility to form CT complex with pyrene in MOF. The exchange between MV²⁺ and the much more excess amount of (BNA₂)²⁺ is quite possible. This is also what we would like to discuss with the author. Since it is hard to believe that MV²⁺ will be just retained after photocatalysis. If the photocatalyst can be reused without re-loading the MV²⁺, there should be something replacing MV²⁺ and work exactly in the same role.

Response: We appreciate the reviewer's concern. In response to the reviewer's comment, we have performed DFT calculation. We have calculated the π - π stacking stabilization energy between PBA and MV²⁺ as well as between PBA and BNA₂.

The exchange of MV²⁺ from already formed π - π stacked PBA-MV²⁺ by BNA₂ :

From eqn 2 - eqn 1:

For PBA-MV²⁺, π-π stabilization energy is calculated to be 5.13 kcal/mol, whereas that for PBA-BNA₂ is 2.96 kcal/mol, suggesting higher stability of PBA-MV²⁺ complex compared to PBA-BNA₂ complex. So, the exchange of already π-stabilized MV²⁺ from PBA-MV²⁺ complex by BNA₂ will be difficult, which is also well supported by the endothermic exchange process of MV²⁺ from PBA-MV²⁺ complex by BNA₂ (ΔG=+2.17 kcal/mol) as shown in above equations (eqn 1-3). Thus, the substantial exchange of MV²⁺ from the PBA-MV²⁺ complex by BNA₂ to form the PBA-BNA₂ complex can be ruled out. Furthermore, the difficulty of forming PBA-BNA₂ π-complex can also be easily seen from the highly distorted non-planar structure of BNA₂ (optimized via DFT calculation) (Fig. R4). Moreover, the formation of the PBA-BNA₂ complex via π-π stacking requires more space to accommodate, which is also a reason not to form the PBA-BNA₂ complex inside the MOF pore. Additionally, we have also performed the ¹H NMR experiment with digested **MOF-808-PBA-MV** after the photocatalysis experiment. We found the integration ratio remained unchanged with the pristine MOF, which suggested no exchange between MV²⁺ and the BNA₂ occurred during catalysis (see comment 13 of reviewer 2).

Fig. R4 Optimized π- π stacked model between PBA with MV²⁺ or BNA₂.

(7) The author claimed the possible Zr(V) intermediate, the ref. will be needed.

Response: We thank the reviewer for giving us the opportunity to clear the doubt about the formation of Zr^V intermediate. In our catalytic cycle, after one-electron reduction of f-

$[\text{Zr}^{\text{IV}}(\text{BTC})_2(\text{H}_2\text{O})_2]$ (**1**) generates $\text{f-}[\text{Zr}^{\text{IV}}(\text{BTC}^{\cdot-})(\text{BTC})(\text{H}_2\text{O})]^-$ (**2**) with the removal of one water molecule ($\Delta\text{G} = -0.48 \text{ eV}$). Moreover, $\text{f-}[\text{Zr}^{\text{IV}}(\text{BTC}^{\cdot-})(\text{BTC})(\text{H}_2\text{O})]^-$ (**2**) is the active species for the catalytic cycle. Next, there are two possible pathways to form CO_2 -bound species $\text{f-}[\text{Zr}^{\text{IV}}(\text{BTC})(\text{BTC})(\text{H}_2\text{O})(\text{CO}_2^{2-})]^{2-}$ (**5**).

Possibility 1: The reduction of $\text{f-}[\text{Zr}^{\text{IV}}(\text{BTC}^{\cdot-})(\text{BTC})(\text{H}_2\text{O})]^-$ (**2**) followed by oxidative addition of CO_2 to $\text{f-}[\text{Zr}^{\text{III}}(\text{BTC}^{\cdot-})(\text{BTC})(\text{H}_2\text{O})]^{2-}$ (**3**) may occur.

Possibility 2: Oxidative addition of CO_2 to $\text{f-}[\text{Zr}^{\text{IV}}(\text{BTC}^{\cdot-})(\text{BTC})(\text{H}_2\text{O})]^-$ (**2**) followed by reduction of $\text{f-}[\text{Zr}^{\text{V}}(\text{BTC})(\text{BTC})(\text{H}_2\text{O})(\text{CO}_2^{2-})]^-$ (**4'**) may occur.

We found the binding of CO_2 with $\text{f-}[\text{Zr}^{\text{III}}(\text{BTC}^{\cdot-})(\text{BTC})(\text{H}_2\text{O})]^{2-}$ (**3**) is an exergonic process ($\Delta\text{G} = -0.72 \text{ eV}$) with a lower activation barrier ($\Delta\text{G}^\ddagger = 0.55 \text{ eV}$) to generate CO_2 bound species whereas the CO_2 binding pathway with $\text{f-}[\text{Zr}^{\text{IV}}(\text{BTC}^{\cdot-})(\text{BTC})(\text{H}_2\text{O})]^-$ (**2**) is an endergonic ($\Delta\text{G} = +0.36 \text{ eV}$) with a higher activation barrier ($\Delta\text{G}^\ddagger = 0.60 \text{ eV}$) (Fig. 6b and Supplementary Table 15-16). This inferred that the CO_2 binding to $\text{f-}[\text{Zr}^{\text{III}}(\text{BTC}^{\cdot-})(\text{BTC})(\text{H}_2\text{O})]^{2-}$ (**3**) is the major pathway under the experimental conditions. This is quite expected because a lower oxidation state on Zr prior to the binding of CO_2 will facilitate the oxidative addition of CO_2 . Hence, we want to say, the formation of $\text{f-}[\text{Zr}^{\text{V}}(\text{BTC})(\text{BTC})(\text{H}_2\text{O})(\text{CO}_2^{2-})]^-$ species as an intermediate is quite unexpected, and CO_2 bound species is formed via $\text{f-}[\text{Zr}^{\text{III}}(\text{BTC}^{\cdot-})(\text{BTC})(\text{H}_2\text{O})]^{2-}$ (**3**) intermediate pathway. Additionally, the high valent Zr^{V} is also non-existent. Here, we have performed DFT calculation to rule out the formation of Zr^{V} species as the intermediate as it is thermodynamically unfavourable. However, in the main text, we still consider the Zr^{V} in the discussion where spin density calculation further reinforces the non-existence of Zr^{V} oxidation state (Supplementary Fig. 55).

(8) Supplementary Fig. 17, the peak of $\text{MV}^{\cdot+}/\text{MV}^{2+}$ in the anode oxidation half curve disappeared; please clarify the relationship of this phenomenon with CT complex formation.

Response: We thank the reviewer for the suggestion. Methyl viologen undergo two consecutive one-electron reductions to the radical cation species ($\text{MV}^{\cdot+}$) at $E_{1/2}^1 = -0.38 \text{ V}$ vs Ag/AgCl and neutral species (MV^0) at $E_{1/2}^2 = -0.78 \text{ V}$ vs Ag/AgCl (Supplementary Fig. 20). According to the reviewer's suggestion, we looked at and rechecked the CV very carefully. We found that both the cathodic and anodic peaks correspond to the generation of $\text{MV}^{\cdot+} \rightleftharpoons \text{MV}^{2+}$ and $\text{MV} \rightleftharpoons \text{MV}^{\cdot+}$ are present (Supplementary Fig. 21). However, the peak

intensity corresponding to $MV^{•+} \rightarrow MV^{2+}$ was significantly decreased due to the formation of stable $PBA^{•+}-MV^{•+}$ CT interaction inside the MOF pore. To further clarify the decrease of the anodic peak corresponding to $MV^{•+} \rightarrow MV^{2+}$ formation, three consecutive CV cycles were scanned, where we noticed after 1st segment, the peak current corresponding to $MV^{2+} \rightleftharpoons MV^{•+}$ was significantly decreased due to the formation of stable $PBA^{•+}-MV^{•+}$ CT complex inside the MOF pore. This observation firmly signifies that it is only possible if the generated $MV^{•+}$ was consumed by $PBA^{•+}$ to form CT complex. This observation firmly signifies that it is only possible if the generated $MV^{•+}$ was stabilized by $PBA^{•+}$ in the CT complex. Additionally, to further clarify the CT complex role, we have prepared an analogous MV^{2+} (1-(carboxypentyl)-1'-methyl-4,4'-bipyridinium) incorporated MOF-808 named as **MOF-808-MV-COOH** (Scheme-R1). The cyclic voltammetric study of this compound showed both the reversible peak of $MV^{•+} \rightleftharpoons MV^{2+}$ and $MV \rightleftharpoons MV^{•+}$ are present in this compound due to the absence of CT complex (Fig.R5).

Supplementary Fig. 21 Cyclic voltammogram of **MOF-808-PBA-MV** recorded in MeCN (0.1 M TBAPF₆).

Scheme-R1 Synthetic scheme: synthesis of **MOF-808-MV-COOH** via post-synthetic linker exchange.

Fig. R5 Cyclic voltammogram of **MOF-808-MV-COOH** recorded in MeCN (0.1 M TBAPF₆).

Reviewer #2 (Remarks to the Author):

This referee found the paper to be well written with a logical material characterization process and material design rationale. The scope of analysis methods is warranted for the claims made and are coherently presented. While this specific MOF/material has been

reported several times for photocatalytic CO₂ reduction recently (e.g., 10.1038/s41929-021-00665-3, 10.1021/jacs.1c03283 and 10.1039/d0ee03643a), the presented donor-acceptor charge separated state rationale and selective CO₂-to-CH₄ reduction is interesting and may warrant publication in Nature Communications. The material synthesis and characterization is well-documented and meets state-of-the-art criteria for the field. The Supporting Information is extensive and, in most cases (see technical comments below), should be detailed further to warrant reproducibility. However, some key mechanistic and technical points are not made clear enough in the manuscript's current form and strongly impact the authors' conclusions. Accordingly, this referee concludes the manuscript may be reconsidered for publication in Nature Communications, providing that the following key points can be addressed satisfactorily:

We are grateful to the reviewer for appreciating our work. As per reviewer suggestions, we have addressed the concerns and revised the manuscript accordingly. The constructive comments are very helpful for improving the quality and value of this article.

1. The charge transfer state and electron cascade as currently explained are unclear and somehow confusing. For the D-A charge transfer system, further characterization of this species and/or clarification should be provided on its nature. From the text and EPR experiments in S16, it is suggested that a radical species is formed immediately after introduction of MV in the MOF in the dark under no stimulus, with this new species displaying an absorption at around 590 nm. This is very surprising and, if confirmed, remarkable. First, this referee wishes to know if the MV-loading experiments and the resulting materials handling was strictly done in the dark? The referee suspects that a photoinduced charge separation resulting in a long-lived radical ion pair would make sense to explain the following observed behavior? If the above behavior is confirmed, the authors ascribe this species to "the formation of charge-separated species (PBA⁺-MV⁺)", as a radical ion pair, with reduced MV radical species being observed in S16 and PBA being the electron donor. However, for the rest of the MS discussion, it seems that the authors consider PBA/MV in their ground state rather than as a radical ion pair. For instance, the authors ascribed the absorption feature by TDDFT to "the charge transfer transition from PBA to MV". Do they mean PBA⁺ to MV⁺? As MV⁺ is known for its strong absorption at around 600 nm, this could rather explain the broad absorption feature. Similarly, the suggested mechanism depicts (PBA-MV)²⁺ → (PBA⁺-MV⁺)^{2+*} as the first step under irradiation (as for instance in fig 1). This would contradict the EPR S16, which would rather imply a (PBA⁺-

$MV^{+*})^{2+} \rightarrow (PBA^{+*}-MV^{+*})^{2+*}$. The same uncertainty arises at the TA spectroscopy experiments. Clarifications should be provided with clear differentiation between the different states and feature allocation.

Response: We appreciate and thank the reviewer for the suggestion. To understand the nature of CT complex inside the MOF pore, we prepared **MOF-808-PBA-MV** under completely dark conditions by wrapping the vial with aluminium foil and designated it as **MOF-808-PBA-MV_d** (d= dark). However, it is worth mentioning that previously we prepared the sample under normal daylight conditions. Next, we measured the EPR spectra of both the compounds prepared under completely dark state as well as normal daylight conditions. **MOF-808-PBA-MV_d** was studied in the dark, where it showed no recognizable EPR signal (Supplementary Fig. 17, 19, condition i). This clearly suggests no charge-separated species formed in that state. Now, we exposed the same sample under daylight for two hours and then recorded the EPR spectrum *in situ* and also further, the sample was continued to irradiate to an external visible light source ($\lambda=380-780$ nm) (Fig.4b; condition ii, iii). After daylight irradiation **MOF-808-PBA-MV** showed a significant peak at $g= 1.98$ (Fig. 4b; condition ii). This phenomenon confirms that daylight act as a stimulus. Finally, when the sample was irradiated to external visible light, the intensity of the signal corresponding to the CT band increased drastically (condition iii). It is worth mentioning that the intensity of EPR signal corresponds to CT complex (MV^{+*}) drastically increase, which further supports the above claim that the % of a charge-separated species inside the MOF pore is very small. Interestingly, a new EPR signal appeared at $g= 2.01$, corresponding to the Zr^{III} species, which attributed to the sensitization of the Zr_6 oxo cluster through the electron transfer from the CT complex to the Zr_6 -oxo cluster. The observed phenomenon is well documented in the previous literature, such as *J. Mater. Chem. A*, 2016, **4**, 2657–2662, *J. Am. Chem. Soc.* 2015, **137**, 13440-13443; *Phys. Chem. Chem. Phys.*, 2000, **2**, 2635-2639; *Chem. Commun.*, 2012, **48**, 11656–11658. As EPR cannot quantify the charge separation percentage, we performed magnetic measurement by SQUID magnetometer to quantify the extent of charge separation. Unfortunately, we were unable to get an appreciable magnetic moment from both the sample prepared in dark as well as daylight. This also suggests very small % of the PBA-MV²⁺ CT pair is in charge separated state. Afterwards, we performed DFT ONIOM calculation for the charge-separated state and utilizing the Boltzmann distribution formula, the percentage of charge-separated species present under a completely dark state was calculated. The result

suggests a very negligible amount of charge separated state ($2.3 * 10^{-18}$ %) under completely dark state, which is well reflected in the EPR by the non-emergence of any signal.

Supplementary Fig. 17 Images of the set-up utilized for EPR spectra. Condition i: Under completely dark conditions. Condition ii: Exposed under normal daylight. Condition iii: Irradiated with visible light source (Leica KL1600 LED).

Supplementary Fig. 19 EPR spectra of MOF-808-PBA-MV_d under dark condition (Condition i).

Fig. 4b EPR spectra of MOF-808-PBA-MV under dark conditions (Condition i), exposed under visible light (Condition ii) and irradiated with a LED visible light source (Condition iii).

2. The in-situ DRIFTS measurements raise several key fundamental mechanism questions which should be addressed in full. As these experiments are performed with dry powders, CO₂ and water vapor but without an electron source (BNAH or TEA), the occurrence of

bands assigned by the authors to HCO_3^- , CO_3^{2-} , COOH^* , CO , CHO^* , CH_2O^* , and CH_3O^* during irradiation is puzzling. Where do the electrons supposedly come from? Particularly as these intermediates require multi-electron reductions to be reached. How comparable are the drawn conclusions to colloidal conditions with stirring, solvent, and BNAH/TEA present?

Response: We thank the reviewer for the insightful comment. We have performed the *in-situ* DRIFT experiment by spin-coating over a glass slide and placing it in the center of the designed reaction cell (Supplementary Fig. 53). We did the experiment by purging CO_2 and water vapour. A similar procedure was followed in several literatures, such as *Nat. Energy*, 2019, **4**, 690–699, *J. Mater. Chem. A*, 2017, **5**, 5020-5029; *Angew. Chem. Int. Ed.*, 2022, **61**, e202116094; *Nat. Commun.*, 2020, **11**, 1149; *Energy Environ. Sci.*, 2021, **14**, 2429-2440; *Energy Environ. Sci.*, 2018, **11**, 2382-2389. As we have mentioned in the supplementary table 6, **MOF-808-PBA-MV** is also catalytically active in water without any sacrificial electron donor with a production of $460 \mu\text{mol g}^{-1}$ of CH_4 along with $54 \mu\text{mol g}^{-1}$ of CO . Here, the main purpose of performing *in-situ* IR is to understand and identify the reaction intermediate in the CO_2RR process. Importantly, in the water medium, water can act as an electron and H^+ donor in the absence of BNAH and TEA. Hence, it is sufficient to observe the reaction intermediate bands by using water. In fact, it is impossible to carry out the photoreduction process in the solution phase in that *in situ* set-up. However, as per the reviewer's suggestion, we have performed *in-situ* DRIFT experiment in the presence of BNAH and TEA. To perform the experiment, we dispersed the catalyst along with BNAH and thereafter coated it over a glass surface. Before starting the experiment, TEA (10 μL) was added to the glass surface. After that, a similar procedure was followed, as mentioned in the supplementary information. The peaks that appeared at 1314 and 1502 cm^{-1} after introducing CO_2 and H_2O vapour into the reactor could be assigned to the monodentate carbonate group (m-CO_3^{2-}) (*Energy Environ. Sci.*, 2021, **14**, 2429-2440; *J. Am. Chem. Soc.*, 2017, **139**, 18044–18051). The peak at 1436 cm^{-1} corresponds to HCO_3^- group (*Nat. Energy*, 2019, **4**, 690–699). Peaks appearing after photo-irradiation at 1343 and 1610 cm^{-1} correspond to COOH^* intermediate, which is a crucial intermediate during the photochemical conversion of CO_2 to CH_4 (*J. Am. Chem. Soc.*, 2021, **143**, 16284–16292; *Nat. Energy*, 2019, **4**, 690–699). Notably, a strong band appeared at 1723 cm^{-1} , which could be attributed to the bending vibration of $\text{C}=\text{O}$ (*ACS Catal.*, 2018, **8**, 9280-9286). The absorption band at 985, 1069, 1188, and 1223 cm^{-1} are assigned to the characteristic bands of C-OH^* , CHO^* , CH_3O^* , and CH_2O^* , respectively (*J. Am. Chem. Soc.* 2018, **140**, 6474-6482; *Nat. Energy*, 2019, **4**, 690–699; *J. Mater. Chem. A*,

2017,5, 5020-5029). The peak at 1643 cm^{-1} belongs to H_2O (*ACS Catal.* 2012, 2, 1817–1828). Besides, the band at 2870 and 2992 cm^{-1} corresponds to C-H stretching frequency of CH_3^* group, and 2930 cm^{-1} corresponds to C-H stretching frequency of CH_2^* group (*ACS Catal.* 2018, 8, 9280-9286). Hence a similar pathway was followed in the presence of BNAH and TEA, as mentioned in the supplementary information.

Supplementary Fig. 53 Time-dependent *in situ* FT-IR spectra of photocatalytic CO_2RR in a mixture of CO_2 and H_2O vapour, TEA and BNAH over **MOF-808-PBA-MV** in the dark and under visible light irradiation. Irradiation time 10, 20, 30, 40, 50, 60, 70, 80 min respectively.

3. Closely related to this point, Supplementary Table 6 entries 7 and 8 also raise doubts concerning the active species and mechanism. To this referee's understanding of the table, the authors state that un-loaded MOF-808 in photocatalytic media produces a mixture of H_2 and CH_4 , and – more importantly – blank photocatalytic media produces significant amounts of

H₂, CO and CH₄ (more than the blank MOF)? This implies that BNAH/TEA alone are active in CO₂ reduction?

Response: We understand the reviewer's concern and recheck the activity by performing those control experiments. Though the combination of BNAH and TEA can produce some amount of H₂, CO and CH₄ (H₂: 274 $\mu\text{mol g}^{-1}$; CO: 97 $\mu\text{mol g}^{-1}$; CH₄: 40 $\mu\text{mol g}^{-1}$), the activity is minimal as compared to our catalyst **MOF-808-PBA-MV** (H₂: 369.3 $\mu\text{mol g}^{-1}$; CH₄: 7274 $\mu\text{mol g}^{-1}$). By comparing the activity of both systems, we can tell a combination of BNAH and TEA alone is catalytically inactive. However, the minimal activity can be justified by the following facts: BNAH absorbs visible light from 400 to 450 nm (Fig R6). The amide group of BNAH can form an adduct with CO₂ after photoexcitation followed by reductive quenching ($\Delta G = -0.19$ eV). Here, TEA plays the role of sacrificial electron donor. On the other hand, metal-organic framework with a coordinatively unsaturated metal site is hugely beneficial for CO₂ adduct formation. We have performed DFT calculation with the Zr-oxo metal cluster and found a ΔG value of -0.72 eV for CO₂ adduction formation. The theoretical results clearly indicate that in the presence of the **MOF-808**, CO₂ will mostly form an adduct with the Zr-oxo cluster. Now in the control experiment, by using pristine MOF-808, we got significantly less activity as compared to **MOF-808-PBA-MV** (Supplementary Table 6). This is justified by the fact that pristine **MOF-808** does not absorb visible light, hence lacks photo absorbing capability making the electron transfer kinetics unsteady and sluggish.

Fig. R6 UV-vis spectra of BNAH.

4. In addition to the above points, the authors should consider performing photocatalysis control experiments and further DRIFTS analysis using CO (ideally labelled ^{13}C carbon monoxide) instead of CO_2 to further support their mechanism and the origin of CH_4 .

Response: We thank the reviewer for the suggestion. We have performed a photocatalytic CO reduction reaction in a CO-saturated aqueous medium under visible light irradiation (300 W Xenon Arc lamp, $\lambda > 400$ nm) in the presence of 1-benzyl-1,4-dihydro-nicotinamide (BNAH) and triethylamine (TEA) as a sacrificial agent. It was found that the major product under the photocatalytic process (CO saturated aqueous media) was CH_4 under visible light irradiation. The CH_4 production was continuously increased with irradiation time, and a total of 10.4 mmol g^{-1} was produced in 16 h (Fig. R7). Apart from CH_4 production, a very small amount of H_2 ($386.33 \text{ } \mu\text{mol g}^{-1}$) was also generated in the reaction system. As we know, the conversion of CO_2 to CO is more thermodynamically uphill process compared to other CO_2 -reduced products. The conversion of $\text{CO} \rightarrow \text{CH}_4$ is more facile route, as it already overcomes the 2-electron uphill reduction process, which can justify the increased CH_4 formation during photocatalytic CO reduction.

Fig. R7 The amount of CH_4 and H_2 evolution by MOF-808-PBA-MV by reducing CO as a function of time under visible light irradiation in water medium using BNAH and TEA as a sacrificial electron donor.

Furthermore, *in situ* diffuse reflectance, FT-IR measurement was carried out with **MOF-808-PBA-MV** to reveal the reaction intermediate during the CO reduction process (Fig R8). A similar procedure was followed (as mentioned in the Supplementary Information) to perform *in situ* DRIFT experiment with the addition of BNAH and TEA. After photo irradiation of the **MOF-808-PBA-MV** sample for 10 min resulted in the appearance of several new peaks. The absorption band at 1522 cm^{-1} belongs to adsorbed CO (*ACS Catal.* 2012, 2, 1817–1828). The absorption band at 990, 1090, 1175, and 1230 cm^{-1} are assigned to the characteristic bands of C-OH*, CHO*, CH₃O*, CH₂O*, respectively (*J. Am. Chem. Soc.* 2018, **140**, 6474-6482; *Nat. Energy*, 2019, **4**, 690–699; *J. Mater. Chem. A*, 2017,**5**, 5020-5029). The band that appeared at 1405 and 1304 cm^{-1} corresponds to the deformation vibrations of CH₃* and CH₄ group (*ACS Catal.* 2018, 8, 9280-9286; *Anal. Methods*, 2016, 8, 756–762). Moreover, the peak at 1647 cm^{-1} corresponds to H₂O (*ACS Catal.* 2012, 2, 1817–1828). Besides, the band at 2870 and 2970 cm^{-1} corresponds to C-H stretching frequency of CH₃* group and 2925 cm^{-1} corresponds to C-H stretching frequency of CH₂* group (*ACS Catal.* 2018, 8, 9280-9286).

Fig. R8 Time-dependent *in situ* FT-IR spectra of photocatalytic CO reduction reaction in a mixture of CO₂ and H₂O vapour, TEA and BNAH over **MOF-808-PBA-MV** in the dark and under visible light irradiation. Irradiation time 10, 20, 30, 40, 50, 60, 70, 80 min, respectively.

5. Following from here, more experimental data and literature support for the dual role of BNAH/TEA is needed (the experiments described in the SI, Suppl. Figure 30 and 31 are a valid starting point), as it seems the specific combination has a considerable impact on selectivity (page 10, Figure 3d).

Response: We thank the reviewer for the suggestion. Triethylamine (TEA) has been extensively used as a sacrificial electron donor for photocatalytic CO₂ reduction. However, recent literature reports demonstrated with the use of BNAH as a sacrificial electron donor, the selectivity and efficiency of photocatalytic CO₂ reduction could be modulated (*Angew. Chem. Int. Ed.*, 2022, **61**, e202116094; *Inorg. Chem.*, 2015, **54**, 5096-5104; *Coord. Chem. Rev.*, 2018, 373, 333-356; *Nat. Chem.*, 2020, **12**, 180–185). This can be associated with the lower oxidation potential of BNAH ($E^{\circ}_{\text{ox}} = 0.57$ V vs. SCE) compared to TEA ($E^{\circ}_{\text{ox}} = 0.69$ V

vs. SCE). Hence, BNAH can easily oxidize as compared to TEA which leads to a faster electron transfer to the catalyst during photocatalysis. According to our results, we got better activity by using BNAH alone as a sacrificial agent as compared to TEA. Now the question arises about the role of TEA with BNAH as an electron donor in the photocatalytic reduction reaction. The oxidation process of BNAH is shown in Supplementary Fig. 34. In the first step, one-electron oxidation product of BNAH was formed, which has a probability of taking back the electron from the catalyst (one-electron reduced species). To suppress this back electron transfer process, TEA is required, which deprotonates the $\text{BNAH}^{\bullet+}$ to form BNA^{\bullet} . The BNA radical (BNA^{\bullet}) produced by deprotonation of the one-electron oxidized species of BNAH rapidly dimerizes to give BNA_2 . The formation of BNA_2 is also supported by LCMS studies (Supplementary Fig. 33). Moreover, the reduction power of BNA_2 ($E^0 = 0.26 \text{ V}$ vs SCE) is stronger than BNAH ($E^0 = 0.57 \text{ V}$ vs SCE). Hence the combined use of BNAH and TEA as a sacrificial electron donor can enhance the reduction ability of many folds (H_2 : $369.3 \mu\text{mol g}^{-1}$; CH_4 : $7274 \mu\text{mol g}^{-1}$) compared to the use of BNAH (H_2 : $265 \mu\text{mol g}^{-1}$; CO : $230 \mu\text{mol g}^{-1}$; CH_4 : $3567 \mu\text{mol g}^{-1}$) and TEA (H_2 : $291 \mu\text{mol g}^{-1}$; CO : $456 \mu\text{mol g}^{-1}$; CH_4 : $1200 \mu\text{mol g}^{-1}$) separately. Apart from these, the stability of the intermediate species can change the course of the electron transfer pathway to lead to different CO_2 reduced products, which we have already shown from DFT calculation (Fig. 6). Additionally, the bound CO species with the catalyst can play an important role for the progression towards CH_4 formation.

Supplementary Fig. 34 Oxidation and dimerization Processes of BNAH.

6. Further analysis and a critical comment from the authors is warranted on mass transport limitations during photocatalysis. With such a heavily reduced BET surface area (down to $357 \text{ m}^2\text{g}^{-1}$) for the loaded MOF samples, and substantial expected pore blocking from the D-A complex, how well can BNAH and TEA still diffuse to active sites? Particularly as a dual role is claimed. Is this the rate limiting step?

Response: Considering the catalyst design, we have seen a decrease in micropore volume after incorporating CT complex (PBA-MV), which is obvious from BET surface area measurement. As shown in supplementary figure 4, the micropore volume is obviously decreasing after post-synthetic modification. However, there is no significant decrease in mesopore volume. So it is logical to say in spite of mass transport limitation, the mesopores will assist in diffusing BNAH and TEA towards the catalytic site. Obviously, the diffusion rate would have been faster without CT complex grafting. However, practically, to achieve successful photocatalytic CO₂RR, grafting a photosensitizer is unavoidable. Hence, the choice of CT complex, as well as optimal loading of CT complex, is crucial. Additionally, we also prepared microporous MOF and checked the catalytic activity (See Supplementary information for details, Supplementary figure 35-40) and found less catalytic activity as compared to mesoporous MOF (CH₄:1815 μmol g⁻¹).

7. The impact of the photocurrent experiments is currently limited and should be further explained as basic experimental details are missing. What is the origin of these Faradaic photocurrents in absence of electron donor? Were they any potential applied during the experiment or only open circuit potential?

Response: We thank the reviewer for pointing out the crucial information needed to provide in the manuscript. The photocurrent experiment was performed in 0.2 M Na₂SO₄ solution, which was already mentioned in the supplementary information. For the photocurrent measurement, we applied a potential of 0.35 V vs Ag/AgCl while performing chronoamperometry measurement. During photocurrent measurement, after applying 0.35 V of applied potential, we observed 0.3 μA cm⁻² of steady current without any light irradiation. After applying light irradiation on the electrode surface, a significant enhancement of current (11.24 μA cm⁻²) was observed due to enhancing photoinduced carrier separation (*Angew. Chem. Int. Ed.* 2020, 59, 18763 – 18767; *J. Am. Chem. Soc.* 2019, 141, 2054-2060; *Nat Commun.* 2021, 12, 7313). In other words, we observed enhanced current due to the production of the charge-separated carrier in the presence of light. In contrast, due to fast charge recombination, **MOF-808-PBA** did not display significant current enhancement even in the presence of light.

8. This referee recommends considering refrain from using Tauc plot analysis for the functionalized material or performed further deconvolution to calculate the MOF-808-based band gap. As it is best used to described (bare) semiconductor behavior, it does not apply

well for modified MOF systems including molecular (non-band behaving) species such as PBA and MV. Please refer to 10.1021/acs.jpcclett.8b02892 for further detail.

Response: We thank the reviewer for the constructive suggestion. To avoid the erroneous band gap, we have modified the Tauc plot as suggested by the reviewer following the literature (*J. Phys. Chem. Lett.* 2018, 9, 6814–6817). Accordingly, we have modified fig. 2e and Supplementary Fig. 26b. After modifying, we found a slight enhancement of the band gap of **MOF-808-PBA-MV** and **MOF-808-PBA** due to the modification. We found the optical band gap of 1.93 eV and 2.75 eV for **MOF-808-PBA-MV** and **MOF-808-PBA**, respectively.

Fig. 2e Tauc plot of **MOF-808-PBA-MV** evaluating the optical bandgap.

Supplementary Fig. 26b Tauc plot of **MOF-808-PBA** evaluating the optical bandgap

9. Although not clearly stated, it seems that the excitation wavelength used for the experiment in Fig S49 is UV-based and unmatching the catalysis-effective light absorption in MOF-808-PBA-MV, which is around 590 nm according to Fig 1d and S27. Additionally, the emission profile is unlikely to stem from the relevant PBA-MV species as occurring at lower wavelength than its absorption. Thus, this referee finds the experiment shown in Fig S49 unable to support the conclusions.

Response: We understand the reviewer's concern. Hence we performed the PL experiment using $\lambda_{\text{ex}} = 590$ and collected the corresponding PL spectra. The PL plot does not show any emission band in this region (610-900 nm) (Fig. R9). However, our motive is to show the quenching process of PBA by BNAH (Oxidative or reductive). Considering the limitation (absence of emission at 610 -900 nm) we have indirectly tried to understand the quenching process, where we have considered the frontier molecular orbitals (FMOs) of PBA and MV separately. As discussed in the main manuscript, the charge transfer transition of PBA-MV CT complex was found to be happening from HOMO, which is on PBA to LUMO+1, which is on MV. Keeping that in mind, for separate PBA and MV band alignment, we have considered the LUMO of PBA unit and LUMO+1 of MV unit. As shown in Fig. R10 the LUMO+1 of MV (-2.266 eV) is low lying compared to the LUMO of PBA (-1.914 eV), suggesting feasible electron transfer from PBA to MV. From the perspective of this band

alignment picture, we can consider solely the emission of PBA unit for PL titration to prove the quenching process. Thus, upon photoexcitation, the hole will be generated on PBA unit only, irrespective of CT complex or individual PBA and MV units. As shown in Supplementary Fig. 54 the PL intensity was gradually decreased with the addition of BNAH, which firmly suggests the photoexcited hole is immediately quenched by the electron transfer from BNAH, and as a result, the radiative transition diminished. Hence, the absorption at 590 nm and the PL quenching of PBA are mutually exclusive from each other.

Fig. R9 a Photoluminescence spectra of **MOF-808-PBA-MV** ($\lambda_{ex} = 590$ nm).

Fig. R10 Electron transfer feasibility from PBA-MV to Zr-cluster using DFT.

In addition, the following technical points should be considered:

10. The authors should clearly state in the abstract and manuscript that the CH₄ selectivity is on an electronic basis and does not refer to the absolute quantities of produced gas, as this is misleading.

Response: We have calculated the selectivity based on the previous literature reports using the following equation (*J. Am. Chem. Soc.* 2017, **139**, 21, 7217-7223; *J. Am. Chem. Soc.* 2021, **143**, 16284–16292):

$$\text{Selectivity}_{\text{CH}_4} = \frac{8 * n_{\text{CH}_4}}{(8 * n_{\text{CH}_4}) + (2 * n_{\text{H}_2})} \times 100 \%$$

Where we have accounted the number of electrons required to convert CO₂ to CH₄ as well as the yield (mmol) of the products in the CO₂ photoreduction process.

$$\begin{aligned} \text{Selectivity}_{\text{CH}_4} &= \frac{8 * 7.3 \text{ mmol } g^{-1}}{(8 * 7.3 \text{ mmol } g^{-1}) + (2 * 0.3 \text{ mmol } g^{-1})} \times 100 \% \\ &= 99 \% \end{aligned}$$

11. The assignment of pore sizes to the enlarged HRTEM image in Figure 1b seems ambiguous based on the provided resolution and this reviewer suggests removing the orange assignments.

Response: We have removed the orange circle from the TEM images (Fig. 1b). However, sponginess denotes the mesopores nature of the sample, which is totally distinct from the microporous **MOF-808** (Fig R11).

Fig R11 HRTEM images of microporous **MOF-808**.

12. Why are not all PBA IR bands visible in the assembly? The authors show the inclusion of new bands at 2945, 1658 and 846 cm^{-1} (Suppl. Figure 9), but why do the bands at e.g., 1200-1300 cm^{-1} not appear?

Response: The bands at 1275 and 1205 cm^{-1} in the IR spectrum of PBA molecule correspond to C-O stretching of COOH group (Supplementary Fig. 9). However, after the exchange with formate ligand inside the MOF pore, the aforementioned bands were found to be slightly red-shifted to 1234 and 1182 cm^{-1} with a significant decrease in intensity. The fact can be reasonably ascribed to the confinement of PBA into the MOF pore, which will restrict the stretching modes significantly, resulting in a decrease in intensity in bands.

13. The structural integrity of the sample post-catalysis is shown (Suppl. Figure 42-45), however an investigation on how much PBA/MV is retained would be interesting. Is the D-A complex completely stable under photocatalytic conditions? Does some leaching occur? This could be investigated by dissolving a post-catalysis assembly for ^1H NMR experiments (as done for the initial characterization).

Response: We thank the reviewer for the suggestion. First, we performed NMR measurement in JEOL-600 spectrometer (600 MHz) to understand the precise ratio of PBA and MV^{2+} inside the MOF pore. Please see comment 1 of reviewer 1. We found the precise ratio of PBA and MV^{2+} is 4:3 in the MOF structure per formula unit. Next, as per the reviewer's

suggestion, we also performed the ^1H NMR experiment with digested **MOF-808-PBA-MV** after the photocatalysis experiment (Supplementary Fig. 46). We found the integration ratio remained unchanged with the pristine MOF, which suggested no leaching occurred during catalysis.

Supplementary Fig. 46 ^1H NMR spectra of digested **MOF-808-PBA-MV** in $\text{KOH}/\text{D}_2\text{O}/\text{DMSO-d}_6$ after catalysis.

14. Regarding the electrochemical experiments S17, could the authors comments on the redox peak at -0.38 V in **MOF-808-PBA-MV**. Why is it irreversible? Further cycling scan should be shown and it would be important to know to which scan the CV in S17 corresponds.

Response: We thank the reviewer for the suggestion. Methyl viologen undergo two consecutive one-electron reductions to the radical cation species ($\text{MV}^{\bullet+}$) at $E_{1/2}^1 = -0.38$ V vs Ag/AgCl and neutral species (MV^0) at $E_{1/2}^2 = -0.78$ V vs Ag/AgCl (Supplementary Fig. 20). According to the reviewer's suggestion, we looked at and rechecked the CV very carefully. We found that both the cathodic and anodic peaks correspond to the generation of $\text{MV}^{2+} \rightleftharpoons \text{MV}^{\bullet+}$ and $\text{MV}^{\bullet+} \rightleftharpoons \text{MV}$ are present (Supplementary Fig. 21). However, the peak intensity corresponding to $\text{MV}^{\bullet+} \rightarrow \text{MV}^{2+}$ was significantly decreased due to the formation of

stable $\text{PBA}^{\bullet+}$ - $\text{MV}^{\bullet+}$ CT interaction inside the MOF pore. To further clarify the decrease of the anodic peak corresponding to $\text{MV}^{\bullet+} \rightarrow \text{MV}^{2+}$ formation, three consecutive CV cycles were scanned, where we noticed after 1st segment, the peak current corresponding to $\text{MV}^{2+} \rightleftharpoons \text{MV}^{\bullet+}$ was significantly decreased due to the formation of stable $\text{PBA}^{\bullet+}$ - $\text{MV}^{\bullet+}$ CT complex inside the MOF pore. This observation firmly signifies that it is only possible if the generated $\text{MV}^{\bullet+}$ was consumed by $\text{PBA}^{\bullet+}$ to form CT complex. Additionally, to further clarify the CT complex role, we have prepared an analogous MV^{2+} (1-(carboxypentyl)-1'-methyl-4,4'-bipyridinium) incorporated MOF-808 named as **MOF-808-MV-COOH** (Scheme-R1). The cyclic voltammetric study of this compound showed both the reversible peak of $\text{MV}^{\bullet+} \rightleftharpoons \text{MV}^{2+}$ and $\text{MV} \rightleftharpoons \text{MV}^{\bullet+}$ are present in this compound due to the absence of CT complex (Fig.R5).

Supplementary Fig. 20 Cyclic voltammogram of **MOF-808-PBA-MV** recorded in MeCN (0.1 M TBAPF_6).

Scheme-R1 Synthetic scheme: synthesis of **MOF-808-MV-COOH** via post-synthetic linker exchange.

Fig. R5 Cyclic voltammogram of **MOF-808-MV-COOH** recorded in MeCN (0.1 M TBAPF₆).

15. Further experimental details are required in the Supporting Information for photocurrent and Nyquist impedance measurements. For example, the solvent media and applied potential for photocurrent. In their current form the experiments cannot be reproduced and the value for the photocatalytic mechanism discussion is questionable.

Response: We thank the reviewer for pointing out the crucial information needed to provide in the manuscript. For both the experiment, photocurrent and impedance measurements we performed in 0.2 M Na₂SO₄ solution, which was already mentioned in the supplementary information. For the photocurrent measurement, we applied a potential of 0.35 V vs Ag/AgCl while performing chronoamperometry measurement. During the impedance study, we applied a potential of -1.5 V vs Ag/AgCl.

16. Readability of the DRIFTS data could be improved by revising Figure 5 to be clearer (e.g., colors, region between 2800 and 2200 cm⁻¹ is irrelevant etc.).

Response: We thank the reviewer for the suggestion, and we have modified fig. 5 accordingly.

17. Figure 6b features several "desorptipn" typos and many unexplained acronyms can be found in the MS.

Response: We are extremely sorry for this mistake. We have corrected the typo error.

18. Clarification over the "TON of 28" should be given. What is the reference here? The Zr-clusters?

Response: The turn over number was calculated using the following formula:

$$TON = n_{CH_4} / n_{catalyst}$$

Where n_{CH₄} is yield (mol) of CH₄ during CO₂ photoreduction and n_{cat} is the amount of catalyst (mol) employed in the reaction.

Using 1 g of catalyst 7.3 mmol of CH₄ was produced.

Here Zr was accounted as catalytic center and the presence of Zr in **MOF-808-PBA-MV** was calculated from ICP-OES measurement where 35% of Zr is present in the catalyst. Hence the active catalytic center per g was found to be **2.6 * 10⁻⁴** mol.

$$\begin{aligned} TON &= 7.3 * 10^{-3} / 2.6 * 10^{-4} \\ &= 28.07 \sim 28 \end{aligned}$$

Reviewer #3 (Remarks to the Author):

In this manuscript, the authors confine D-A supramolecular complex into MOF-808 for photocatalytic CO₂ reduction. The activity and selectivity of CO₂-to-CH₄ are impressive. Notably, the photocatalyst exhibits an around 1 mmol/g CH₄ production rate under natural sunlight irradiation. Despite the exciting high photocatalytic results, the mechanism of CH₄ production is unclear. Overall, some key issues need to be addressed to support the authors' conclusion before reconsideration in Nature Communications.

Response: We thank the reviewer for appreciating the work. As per reviewer suggestions, we have addressed the concerns and revised the manuscript accordingly. The comments are very helpful for improving the quality and value of this article.

1. The authors claim the Zr-oxo cluster is the catalytic site for CH₄ production. However, almost all previous reports have shown that Zr-oxo clusters in MOFs, behaving photocatalytic sites, catalyze the reduction of CO₂ to produce HCOOH or CO --- 2e products. Unexpectedly, in this work, CH₄ can be generated in almost all control experimental conditions – this is difficult to understand. This is the most important point and should be well explained with related evidences.

Response: We understand the reviewer's concern regarding methane formation in the control experiment with **MOF-808** as well as **MOF-808-PBA**. Product and product selectivity can depend upon reaction environment such as sacrificial electron donor, solvent media (*Angew. Chem. Int. Ed.*, 2022, **61**, e202116094; *Inorg. Chem.*, 2015, **54**, 5096-5104; *Coord. Chem. Rev.*, 2018, 373, 333-356; *Nature Chemistry*, 2020, **12**, 180–185; *Energy Environ. Sci.*, 2021,**14**, 2429-2440). All our above-mentioned control experiments were performed with TEA and BNAH, and we found the combination of TEA and BNAH also provide some minimal product formation. Previous studies on Zr^{IV}-based MOFs showed that the Zr-oxo cluster in PCN222 (Zr^{IV}-porphyrin MOF) and PCN136 (Zr^{IV}-hexakis(4-carboxyphenyl)hexabenzocoronene MOF) act as an active catalytic site, where they found HCOOH as a CO₂ reduced product and used different sacrificial agent as well as solvent media (*J. Am. Chem. Soc.*, 2015, **137**, 13440-13443; *J. Am. Chem. Soc.*, 2019, **141**, 2054-2060).

In this work, the design strategies towards catalyst make the proximity between the catalytic center and light-harvesting unit, which enhances the charge mobility to catalytic center from

light-harvesting unit for CO₂ conversion to get a higher reduced product like CH₄. Moreover, close confinement inside the coordination nanospace will provide a high excited-state lifetime of the photogenerated electrons by decreasing the nonradiative electron-hole recombination pathway. Hence we achieved a catalytic activity of 7274 μmol g⁻¹ of CH₄ along with 369 μmol g⁻¹ of H₂ by using **MOF-808-PBA-MV**. we found **MOF-808-PBA** and **MOF-808** produced 180 and 102 μmol g⁻¹ of CH₄ and some amount of H₂ production. However, we did not notice any CO production in this process. Remarkably, **MOF-808-PBA-MV** showed 40-fold and 71-fold enhancements for CH₄ production as compared to **MOF-808-PBA** and **MOF-808**, respectively. This is due to less visible light absorptivity and unsteady electron transfer kinetics in **MOF-808-PBA** and **MOF-808**.

2. Page. 10, Line 257: In the presence of only TEA or BNAH, CO will be produced. Why coexistence of TEA and BNAH does not produce CO? The detailed explanations with evidences behind the results should be provided.

Response: We thank the reviewer for the suggestion. Triethylamine (TEA) has been extensively used as a sacrificial electron donor for photocatalytic CO₂ reduction. However, recent literature reports demonstrated with the use of BNAH as a sacrificial electron donor, the selectivity and efficiency of photocatalytic CO₂ reduction could be modulated (*Angew. Chem. Int. Ed.*, 2022, **61**, e202116094; *Inorg. Chem.*, 2015, **54**, 5096-5104; *Coord. Chem. Rev.*, 2018, 373, 333-356; *Nature Chemistry*, 2020, **12**, 180–185). This can be associated with the lower oxidation potential of BNAH ($E^{\circ}_{\text{OX}} = 0.57$ V vs. SCE) compared to TEA ($E^{\circ}_{\text{OX}} = 0.69$ V vs. SCE). Hence, BNAH can easily oxidize as compared to TEA which leads to a faster electron transfer to the catalyst during photocatalysis. According to our results, we got better activity by using BNAH alone as a sacrificial agent as compared to TEA. Now the question arises about the role of TEA with BNAH as an electron donor in the photocatalytic reduction reaction. The oxidation process of BNAH is shown in Supplementary Fig. 34. In the first step, one-electron oxidation product of BNAH was formed, which has a probability of taking back the electron from the catalyst (one-electron reduced species). To suppress this back electron transfer process, TEA is required, which deprotonates the BNAH⁺ to form BNA[•]. The BNA radical (BNA[•]) produced by deprotonation of the one-electron oxidized species of BNAH rapidly dimerizes to give BNA₂. The formation of BNA₂ is also supported by LCMS studies (Supplementary Fig.33). Moreover, the reduction power of BNA₂ ($E^0 = 0.26$ V vs SCE) is stronger than BNAH ($E^0 = 0.57$ V vs SCE). Hence the combined use of

BNAH and TEA as a sacrificial electron donor can enhance the reduction ability of many folds (H_2 : $369.3 \mu\text{mol g}^{-1}$; CH_4 : $7274 \mu\text{mol g}^{-1}$) compared to the use of BNAH (H_2 : $265 \mu\text{mol g}^{-1}$; CO : $230 \mu\text{mol g}^{-1}$; CH_4 : $3567 \mu\text{mol g}^{-1}$) and TEA (H_2 : $291 \mu\text{mol g}^{-1}$; CO : $456 \mu\text{mol g}^{-1}$; CH_4 : $1200 \mu\text{mol g}^{-1}$) separately. Apart from these, the stability of the intermediate species can change the course of the electron transfer pathway to lead to different CO_2 reduced products, which we have already shown from DFT calculation (Fig. 6). Additionally, the bound CO species with the catalyst can play an important role for the progression towards CH_4 formation.

Supplementary Fig. 34 Oxidation and dimerization Processes of BNAH.

3. Page. 6, Line 137: The authors think the PBA molecules will occupy the micropore of MOF-808. In my opinion, PBA may occupy mesopores more easily due to the steric hindrance. Can the authors give some possible explanations?

Response: We thank the reviewer for the query. Mesopores are generated from linker defect, which is randomly oriented throughout the framework and less in number. On the other hand, micropores are surrounded by the SBU related to Zr cluster. Notably, formates are oriented in the micropore. From the Brunauer-Emmett-Teller (BET) measurement, we observed a significant decrease in the uptake in the microporous region, whereas uptake in the mesoporous region remains almost unaltered (Supplementary Fig. 4). Consequently, we observed a significant decline in microporous volume as compared to mesoporous volume. For the pristine **MOF-808**, the micro and mesopore volume was evaluated to be 0.3 and $0.78 \text{ cm}^3 \text{ g}^{-1}$, respectively, with $V_{\text{micro}}/V_{\text{meso}}$ ratio of 0.39 . Here mesopore helps to diffuse the PBA molecules within their inner porous structure to get access for post-synthetic exchange with formate ligand. After incorporating PBA molecule via post-synthetic ligand exchange, we observed a significant decrease in micropore volume to $0.148 \text{ cm}^3 \text{ g}^{-1}$, whereas mesopore volume remained intact ($0.775 \text{ cm}^3 \text{ g}^{-1}$) with $V_{\text{micro}}/V_{\text{meso}}$ ratio of 0.191 . Hence, from the BET

measurement, it can be concluded mesopore helps to diffuse the molecule into the inner core of the MOF for post-synthetic modification.

4. Page. 6, Line 149: The authors ascribe the PL spectrum change of MOF-808-PBA to the close proximity of PBA molecules. Whether to lower the loading of PBA, the PL spectra will be similar to PBA monomer?

Response: We thank the reviewer for the suggestion. We have prepared **MOF-808-PBA** with a minimal concentration of PBA (0.001 M) and performed photoluminescence studies. As we discussed earlier, the photoluminescence spectra of **MOF-808-PBA** (0.1 M) showed a red-shifted, broad, structureless excimer emission band at 480 nm ($\lambda_{\text{ex}}= 370$ nm), which can be attributed to the close proximity of PBA molecules in the confined nanospace of MOF, hence enabling excimer formation (Fig. R12a). However, when we performed the photoluminescence study with **MOF-808-PBA** (0.001 M), we found significant excimer emissions along with some slightly enhanced monomeric emission (350-420 nm) present in the sample (Fig. R12b). This result suggested due to the close proximity of the PBA ligand to the MOF pocket, and it can easily form pyrene excimer with lower concentration also. In fact, from ^1H NMR study using the digested **MOF-808-PBA** (0.001 M), we found that only two formate ligands exchanged with PBA.

Fig. R12 a Photoluminescence spectra of **MOF-808-PBA** (0.1M). **b** Photoluminescence spectra of **MOF-808-PBA** (0.001 M).

5. Page. 6, Line 156: How to get the quantum yield of 23.88%?

Response: Here, we measured the fluorescence quantum yield (Φ_f) of **MOF-808-PBA**. The quantum yield (Φ) measures the efficiency of photon emission as defined by the ratio of the number of photons emitted to the number of photons absorbed. Here, confining the PBA molecule inside the nanospace will restrict the nonradiative pathways; thus, enhanced emission lifetime and quantum yield of the PBA molecule significantly compared to the free PBA molecule. The observed phenomenon is well documented in the previous literature, such as *Energy Environ. Sci*, 2021, **14**, 2429; *ACS Appl. Mater. Interfaces* 2017, **9**, 5699; *J. Am. Chem. Soc.* 2019, **141**, 7115.

6. The unit the of x-axis in Supplementary Fig. 3. Should be °C.

Response: We are extremely sorry for this mistake. We have corrected Supplementary Fig. 3.

Comments from Reviewer #1:

In the revised version, the authors significantly improved the quality and well responded the concerns of reviewers, thus I principally recommended the publication of this manuscript. But, there are still a few items not fully clear. For example, the radical anionic BTC.- was shown in DFT calculation, as part of MOF intermediate. Generally speaking, the radical anions of bigger pai systems such as pyrene.-, naphthalene.- were reported in literatures, but radical anionic BTC.- was rarely seen in literature. How do the authors think about this?

Comments from Reviewer #2:

The authors have partially addressed the concerns raised in the previous peer review round with new experiments. The clarification of dark vs daylight handling of the material and the origin of charge separation vs ground state (previous comment #1) is a welcome addition. Nevertheless, this reviewer is still very skeptical (similar to Reviewer #3's comments) about the precise nature of CH₄ formation, the implications of the control experiments and the discussions of the band/state and electron cascade. In addition, many of the experimental clarifications requested have not been added to the MS or the SI. While this referee wishes to emphasize that the performance reported and the original strategy are novel and noteworthy, the understanding and discussion of the system remain unclear with some key experiments being still missing. Thus, it is of this referee's opinion that the following major issues remain, and that this MS needs significantly revised/rephrased to be considered for publication in Nature Communications:

Please see the following comments as a few highlights of the aforementioned issue:

1. In response to our previous comment #2 on DRIFTS assignment and electron source, the authors have performed additional nice measurements with BNAH/TEA present. However, the key claim remains that "MOF-808-PBA-MV is also catalytically active in water without any sacrificial electron donor with a production of 460 $\mu\text{mol g}^{-1}$ of CH₄". This remains extremely surprising as this is an 8-electron reduction and water oxidation is challenging in these conditions. The authors' claim should be proven by detecting and quantifying the oxidation product (O₂, H₂O₂, OH radicals?) when no BNAH/TEA is present.

2. Our previous comment #3 questioned why BNAH/TEA by themselves can produce almost as much hydrogen as when the MOF is added, as well as traces of CH₄ (particularly surprising!). The authors state in their response that "By comparing the activity of both systems, we can tell a combination of BNAH and TEA alone is catalytically inactive. However, the minimal activity can be justified by the following facts: BNAH absorbs visible light from 400 to 450 nm". To this reviewer, this is not a full justification and particularly the CH₄ formation needs to be fully clarified to avoid experimental artefacts impeding the conclusions on the presented MOF material.

3. The authors have provided important control experiments with CO as a starting gas in the response to our previous comment #4 (Figures R7 and R8). This is a welcome addition and should be added to the SI with short comments in the manuscript. However, the requested ¹³CO labelling experiment was not performed, this could be added accordingly.

4. In the response to our comment #6 the authors assign BNAH/TEA diffusion to the available mesopores, allowing electrons to be delivered to the active sites. To support and prove this, this reviewer suggests (i) determining the incremental pore volume for pristine MOF-808 and loaded MOF-808 (the actual availability of mesopores is difficult to judge from supplementary Figure 4 alone). Additionally, (ii) the authors should comment on how MV/PBA appears to be highly preferentially anchored in the micropores, which still raises the question how these blocked micropores can be reached by BNAH/TEA during photocatalysis?

5. In response to comment #1, the dark experiment is a welcome and important addition. The authors should clearly confirm that all original investigations are done on the dark samples (“clean samples”) otherwise it is unclear whether these characterizations refer to the charge separated (radical) state or to the assembled complex. If it is not the case, then these characterizations must be performed and the clear dissociated description of the two states needs to be performed.

6. The questions raised in comment #7 have not been addressed and this referee would respectfully suggest considering further investigations/understanding.

7. Similarly, the comment #8 has not been satisfactorily addressed. Major issues on the Tauc plot remain as all the 3 plots are shown with a different eV scale disabling the clear observation and attribution of the different components. In addition, as in the Tauc plot case, the Mott Schottky analysis is not suitable for all MOFs especially functionalized one (please see: [10.1002/adma.201705512](https://doi.org/10.1002/adma.201705512), [10.1002/anie.202106342](https://doi.org/10.1002/anie.202106342)).

In other words, these analyses are likely suitable for the pristine MOF-808 but not for the functionalised systems. This led to related considerations/discussions sometimes conflicting in the MS, as in Fig2 and Fig R10. This reviewer would respectfully suggest the following considerations to build the discussion and description of the system. Here it is clear that as the MOF-808 remains in the final functionalized architecture and the original band structure is unchanged in the MOF808-PBA-MV. Thus the assembled system is likely more precisely reflected in a photosensitiser (PBA-MV)-semiconductor(MOF-808) way rather than a new material formation that would have new “bands”. This reviewer believes that overall, such a revision won’t change the main aspect or findings of this work, however, it will put to energetic consideration in the most adapted framework for the reader’s understanding.

Comments from Reviewer #3:

The authors have carefully addressed most of the concerns raised by this reviewer. However, this reviewer doesn't think the responses on why Zr-oxo cluster is the catalytic site for CH₄ production is convincing. I agree with that the sacrificial agent can change the selectivity and activity of CO₂ reduction, but according to the references provided by the authors, CO is the major product in the presence of BNAH as the sacrificial agent. As far as I know, the Zr-O clusters in MOFs usually tends to high selectivity of CO and HCOOH. In Table S6, the selectivity will shift to CH₄ upon addition of BNAH. Can all Zr-based MOFs convert CO₂ to CH₄ by simply using BNAH as a sacrificial agent? It will be very important to elucidate the mechanism why the Zr-O clusters can convert CO₂ to CH₄.

Thank you for sending us the reviewer reports on our manuscript titled " **Confining Charge-Transfer Complex in MOF for Modulating Electron Transfer Kinetics towards Photocatalytic Selective CO₂ Reduction in Water**" bearing manuscript id: **NCOMMS-22-08405-A**.

REVIEWER REPORT(S):

Comments from Reviewer #1:

In the revised version, the authors significantly improved the quality and well responded the concerns of reviewers, thus I principally recommended the publication of this manuscript. But, there are still a few items not fully clear. For example, the radical anionic BTC.- was shown in DFT calculation, as part of MOF intermediate. Generally speaking, the radical anions of bigger π systems such as pyrene.-, naphthalene.- were reported in literatures, but radical anionic BTC.- was rarely seen in literature. How do the authors think about this?

Response: We are happy to note that reviewer is satisfied with our revision and thankful to the reviewer for principally recommending for the publication of this manuscript.

We appreciate the reviewer for raising this question and we are sorry that we could not clarify this point in our earlier version. We agree with the reviewer that larger π system like pyrene naphthalene can stabilize extra electron by forming a stable radical anion. Our DFT calculation suggested that the photoexcited electron from MV²⁺ will reach to the Zr-oxo cluster through BTC linker. In photocatalysis, sometime excited state electron transfer happens via different species to the catalytic site and which is common in Cascade electron transfer process. The **f-[Zr^{IV}(BTC^{•-})(BTC)(H₂O)]⁻** state is only a transient one. The photoexcited electron from the photosensitizer will only transiently stay on the BTC and will immediately get transferred to the CO₂-substrate bound Zr to carry out the catalysis. For this type of transiently stable intermediate was realized by the spin density distribution plot, which shows the location of the extra negative charge, and such conversion from the spin density plot was documented in the literatures [ACS Catal., 2019, 9, 4867-4874.]. In our case, the location of the negative charge was found on the BTC ligand from the spin density distribution (Supplementary Table 11, Fig. R1).

Fig. R1. Spin Density Distribution of $f\text{-}[\text{Zr}^{\text{IV}}(\text{BTC}^-)(\text{BTC})(\text{H}_2\text{O})]^-$ obtained from DFT calculation.

We have added a sentence in the modified manuscript as *"The generation of transient state in $f\text{-}[\text{Zr}^{\text{IV}}(\text{BTC}^-)(\text{BTC})(\text{H}_2\text{O})]^-$ with BTC^- (2) is verified by the spin density distribution plot where the spin density of the extra electron is clearly seen on BTC ligand (Supplementary Table 11)."*

Further, to appease the reviewer's curiosity, we can discuss the comparative MO analysis among PBA-MV, SBU (secondary building unit) of Zr-MOF-808, BTC attached SBU and CO_2 -bound BTC attached SBU (Fig. R2). Frontier MO analysis revealed that the LUMO of SBU located on Zr is higher in energy (-1.668 eV) compared to the LUMO of PBA-MV (-2.286 eV) suggesting that the direct transfer of the photoexcited electron from PBA-MV to SBU unit is not possible. However, upon attaching BTC ligands with the SBU unit, the LUMO gets relocated on BTC from Zr [*Chem. Eur. J.*, 2021, 27, 4098-4107.], and it becomes low lying (-2.532 eV) compared to the LUMO of PBA-MV (-2.286 eV) which facilitates the electron transfer from PBA-MV to BTC attached SBU. Whereas after CO_2 -binding, LUMO becomes further low lying (-2.734 eV) and gets relocated from BTC to the CO_2 -bound Zr centre suggesting feasible transfer of photoexcited electron to CO_2 -bound Zr centre suggesting feasible transfer of photoexcited electron to CO_2 -bound Zr centre. This accumulation of the photoexcited electron on the Zr centre is necessary to carry out the catalysis. In fact from EPR, the generation of Zr^{III} was proven. Hence, it can be said that the photoexcited electron is getting relayed to Zr centre through BTC ligand. It is noteworthy to mention here, in case of Ce-UiO-66 MOF, the direct electron transfer to Ce catalytic centre is

possible because the LUMO is located on Ce only and it is sufficiently low lying in energy [*J. Am. Chem. Soc.*, 2018, 140, 7904-7912].

Fig. R2. Frontier MO alignment of PBA-MV, SBU, SBU with BTC and CO₂-bound on BTC attached SBU.

Comments from Reviewer #2:

The authors have partially addressed the concerns raised in the previous peer review round with new experiments. The clarification of dark vs. daylight handling of the material and the origin of charge separation vs. ground state (previous comment #1) is a welcome addition.

Nevertheless, this reviewer is still very skeptical (similar to Reviewer #3's comments) about the precise nature of CH₄ formation, the implications of the control experiments and the discussions of the band/state and electron cascade. In addition, many of the experimental clarifications requested have not been added to the MS or the SI. While this referee wishes to emphasize that the performance reported and the original strategy are novel and noteworthy, the understanding and discussion of the system remain unclear with some key experiments being still missing. Thus, it is of this referee's opinion that the following major issues remain, and that this MS needs significantly revised/rephrased to be considered for publication in Nature Communications.

Response: We are thankful to the reviewer for mentioning that our synthetic approach is original and noteworthy. We also thank the reviewer for the suggestion, and now we have included a few key experimental findings in the main manuscript and SI. In fact, we have already included crucial observations as suggested by the reviewer in the highlighted manuscript (highlighted in yellow) of the previous response.

We are sorry that the some of the experimental details was not included in the MS as we thought to clarify only the reviewer comments. Now based on the reviewer comments we have added crucial observations in the revised manuscript.

Please see the following comments as a few highlights of the aforementioned issue:

1. In response to our previous comment #2 on DRIFTS assignment and electron source, the authors have performed additional nice measurements with BNAH/TEA present. However, the key claim remains that "MOF-808-PBA-MV is also catalytically active in water without any sacrificial electron donor with a production of 460 μmol g⁻¹ of CH₄". This remains extremely surprising as this is an 8-electron reduction and water oxidation is challenging in these conditions. The authors' claim should be proven by detecting and quantifying the oxidation product (O₂, H₂O₂, OH radicals?) when no BNAH/TEA is present.

Response: We understand the reviewer's concern. However, the comments "This remains extremely surprising as this is an 8-electron reduction and water oxidation is challenging in these conditions." is very surprising. Here, we are attaching a few references showing CO₂ reduction in pure water medium and got methane, acetic acid as a reduced product. Even a few references showed ethanol production in a pure water medium, a 12 e⁻ reduction product.

(1) *Nat. Energy*, 2019, **4**, 690–699

(2) *Energy Environ. Sci.*, 2022, **15**, 1967-1976

(3) *ACS Energy Lett.* 2018, 3, 11, 2656–2662.

(4) *J. Am. Chem. Soc.* 2018, 140, 6474–6482.

(5) *J. Am. Chem. Soc.* 142, 75-79 (2020)

As per the reviewer's suggestion, we have performed photocatalytic CO₂ reduction in pure water medium and analyzed the product using gas chromatography-mass spectrometry (SHIMAZU GC-2010 PLUS). In this reaction, only oxygen was detected as an oxidation product due to water oxidation. Additionally, we have quantified O₂ at different time intervals throughout the course of the reaction. The maximum production yield of O₂ was calculated to be 866 μmol g⁻¹ in 16 h (Fig R3).

Fig. R3: (a) O₂ production (oxidation product from water oxidation) over **MOF-808-PBA-MV** as a function of time of irradiation in water. (b) Gas chromatogram of O₂ production over **MOF-808-PBA-MV** in water. (c) Mass spectrum of produced O₂ under visible light over **MOF-808-PBA-MV**.

2. Our previous comment #3 questioned why BNAH/TEA by themselves can produce almost as much hydrogen as when the MOF is added, as well as traces of CH₄ (particularly surprising!). The authors state in their response that "By comparing the activity of both systems, we can tell a combination of BNAH and TEA alone is catalytically inactive. However, the minimal activity can be justified by the following facts: BNAH absorbs visible

light from 400 to 450 nm". To this reviewer, this is not a full justification and particularly the CH₄ formation needs to be fully clarified to avoid experimental artefacts impeding the conclusions on the presented MOF material.

Response: In case of only BNAH and TEA in the absence of MOF catalyst, BNAH will act as the self-sensitized (Since it has visible light absorption) photocatalyst by binding CO₂ through its amide group. At that time, TEA will act as the sacrificial electron donor for CO₂ reduction. Moreover, TEA will not only provide the reduction electron, it will also supply the proton for CH₄ production, which can be seen from the following literatures: [(1) *J. Chem. Soc. Faraday Trans.*, 1993, 89, 1857-1860.; (2) *Angew. Chem. Int. Ed.*, 2022, 61, e202116094.; (3) *C. R. Chimie*, 2017, 20, 283e295]. The generation of Et₂N⁺=CHCH₃ (m/z = 100) was confirmed by the LCMS study, suggesting TEA will act as the sacrificial electron and proton donor when BNAH acts as the catalyst (Fig R4).

Fig. R4 Liquid chromatography-mass spectrometry (LC-MS) of the liquid phase from the reaction system of BNAH+TEA after visible light irradiation

Moreover, we want to clarify the above concern by the following additional facts:

- 1) The activity of MOF-808-PBA-MV was 182 times higher towards CH₄ formation compared to a combination of BNAH and TEA. Notably, 182 times higher activity can't consider as an artefact.
- 2) The selectivity of CH₄ formation is 31% by using BNAH and TEA as a mixture, whereas selectivity increased to 99% when we used catalyst along with sacrificial e⁻ donor.
- 3) The isotopic ¹³CO₂ study over MOF-808-PBA-MV was performed to exclude any experimental artefact.
- 4) Moreover, we have performed few control experiments which suggested the role of CT complex in the confined space as well as the role of Zr-oxo cluster for catalytic CO₂ reduction (Supplementary Fig. 32 and Supplementary Table 6). PBA grafted **MOF-808-PBA** exhibited 180 μmol g⁻¹ of CH₄ along with 76 μmol g⁻¹ of H₂ in 16 h, and the production amount did not increase significantly with irradiation time, demonstrating the unsteady electron relay process from PBA to the catalytic centre, reflecting the sluggish kinetics towards CH₄ production (Rate: 12 μmolg⁻¹ h⁻¹). Afterwards, by directly mixing PBA and MV in a homogeneous reaction condition,

120 $\mu\text{mol g}^{-1}$ of CH_4 and 50 $\mu\text{mol g}^{-1}$ of H_2 were produced, also demonstrating the important role of electron flow in a confined nanospace inside the MOF.

These precise observations suggest **MOF-808-PBA-MV** produces CH_4 from dissolved CO_2 , whereas in the absence of catalyst, CH_4 , CO and H_2 are produced from BNAH and TEA. Importantly, in the many reported literature, they have not provided the details of the control study. Here we have provided these experimental results of the control experiments to exclude the artefact of the catalysis. Also, we have tried to analyze the effect of every individual component in the photocatalytic process.

3. The authors have provided important control experiments with CO as a starting gas in the response to our previous comment #4 (Figures R7 and R8). This is a welcome addition and should be added to the SI with short comments in the manuscript. However, the requested ^{13}C labelling experiment was not performed, this could be added accordingly.

Response: In previous revision the reviewer mentioned, "In addition to the above points, the authors should consider performing photocatalysis control experiments and further DRIFTS analysis using CO (ideally labelled 13-C carbon monoxide) instead of CO_2 to further support their mechanism and the origin of CH_4 ."

With due respect, we really want to know how ^{13}C will provide insight into the mechanism of CH_4 formation and the origin of CH_4 , where the main focus of our work is CO_2 reduction and not CO reduction.

However, as suggested by the reviewer, we have already performed the reaction using CO and also performed DRIFT by using CO . Now we have included that part in the manuscript and supplementary information. However, we didn't find any additional advantage using CO as a starting gas to prove the CH_4 formation mechanism. We also point out that ^{13}C is not easily available, and it is highly expensive. Apart from that, now, we have performed the DRIFT study with $^{13}\text{CO}_2$, which clearly depicted the possible reaction intermediates in this process (Fig. R5). Notably, we have performed the DRIFT experiment with $^{13}\text{CO}_2$ for the 1st time to prove the CO_2RR mechanism. Peaks appearing after photo-irradiation at 1339 cm^{-1} correspond to COOH^* intermediate, which is a crucial intermediate during the photochemical conversion of CO_2 to CH_4 [(1) *J. Am. Chem. Soc.*, 2021, 143, 16284–16292; (2) *Nat. Energy*, 2019, 4, 690–699). Notably, a strong band appeared at 1720 cm^{-1} , which could be attributed

to the bending vibration of C=O (*ACS Catal.*, 2018, **8**, 9280-9286). Importantly, the absorption band at 980, 1095, 1595, and 1101 cm^{-1} belongs to C-OH*, CHO*, CH₂O* and CH₃O* species, respectively, which are pivotal intermediate for CH₄ formation [(1) *J. Am. Chem. Soc.* 2018, **140**, 6474-6482; (2) *Nat. Energy*, 2019, **4**, 690–699; (3) *J. Mater. Chem. A*, 2017,**5**, 5020-5029; (4) *Nat Catal*, 2020,**3**,1034–1043]. The band that appeared at 1405 corresponds to the deformation vibrations of CH₃* and CH₄ group [(1) *ACS Catal.* 2018, **8**, 9280-9286; (2) *Anal. Methods*, 2016, **8**, 756–762]. Besides, the band at 2861 cm^{-1} corresponds to C-H stretching frequency of CH₃* group, and 2922 cm^{-1} correspond to C-H stretching frequency of CH₂* group (*ACS Catal.* 2018, **8**, 9280-9286). We observed all bands corresponding to the crucial intermediate for CH₄ formation were red-shifted compared to when we purged the system with ¹²CO₂, which is due to the isotopic effect (*Nat Catal.*, 2020, **3**, 1034–1043).

So, in our opinion, the study using ¹³CO is irrelevant in the context of photocatalytic CH₄ formation based on MOF-808-PBA-MV catalyst and will not change the overall conclusion of the manuscript. We have provided enough supporting experimental evidence to support the CH₄ formation, which are generally followed by the photocatalysis community, and the ¹³CO experiment will not provide any additional information to understand the mentioned reaction mechanism. Moreover, we could not find any literature regarding the DRIFT experiment using ¹³CO to understand the reaction mechanism of CO₂RR. A list of references (including *Nature Communications*) are mentioned below to justify our argument related to CO₂ reduction mechanism via *in-situ* DRIFTS.

(1) *Nat. Commun.*, 2021, **12**, 7313. (2) *Nat. Energy*, 2019, **4**, 690–699, (3) *Nat. Commun.*, 2020, **11**, 1149; (4) *Angew. Chem. Int. Ed.*, 2022, **61**, e2021160; (4) *ACS Catal.* 2018, **8**, 9280-9286; (5) *Energy Environ. Sci.*, 2018,**11**, 2382-2389; (6) *J. Am. Chem. Soc.* 2018, **140**, 6474-6482; (7) *Energy Environ. Sci.*, 2022,**15**, 1967-1976; (7) *Nano Lett.* 2021, **21**, 2324-2331; (8) *J. Mater. Chem. A*, 2017, **5**, 5020-5029; (9) *J. Catal.*, 2017, **352**, 532–541; (10) *J. Am. Chem. Soc.* 2021, **143**, **39**, 16284–16292; (11) *Energy Environ. Sci.*, 2021, **14**, 2429–2440; (12) *Sci Rep.*, 2020, **10**, 2128 (13) *Catal. Sci. Technol.*, 2018, **8**, 5129–5132; (14) *Chem. Commun.*, 2022, **58**, 6638–6641; (15) *ACS Appl. Energy Mater.* 2018, **1**, 6781–6789.

Fig. R5 Time-dependent *in situ* FT-IR spectra of photocatalytic $^{13}\text{CO}_2\text{RR}$ in a mixture of $^{13}\text{CO}_2$ and H_2O vapour, TEA and BNAH over **MOF-808-PBA-MV** in the dark and under visible light irradiation. Irradiation times 10, 20, 30, 40, 50, 60, 70, 80 min, respectively.

4. In the response to our comment #6 the authors assign BNAH/TEA diffusion to the available mesopores, allowing electrons to be delivered to the active sites. To support and prove this, this reviewer suggests (i) determining the incremental pore volume for pristine MOF-808 and loaded MOF-808 (the actual availability of mesopores is difficult to judge from supplementary Figure 4 alone). Additionally, (ii) the authors should comment on how MV/PBA appears to be highly preferentially anchored in the micropores, which still raises the question how these blocked micropores can be reached by BNAH/TEA during photocatalysis?

Response: In response to comment (i) determining the incremental pore volume for pristine MOF-808 and loaded MOF-808 (the actual availability of mesopores is difficult to judge from supplementary Figure 4 alone), we already mentioned the meso and microporous volume of pristine MOF and the post-synthetically modified MOF in our previous response (comment no 3, reviewer 3) as well as in the main manuscript. As the CT Complex has formed in between PBA and MV the aromatic rings of PBA and MV are cofacial (face to face stacking or H stacking) [(1) *J. Phys. Chem. B* 2021, 125, 8539–8549; (2) *Chem. Commun.*, 2010, 46, 5464–5466]. Without the close proximity inside the MOF pore, CT

complex will not form as supported by the EPR and TA spectroscopic studies (Fig. 4b,4c). Further, in control study in a homogenous medium such CT complex was not formed as shown in TA study (Supplementary Fig. 53). Finally, after incorporating PBA and MV inside the MOF pore, **MOF-808-PBA-MV** showed an uptake of $76 \text{ cm}^3\text{g}^{-1}$ and possessed a BET surface area of $357 \text{ m}^2 \text{ g}^{-1}$ with $V_{\text{micro}}/V_{\text{meso}}$ ratio of 0.125. The pore size distribution suggested a presence of micropore at 1.6 nm with a decrease micropore volume of $0.09 \text{ cm}^3 \text{ g}^{-1}$ (Fig. R6). Furthermore, the micropore is not completely blocked and BNAH/TEA can diffuse for electrons and protons transfer to the catalytic site. Hence the reviewer might overlook this part. However, BNAH/TEA not necessarily diffuse into the pore of the MOF for electron and proton transfer during catalysis. CO_2 reduction is a complex process, and electrons and protons can transfer into the surface of the nanoscale MOF containing a high active site for reduction. Even there are many literatures where they have used sacrificial electron donors in post-synthetically modified MOF with reduced pore size [(1) *J. Am. Chem. Soc.*, 2019, 141, 12219-12223; (2) *J. Am. Chem. Soc.*, 2017, 139, 356–362; (3) *J. Am. Chem. Soc.* 2018, 140, 16, 5326–5329;]. Also, there are many non-porous semiconductor or photocatalytic systems where such sacrificial electron donors are used. So, it is not necessary always that such sacrificial electron donors need to be in close contact with the catalytic center.

Fig. R6 a, N₂ adsorption isotherm of **MOF-808**, **MOF-808-PBA** and **MOF-808-PBA-MV** at 77 K. **b**, Pore size distribution for **MOF-808**, **MOF-808-PBA**, and **MOF-808-PBA-MV**.

5. In response to comment #1, the dark experiment is a welcome and important addition. The authors should clearly confirm that all original investigations are done on the dark samples ("clean samples") otherwise it is unclear whether these characterizations refer to the charge

separated (radical) state or to the assembled complex. If it is not the case, then these characterizations must be performed and the clear dissociated description of the two states needs to be performed.

Response: As suggested by the reviewer, we prepared the catalyst (MOF-808-PBA-MV_d) under dark conditions to prove the daylight effect to form the CT complex. The method of preparation of the dark sample is the same as we mentioned in the manuscript, except, in this case, we didn't expose the sample in the light. As soon as the sample is exposed under daylight, it forms a partially charge-separated state as daylight act as a stimulus, observed from the EPR spectra (Fig. R7, R8). However, such charge separation is not 100% as it is clear from the visible light (LED) exposed EPR spectra. As with light irradiation time, the intensity of MV^{•+} peak and Zr³⁺ peak increases. Hence we think the reviewer misunderstood our point. So it is unrelated to perform catalysis using MOF-808-PBA-MV_d as the whole catalysis process is under photo irradiation conditions.

Fig. R7 EPR spectra of MOF-808-PBA-MV_d under dark condition.

Fig. R8 EPR spectra of MOF-808-PBA-MV under daylight and irradiated with the LED light source.

6. The questions raised in comment #7 have not been addressed and this referee would respectfully suggest considering further investigations/understanding.

Response: In the previous revision, the reviewer asked for basic experimental details along with two more questions. What is the origin of these Faradaic photocurrents in the absence of electron donor? Were they any potential applied during the experiment or only open circuit potential?

We have already mentioned in our previous response that a potential of 0.35 V vs. Ag/AgCl was applied while performing chronoamperometry measurement.

We measure the photocurrent response to understand the photoactive behaviour of the catalyst in the presence of light. Transient photocurrent analysis interprets the dynamic response of a photoelectrode. In a typical experiment, the current response is collected as an incident light source is periodically switched on and off. Current spikes, which rapidly decay to a steady-state value termed transients, often occur directly after each switch. Subsequently, the generated electrons coming out towards the surface by light irradiation were measured in terms of current. The photocurrent activity also depends upon the light intensity. Thus, the

power of the xenon lamp has been maintained constant for all samples. This experiment usually performs to understand the electron transfer process in presence of light [see some of the references here (1) *J. Am. Chem. Soc.* 2019, 141, 2054-2060; (2) *J. Am. Chem. Soc.* 2021, 143, 16284-16292; (3) *Chem. Sci.*, 2018, 9, 8890-8894; (4) *J. Mater. Chem. A*, 2021, 9, 5780; (5) *Adv.Sci.* 2018, 5, 1700844; (6) *Nat Commun*, 2021, **12**, 7313; (7) *ACS Catal.* 2022, 12, 687-697; (8) *Adv.Sci.*,2018, 5, 1700844; (9) *J. Am. Chem. Soc.* 2017, 139, 4123-4129; (10) *Nat. Commun*, 2021, **12**, 2682; (11) *Nanoscale*, 2020, 12, 9533-9540]. Hence in the absence of electron donor, the photocurrent activity comes from light irradiation.

As per our understanding, for the 2nd query reviewer ask to perform photocurrent in presence of donor molecules. We thank the reviewer for suggesting this experiment. Hence, we performed photocurrent measurement in presence of the donor molecule and found the current density was 13.26 $\mu\text{A}/\text{cm}^2$ (Fig. R9). The observed photocurrent was slightly higher compared to what we observed previously without sacrificial electron donor 11.24 $\mu\text{A}/\text{cm}^2$, which is due to the hole quenching.

Fig. R9 Transient photocurrent responses of **MOF-808-PBA-MV** along with sacrificial electron donor in 0.2 M Na_2SO_4 aqueous solution under visible-light irradiation.

7. Similarly, the comment #8 has not been satisfactorily addressed. Major issues on the Tauc plot remain as all the 3 plots are shown with a different eV scale disabling the clear observation and attribution of the different components. In addition, as in the Tauc plot case,

the Mott Schottky analysis is not suitable for all MOFs especially functionalized one (please see: 10.1002/adma.201705512, 10.1002/anie.202106342).

In other words, these analyses are likely suitable for the pristine MOF-808 but not for the functionalized systems. This led to related considerations/discussions sometimes conflicting in the MS, as in Fig2 and Fig R10. This reviewer would respectfully suggest the following considerations to build the discussion and description of the system. Here it is clear that as the MOF-808 remains in the final functionalized architecture and the original band structure is unchanged in the MOF808-PBA-MV. Thus the assembled system is likely more precisely reflected in a photosensitizer (PBA-MV)-semiconductor (MOF-808) way rather than a new material formation that would have new "bands". This reviewer believes that overall, such a revision won't change the main aspect or findings of this work, however, it will put to energetic consideration in the most adapted framework for the reader's understanding.

Response: Thank you for this question. We are probably not able to convey our message clearly. With due respect, let me 1st build the story. In this work, we have prepared MOF-808 with the formula of $[\text{Zr}_6\text{O}_5(\text{OH})_3(\text{BTC})_2(\text{HCOO})_5(\text{H}_2\text{O})_2]$ (BTC: BTC = 1,3,5-benzenetricarboxylate). Next, we replaced the formate ligand with 1-pyrene butyric acid and we obtained a new MOF with the molecular formula $[\text{Zr}_6\text{O}_5(\text{OH})_3(\text{BTC})_2(\text{HCOO})_1(\text{C}_{20}\text{H}_{15}\text{O}_2)(\text{H}_2\text{O})_2]$. This MOF optoelectronic property is different compared to pristine MOF-808. Next, a donor molecule MV^{2+} was introduced inside the MOF pore to form a donor-acceptor compound in MOF pocket. All three components are different, representing three new systems with different light harvesting and optoelectronic properties. Hence in all three cases, the bandgap is also different. In the previous response, as suggested by the reviewer, all the Tauc plots were plotted using the baseline method as reported in *J. Phys. Chem. Lett.* 2018, 9, 6814–6817, whereas a linear fit was used as an abscissa applied for the slope below the fundamental absorption. To clarify the doubt of the reviewer, we have merged the Tauc plot in the same diagram to show the bandgap together (Fig R10). It can be seen that after modifying the MOF pore with PBA and also with PBA-MV complex, the optoelectronic properties of all three compounds significantly change. As a result of that, the bandgap of MOF-808 → MOF-808-PBA → MOF-808-PBA-MV changed from 4.06 to 2.75 to 1.93 eV (corresponds to the CT band). So, it is clear that modification with PBA and MV significantly affects the decrease in the bandgap. As we are dealing with visible light-driven catalysis, the decrease of bandgap from the UV region towards the visible region is crucial, which we achieve via engineering the MOF pore (MOF-808). We also

found several literature reports regarding the bandgap calculation from the Tauc plot using the functionalized MOF [(1) *J. Am. Chem. Soc.* 2019, 141, 2054-2060; (2) *Energy Environ. Sci.*, 2021,14, 2429-2440; (3) *J. Am. Chem. Soc.* 2020, 142, 12515-12523; (4) *Catal. Lett.*, 2020, 150, 2648–2659; (5) *J. Am. Chem. Soc.* 2015, 137, 13440-13443; (6) *J. Phys. Chem. C*,2022, 126, 30, 12348–12360].

Fig R10 Tauc plot-optical bandgap energy calculation for **MOF-808**, **MOF-808-PBA**, **MOF-808-PBA-MV**.

The same explanation is also valid for the M-S plot. After post-synthetic modification, we are preparing a new system with different molecular formulas and different properties. Hence we strongly disagree with the reviewer's comment on M-S measurement, as the flat-band potential of n-type MOFs is determined from the Mott–Schottky plot [(1) *J. Am. Chem. Soc.* 2019, 141, 2054-2060; (2) *J. Am. Chem. Soc.* 2015, 137, 13440-13443; (3) *J Mater Sci* ,2018,53,12016–12029; (4) *Chem. Sci.*, 2015, 6, 3926–3930; (5) *Energy Environ. Sci.*, 2021,14, 2429-2440; (6) *Nanoscale*, 2020, 12, 9533–9540] and is approximately equal to the intercept on the x-axis of the tangent to the inflection point of the C^2 –E curve.

Comments from Reviewer #3:

The authors have carefully addressed most of the concerns raised by this reviewer. However, this reviewer doesn't think the responses on why Zr-oxo cluster is the catalytic site for CH₄ production is convincing. I agree with that the sacrificial agent can change the selectivity and activity of CO₂ reduction, but according to the references provided by the authors, CO is the major product in the presence of BNAH as the sacrificial agent. As far as I know, the Zr-O clusters in MOFs usually tends to high selectivity of CO and HCOOH. In Table S6, the selectivity will shift to CH₄ upon addition of BNAH. Can all Zr-based MOFs convert CO₂ to CH₄ by simply using BNAH as a sacrificial agent? It will be very important to elucidate the mechanism why the Zr-O clusters can convert CO₂ to CH₄.

Response: We are thankful to the reviewer as the reviewer thinks that we have carefully addressed most of the concerns.

Further, thanks to the reviewer for asking this important question. Probably we have not clarified this aspect. The catalytic activity of a system depends upon many factors:

- 1) Visible light absorption.
- 2) Bandgap and HOMO LUMO position of the catalyst.
- 3) The lifetime of the excited state electron and e⁻ transfer kinetics.
- 4) The solvent media.
- 5) Available CO₂ binding site in the catalyst.
- 6) The role of Sacrificial electron donor.
- 7) Presence of the Co-catalyst
- 8) Competitive binding site

In this work, we describe the establishment of a new MOF material by introducing the donor-acceptor system PBA-MV in the MOF-808 (Zr), where we find visible light absorption: effect of the donor-acceptor entity. The push-pull effect, electron-hole separation exciton binding energy and confinement effect has a significant role in product formation (Product selectivity as well as product yield) [*J. Am. Chem. Soc.* 2021, 143, 16284–16292]. In many cases, electron recombination is very high. Hence product selectivity is limited to 2 e⁻ reduced products. We engineered the kinetics of the electron transfer, which we supported by the transient absorption experiment, and that is the beauty of our work. Hence we can't generalize this system with other Zr-MOF. In our understanding, all the Zr-based MOFs will not form CH₄ in the presence of BNAH. e.g. recently, Yu *et al.* reported CH₃COOH (8 e⁻ reduced product) product from CO₂ photoreduction using a modified Zr-MOF (UiO-66)

[*Angew. Chem. Int. Ed.*, 2021,60, 24849 –24853]. Sonowal *et al.* reported the formation of CH₃OH (6 e⁻ reduced product) by using a Zr(IV)-based MOF composite with g-C₃N₄ quantum dots [*J. CO₂ Util.* 2022, 57, 101905]. Importantly, for **MOF-808-PBA-MV** also showed, different catalytic activities in the presence of BNAH and TEA solely (Supplementary Table. 6). Hence it's justified to say that by engineering, we developed a new material, which alters its light harvesting as well as electronic properties, and makes charge transfer kinetics faster. Hence it's able to produce 8 e⁻ transfer product CH₄ with >99% selectivity in the presence of BNAH and TEA.

We are thankful to all the reviewers for their appreciation, critical comments and suggestions which has helped to improve the overall quality of the manuscript.

REVIEWER COMMENTS

Reviewer #1 (Remarks to the Author):

I am convinced that the revised version of this manuscript and the response to reviewers that the authors have provided numerous interesting observations and a roadmap to expanding the applicability of photo-catalytic carbon dioxide reduction. I recommend that this manuscript should be accepted in the Journal.

Reviewer #2 (Remarks to the Author):

In this revised version the authors have provided new experiments and argumentation to address previous concerns. This reviewer appreciates the effort and detailed response to the role of BNAH/TEA, the literature overview on DRIFTS bands, and the dark EPR sample. Overall, the manuscript has been significantly improved but a key conceptual aspect and a technical point remain.

If the authors can satisfactorily address these, the manuscript can be considered for publication in Nature Communications:

1. Technical: The response to our previous comment #1 on oxidation product detection is very welcome with the full detection of oxygen. These plots should be added to the SI and crucially, a plot is required which shows the CO₂ reduction product evolution and the oxygen evolution side-by-side. Here the number of transferred electrons should match to prove the connection between CH₄ and O₂ evolution.
2. Concept: Respectfully, the authors' response to the previous comment #7 misses the point. This reviewer is not criticizing a technical aspect but rather the misleading conceptual categorization of the samples.

While the Tauc analysis presented in the revised version Fig. R10 with the same eV scale is technically correct in terms of how the eV values were determined and that other systems in literature use this method (as the authors also stated), this reviewer regards this analysis method as generally not suitable for molecularly functionalized MOFs. While the pristine MOF can be considered as a semiconductor (where Tauc and Mott Schottky analyses are valid) the included molecules PBA and MV themselves certainly cannot. This is why the

previous comment #7 suggested the authors treat their functionalized systems as a sensitized semiconductor (see extensive literature on dye-sensitized semiconductors/solar cells etc.). Here PBA-MV perform as molecular sensitizers with discrete orbitals while the MOF can be treated as a semiconductor with band structure. Accordingly, for the MOF-PBA and MOF-PBA-MV samples shown in Fig. R10 the Tauc and Mott-Schottky analysis methods are likely not a suitable to accurately describe the optoelectronic behaviour of the samples (see the corresponding discussion in previous literature reviews 10.1002/adma.201705512, 10.1002/anie.202106342).

These aspects should be clarified, stated and the manuscript rephrased in all places necessary.

Reviewer #3 (Remarks to the Author):

The authors have tried to address reviewers' comments. However, according to authors' new response to all reviewers, this reviewer is afraid of the reliability of the obtained results.

1. The response 1 to Reviewer #2 -- Although the authors supply some references about CO₂ reduction in pure water to 8/12-electrons products, the most examples of CO₂ to CH₄ were performed in water vapor (even CO₂ hydrogenation).

2. Fig. R3. -- In Supplementary Table 6, MOF-808-PBA-MV in pure water can produce H₂, CH₄, CO and O₂. According to the analytic method in text (line 530-533), H₂, O₂, N₂, CO, CH₄, even CO₂ should be observable in the spectrum. However, only O₂ in Fig. R3b, which is more likely to be a signal of air.

3. Fig. R7. -- EPR signal comes from single electron. It is very strange that there is no any signal in Fig. R7. Actually, in my opinion, even defects, absorbed O₂, residual solvent/ligand will produce a EPR signal (at least free electron signal). In particular, the authors highlight many defects in this photocatalysts.

4. Response to Reviewer #2 -- I am not satisfied with the answer to this question ----the authors' response does not give any real support. In addition, I don't agree that "We engineered the kinetics of the electron transfer, which we supported by the transient absorption experiment, and that is the beauty of our work." The photocurrent intensity is about 10 $\mu\text{A}/\text{cm}^2$ that is a normal value in MOF photocatalysis. Overall, I have reservations on the acceptance of this manuscript.

Thank you for sending us the reviewer reports on our manuscript titled " **Confining Charge-Transfer Complex in MOF for Modulating Electron Transfer Kinetics towards Photocatalytic Selective CO₂ Reduction in Water**" bearing manuscript id: **NCOMMS-22-08405C**.

Reviewer's response (NCOMMS-22-08405C)

Thank you for sending us the reviewer reports on our manuscript titled " **Confining Charge-Transfer Complex in MOF for Modulating Electron Transfer Kinetics towards Photocatalytic Selective CO₂ Reduction in Water**" bearing manuscript id: **NCOMMS-22-08405C**.

REVIEWER REPORT(S):

Reviewer #1 (Remarks to the Author):

I am convinced that the revised version of this manuscript and the response to reviewers that the authors have provided numerous interesting observations and a roadmap to expanding the applicability of photocatalytic carbon dioxide reduction. I recommend that this manuscript should be accepted in the Journal.

Response: We are happy to note that the reviewer is satisfied with our revision and also thankful to the reviewer for accepting our manuscript.

Reviewer #2 (Remarks to the Author):

In this revised version the authors have provided new experiments and argumentation to address previous concerns. This reviewer appreciates the effort and detailed response to the role of BNAH/TEA, the literature overview on DRIFTS bands, and the dark EPR sample. Overall, the manuscript has been significantly improved, but a key conceptual aspect and a technical point remain. If the authors can satisfactorily address these, the manuscript can be considered for publication in Nature Communications:

Response: We are thankful to the reviewer for allowing us to clear out many doubts via experiment and DFT calculation. We are also grateful for giving us many constructive suggestions,

which uplift the paper's quality. Besides we also did a few experiments to justify the key conceptual concept adequately.

1. Technical: The response to our previous comment #1 on oxidation product detection is very welcome with the full detection of oxygen. These plots should be added to the SI and crucially, a plot is required which shows the CO₂ reduction product evolution and the oxygen evolution side-by-side. Here the number of transferred electrons should match to prove the connection between CH₄ and O₂ evolution.

Response: We thank the reviewer for the constructive suggestion. Now, we have plotted the oxygen evolution data along with CO, CH₄, H₂ and provided in the supplementary information fig. 34. In the course of CO₂RR in water medium the total amount of O₂ production is 866 μmol g⁻¹, which is in accordance with the reduced products (CH₄, CO, H₂).

Supplementary Fig. 34 **a** O₂ production (oxidation product from water oxidation) over MOF-808-PBA-MV as a function of time of irradiation in water. **b** Gas chromatogram of O₂ production over MOF-808-PBA-MV in water. **c** Mass spectrum of produced O₂ under visible light over MOF-808-PBA-MV.

2. Concept: Respectfully, the authors' response to the previous comment #7 misses the point. This reviewer is not criticizing a technical aspect but rather the misleading conceptual categorization of the samples. While the Tauc analysis presented in the revised version Fig. R10 with the same

eV scale is technically correct in terms of how the eV values were determined and that other systems in literature use this method (as the authors also stated), this reviewer regards this analysis method as generally not suitable for molecularly functionalized MOFs. While the pristine MOF can be considered as a semiconductor (where Tauc and Mott Schottky analyses are valid) the included molecules PBA and MV themselves certainly cannot. This is why the previous comment #7 suggested the authors treat their functionalized systems as a sensitized semiconductor (see extensive literature on dye-sensitized semiconductors/solar cells etc.). Here PBA-MV perform as molecular sensitizers with discrete orbitals while the MOF can be treated as a semiconductor with band structure. Accordingly, for the MOF-PBA and MOF-PBA-MV samples shown in Fig. R10 the Tauc and Mott-Schottky analysis methods are likely not a suitable to accurately describe the optoelectronic behaviour of the samples (see the corresponding discussion in previous literature reviews 10.1002/adma.201705512, 10.1002/anie.202106342). These aspects should be clarified, stated and the manuscript rephrased in all places necessary.

Response: We are thankful to the reviewer for the suggestion and for providing the related literatures. We have gone through the suggested literatures and found there are many techniques to determine the valance band and conduction band position of semiconductor material used for photocatalyst or photovoltaics applications. In 10.1002/anie.202106342, they mentioned for determining the valance band edge, a combination of techniques can be useful, e.g. Mott-Schottky, ultraviolet photoelectron spectroscopy (UPS), X-ray photoelectron spectroscopy (XPS) and so on.

According to the suggestion we have used ultraviolet photoelectron spectroscopy (UPS) to further understand the valance band and conduction band position of MOF-808, MOF-808-PBA, MOF-808-PBA-MV (**1. Phys. Chem. Chem. Phys.**, 2019, 21, 2318; **2. J. Mater. Chem. A**, 2019, 7, 221; **3. Angew. Chem. Int. Ed.** **2021**, 60, 26038). We followed a few previous literature reports to calculate the valance band position (**1. Langmuir** 2022, 38, 3139–3148; **2. Adv.Mater.**2023, 35, 2205994). The UPS results revealed that **MOF-808**, **MOF-808-PBA** and **MOF-808-PBA-MV** have the highest occupied states around 3.75, 2.28 and 1.3 eV, respectively (Supplementary Fig. 27). Next, the conduction band (LUMO) positions were derived based on UPS and the TAUC plots, which were found to be at -0.31, -0.47 and -0.63 V vs. NHE at pH=7 for **MOF-808**, **MOF-808-PBA** and **MOF-808-PBA-MV**, respectively (Supplementary Table 3). Hence the obtained value for **MOF-808-PBA** and **MOF-808-PBA-MV** is very close to the values obtained from Mott-Schottky. Here we applied a combination technique (Mott-Schottky and UPS) to obtain the band position of **MOF-808**, **MOF-808-PBA** and **MOF-808-PBA-MV**. Hence, we are not removing the Mott Schottky part from the manuscript. Also, we have added the UPS part in the manuscript and the supplementary information (Supplementary Fig. 27, Supplementary Table 3).

Please note that here as-synthesized MOF-808 is a wide bandgap semiconductor, which can not act as a visible light harvester. Hence, we introduce a CT complex inside the MOF pore to create an entirely new material with narrow bandgap properties. The CT complex acts as a visible light harvester and transfers the electrons to the catalytically active Zr_6 -oxo cluster.

Supplementary Fig. 27. UPS spectra in the cutoff and the onset energy regions of **MOF-808, MOF-808-PBA, MOF-808-PBA-MV.**

Supplementary Table 3 The energy levels within the **MOF-808, MOF-808-PBA, MOF-808-PBA-MV** obtained from the UPS and Mott-Schottky tests.

COMPOUNDS		MOF-808	MOF-808-PBA	MOF-808-PBA-MV
Band gaph		4.06 eV	2.75 eV	1.93 eV
UPS	HOMO	3.75 V	2.28 V	1.3 V
	LUMO	-0.31 V	-0.47 V	-0.63 V
M-S	HOMO	3.75 V	2.26 V	1.22 V
	LUMO	-0.31 V	-0.4 V	-0.67 V

Reviewer #3 (Remarks to the Author):

The authors have tried to address reviewers' comments. However, according to authors' new response to all reviewers, this reviewer is afraid of the reliability of the obtained results.

1. The response 1 to Reviewer #2 -- Although the authors supply some references about CO₂ reduction in pure water to 8/12-electrons products, the most examples of CO₂ to CH₄ were performed in water vapor (even CO₂ hydrogenation).

Response: We understand the reviewer's concern. The photocatalytic reduction of CO₂ in the presence of sunlight and water is the ultimate goal towards artificial photocatalysis (**1. *Angew. Chem. Int. Ed.* 2019, **58**, 632; **2. *ChemSusChem* 2020, **13**, 1725**). The photocatalytic CO₂ reduction has been performed in different condition with varied solvent systems with diverse solvent ratios, along with various sacrificial agents (triethanolamine (TEOA), triethylamine (TEA), triisopropanolamine (TIPA)) or without sacrificial electron donor. Some of the references are following [e.g. acetonitrile (*J. Am. Chem. Soc.* 2015, **137**, 13440-13443); water: acetonitrile (**1. *Nat. Commun.* 2020, **11**, 1149; **2. *Nat Catal*, 2021, **4**, 719–729**); water vapour (**1. *J. Am. Chem. Soc.* 2014, **136**, 15969-15976; **2. *small* 2015, **11**, 5262; **3. *ChemSusChem* 2020, **13**, 1725**); water (**1. *J. Am. Chem. Soc.* 2018, **140**, 6474; **2. *Nano Lett.* 2021, **21**, 2324; **3. *Angew. Chem. Int. Ed.* 2019, **58**, 632; **4. *ACS Appl. Mater. Interfaces* 2018, **10**, 2526; **4. *Energy Environ. Sci.*, 2022, **15**, 1967, and so on**). Many factors inhibit the use of dispersing the catalyst in pure water medium. Firstly, several catalysts are not stable in a water medium, e.g. perovskite catalyst is unstable in water (**1. *ACS Energy Lett.* 2018, **3**, 2656–2662; **2. *ACS Energy Lett.* 2021, **6**, 9, 3270–3274**). In that case, a minimal amount of water has been used for the proton and electron donor. Secondly, the particle size of the catalyst is another parameter that inhibits the reaction by dispersing in the catalytic solution. Most of the catalyst particle is bigger in size ($\sim 10^{-6}$ m). Hence it settles down in the reaction vessel and lacks good dispersibility. In that state, the catalyst needs to coat on a quartz surface so that light can fall directly on the catalyst surface. In our case, the catalyst is highly robust and nanoscale in size, which can form stable dispersion in the water medium. Hence, We performed CO₂RR by making a dispersion solution with our catalyst. Moreover, we believe performing the photocatalytic CO₂RR in water or water vapour is the same as in the absence of sacrificial electron donor (TEA, BNAH, TEOA, TIPA); it acts as a proton and electron source (**1. *ACS Appl. Mater. Interfaces* 2018, **10**, 2526; **2. *Angew. Chem. Int. Ed.* 2019, **58**, 632**).********************

Fig. R3. -- In Supplementary Table 6, MOF-808-PBA-MV in pure water can produce H₂, CH₄, CO and O₂. According to the analytic method in text (line 530-533), H₂, O₂, N₂, CO, CH₄, even CO₂ should be observable in the spectrum. However, only O₂ in Fig. R3b, which is more likely to be a signal of air.

Response: We understand the reviewer's concern. For that purpose, we want to say the oxidative product of the CO₂RR process was analyzed by the "Selected ion monitoring" (SIM) method using gas chromatography Mass Spectrometry (GCMS; Model-SHIMAZU GC-2010 PLUS). Hence, the peaks for other gases are not visible in the chromatogram. To clarify the reviewer's concern, we detected all the gases in full range using the scan mode, where the injected gas displayed the peaks of H₂, O₂, N₂, CH₄, CO and CO₂ in the chromatogram (Fig. R1). Generally, the full scan mode is useful for qualitative detection, mass scouting and for studies on the fragmentation patterns of unknowns. On the other hand, with the SIM mode lower detection limit with more accuracy can be obtained, which is better for quantitative analysis. Moreover, in SIM mode, the sensitivity is ten times better compared to scan mode. Hence for quantitative analysis, we preferred SIM mode rather than scan mode. In our experiment on oxidative products detection, we performed the catalysis in CO₂ atmosphere (purged with Ar followed by CO₂ purging). We observed an increment of O₂ evolution with time, which certainly comes from the oxidation of H₂O. Hence, we can strongly say the product (O₂) comes from an oxidative reaction of CO₂RR, not from the air.

Fig. R1 Gas chromatogram of O₂, H₂, CH₄, CO production over **MOF-808-PBA-MV** in water.

2. Fig. R7. – EPR signal comes from single electron. It is very strange that there is no any signal in Fig. R7. Actually, in my opinion, even defects, absorbed O₂, residual solvent/ligand will produce a EPR signal (at least free electron signal). In particular, the authors highlight many defects in this photocatalysts.

Response: We understand the reviewer's point. MOF-808 without defect formulated as $\{Zr_6(\mu^3-O)_4(\mu^3-OH)_4(BTC)_2(HCOO)_6\}$. We used defect-regulated Zr-based MOF 808, where a few formates or BTC linker is missing. The water molecule will occupy the missing linker site (**1. Chem. Mater.**, 2022, 34, 6734), and the charge will be balanced by μ^3-OH and μ^3-OH , which is well reported (**1. J. Am. Chem. Soc.** 2014, 136, 12844; **2. Energy Environ. Sci.**, 2021, 14, 2429). Hence no vacant site

is there for absorbed O_2 . Also, as the charge balance occurs by μ_3-O and μ_3-OH , as a result of that, the charge of the Zr^{4+} remains the same for pristine and post-modified Zr-MOF. Moreover, the defect-regulated pristine Zr-MOF does not show any EPR signal (Fig. R2). The EPR signal we only obtained when the post-modified MOF-808-PBA-MV sample was irradiated with light, which is due to the formation of charge separated state between PBA and MV and also for the formation of Zr^{3+} (Fig. 4b).

Fig. R2 EPR spectra of MOF-808 under daylight (Condition i) and irradiated with the LED light source (Condition ii).

3. Response to Reviewer #2 -- I am not satisfied with the answer to this question ---
-the authors' response does not give any real support. In addition, I don't agree that "We engineered the kinetics of the electron transfer, which we supported by the transient absorption experiment, and that is the beauty of our work." The photocurrent intensity is about $10 \mu A/cm^2$ that is a normal value in MOF photocatalysis. Overall, I have reservations on the acceptance of this manuscript.

Response: We understand the reviewer's concern. In this work we modified the electron transfer kinetics with the close proximity of photosensitizer and active catalyst site. The fast electron

transfer kinetics was experimentally proved by transient photo current experiment. The confinement effect of the MOF nanospace make the electron transfer kinetics faster and more number of electron is accumulated near the Zr-oxo cluster, which can produce higher order reduced product (*Energy Environ. Sci.*, 2023, DOI: [10.1039/D2EE03755F](https://doi.org/10.1039/D2EE03755F)).

However, "The photocurrent intensity is about 10 $\mu\text{A}/\text{cm}^2$ that is a normal value in MOF photocatalysis."-is startling.

Here we are providing few literatures based on Photocurrent experiment in MOF based catalyst, where the photocurrent value is much lesser compared to the obtained value in MOF-808-PBA-MV

1. *J. Am. Chem. Soc.* 2019, 141, 2054.
2. *J. Am. Chem. Soc.* 2020, 142, 19339
3. *J. Am. Chem. Soc.* 2021, 143, 12220.
4. *Chem. Sci.*, 2019, 10, 10577.
5. *ACS Appl. Energy Mater.* 2018, 1, 1913.
6. *Angew. Chem. Int. Ed.* 2021, 60, 16372.
7. *Chem. Sci.*, 2022, 13, 6696–6703.
8. *J. Am. Chem. Soc.* 2021, 143, 20792.

We are thankful to all the reviewers for their appreciation, critical comments and suggestions which has helped to improve the overall quality of the manuscript.

REVIEWER COMMENTS

Reviewer #2 (Remarks to the Author):

Assessment:

- The authors addressed point #1.
- Regarding point #2, I am not convinced by the authors' reply. They do not get the key problem and keep, in my opinion, making an error in their fundamental categorization of the materials. While they are not alone in doing so, it is an error that I wish to see less in the literature.
- Overall, while their results are truly interesting and they came a long way and improved the MS, the materials design concept, data interpretations and discussion have been puzzling from the beginning, which issues have not been sufficiently resolved.

Discussion:

A molecule functionalised-MOF cannot be considered as a new homogeneous material/semiconductor compared to the corresponding pristine MOF (some of which indeed can be; e.g. showing characteristic semiconductor behavior including charge carrier densities, mobilities etc.). MOF-anchored molecules may, for example, absorb light and have accessible HOMO & LUMO, but they do lack the energy continuum to create bands or band gap within the host material. This is especially true as the discrete molecular species are irregularly distributed, poorly interacting and outnumbered in the host MOF. Essentially, these species cannot be just regarded as a dopant like in classic semiconductors. For the same reasons, it is unlikely that they would dramatically modify the bands of their host. Therefore, Mott-Schottky analysis and other traditional semiconductor techniques (e.g., UPS, Tauc plot) are not applicable to reflect the band alignment and band gap of this new mixed material and characterizing the specific interaction between the MOF and the molecular guest. These data may indeed reflect changes induced by the MOF post-synthesis molecular functionalisation, but it is incorrect to assume that they reflect the behaviour of a truly hybridised homogeneous material (like a solid solution mixed element compound semiconductor).

This is the fundamental point of disagreement with the authors' materials concept and interpretation of their characterization data (including the references they provide). To me (and some few literature) each component of the functionalised-MOF stays energetically rather independent from the others, nevertheless, they interact (as it is in discrete supramolecular systems). Essentially, the authors' material show a more localized supramolecular behavior taking place in a solid state material.

I have been highlighting the above arguments for almost a year now and the authors do not specifically resolve this issue: rather they keep their interpretation, and, however, do not provide convincing arguments in opposition to our point. If they want to use Mott-Schottky, Tauc, UPS, etc., they need to prove that their functionalised material is indeed a homogeneous semiconductor with some clear

(unambiguous) semiconductor-defining experiments.

Alternatively, they could just revise their interpretation model/discussion into a (molecular/supramolecular) dye + MOF system instead of their current dyeMOF system. They may highlight these two concepts and provide the key descriptors of both to decide which of the concepts would be appropriate for interpretation of their materials (based on experimental data).

Reviewer #3 (Remarks to the Author):

Now the manuscript is ready for acceptance.

Reviewer's response (NCOMMS-22-08405C)

REVIEWER REPORT(S):

Thank you for sending us the reviewer reports on our manuscript titled " **Confining Charge-Transfer Complex in MOF for Modulating Electron Transfer Kinetics towards Photocatalytic Selective CO₂ Reduction in Water**" bearing manuscript id: **NCOMMS-22-08405C**.

REVIEWER REPORT(S):

Reviewer #2 (Remarks to the Author):

Assessment:

- The authors addressed point #1.

Regarding point #2, I am not convinced by the authors' reply. They do not get the key problem and keep, in my opinion, making an error in their fundamental categorization of the materials. While they are not alone in doing so, it is an error that I wish to see less in the literature. Overall, while their results are truly interesting and they came a long way and improved the MS, the materials design concept, data interpretations and discussion have been puzzling from the beginning, which issues have not been sufficiently resolved.

Discussion:

A molecule functionalized MOF cannot be considered as a new homogeneous material/semiconductor compared to the corresponding pristine MOF (some of which indeed can be, e.g., showing characteristic semiconductor behaviour including charge carrier densities, mobilities etc.). MOF-anchored molecules may, for example, absorb light and have accessible HOMO & LUMO, but they do lack the energy continuum to create bands or band gap within the host material. This is especially true as the discrete molecular species are irregularly distributed, poorly interacting and outnumbered in the host MOF. Essentially, these species cannot be just regarded as a dopant like in classic semiconductors. For the same reasons, it is unlikely that they would dramatically modify the bands of their host. Therefore, Mott-Schottky analysis and other traditional semiconductor techniques (e.g., UPS, Tauc plot) are not applicable to reflect the band

alignment and band gap of this new mixed material and characterizing the specific interaction between the MOF and the molecular guest. These data may indeed reflect changes induced by the MOF post-synthesis molecular functionalization, but it is incorrect to assume that they reflect the behaviour of a truly hybridized homogeneous material (like a solid solution mixed element compound semiconductor). This is the fundamental point of disagreement with the authors' materials concept and interpretation of their characterization data (including the references they provide). To me (and some few literature) each component of the functionalised-MOF stays energetically rather independent from the others, nevertheless, they interact (as it is in discrete supramolecular systems). Essentially, the authors' material show a more localized supramolecular behaviour taking place in a solid state material. I have been highlighting the above arguments for almost a year now and the authors do not specifically resolve this issue: rather they keep their interpretation, and, however, do not provide convincing arguments in opposition to our point. If they want to use Mott-Schottky, Tauc, UPS, etc., they need to prove that their functionalized material is indeed a homogeneous semiconductor with some clear (unambiguous) semiconductor-defining experiments. Alternatively, they could just revise their interpretation model/discussion into a (molecular/supramolecular) dye + MOF system instead of their current dye MOF system. They may highlight these two concepts and provide the key descriptors of both to decide which of the concepts would be appropriate for interpretation of their materials (based on experimental data).

Response: Thank you for the suggestions and for raising this very important aspect of MOF semiconductor material.

The major concern is that the reviewer does not agree that the CT MOF is truly hybridized, homogeneous material, like a solid solution mixed element semiconductor. Hence a band continuum is not expected, and the electronic structure cannot be described by the Mott-Schottky plot.

This might be true for some of the earlier reported MOFs with dopants. Post-synthetic modification (PSM) in metal-organic frameworks (MOFs) with organic moieties can lead to different band distributions depending on whether the introduction of such organic moieties is homogeneous or heterogeneous. Uniform stoichiometric distribution of organic molecules in the MOF lattice, by post-synthetic covalently replacing some of the original ligand, results in homogeneous distribution throughout the lattice. This

results in a consistent band distribution and electronic properties throughout the material. In contrast, the introduction of new organic molecule(s) by physical mixing with the MOF can lead to heterogeneous non-uniform distribution of the molecules. This non-uniform distribution creates localized regions with different band structures, leading to variations in the material's electronic properties. Understanding these differences is crucial as they directly impact the MOF's electrical and optical properties, including conductivity, carrier mobility, and bandgap, which are relevant to its performance in electronic, optoelectronic and catalytic applications. We have added a brief discussion on this aspect in the revised SI.

However, the current MOF system (MOF-808-PBA-MV) does not belong to this category. By using post-synthetic strategy, a new MOF semiconductor has been synthesized. We have performed post-synthetic modification of MOF-808 using 1-pyrene butyric acid (PBA) and methyl viologen (MV^{2+}). The formate ligand of MOF-808 was post-synthetically replaced with PBA, followed by the introduction of MV^{2+} to form the CT complex inside the MOF pore. The 1H -NMR study of the modified MOF-808-PBA indicated that four out of five formate ligands were exchanged with PBA molecules. Moreover, the 1H -NMR study suggests a PBA-MV ratio of 4:3, leading to the formulation of a new MOF: $[Zr_6O_5(OH)_3(BTC)_2(HCOO)(PBA)_4(MV^{2+})_3(H_2O)_2]$. Indeed, the periodicity of the MOF structure ensures the homogeneous distribution of the charge transfer (CT) complex (PBA-MV) throughout the MOF framework while maintaining stoichiometry of the CT complex. As a result of this, the interaction between the cluster and the CT complex is strong, and the band structure of the MOF has been completely changed compared to the pristine MOF. Hence, this is not a (molecular/supramolecular) dye + MOF system, as the CT complex is not isolated rather a part of the MOF.

To get a better vision of homogenous and heterogeneous systems in MOF through post-synthetic modification, we prepared two compounds: one by physically mixing MOF-808 with PBA, and another by physically mixing MOF-808 with PBA-MV. These physical mixtures were prepared by grinding the MOF with the respective ligands in the presence of a small quantity of DMF/MeOH. This physical mixture MOF +PBA+MV can be considered as (molecular/supramolecular) dye + MOF system. Now, we have compared the absorption spectra of the post-synthetically modified MOF and a

(molecular/supramolecular) dye+MOF system (MOF+PBA+MV). The (molecular/supramolecular) dye + MOF system does exhibit an energetically high CT state, very dissimilar to the MOF-808-PBA-MV CT, suggesting the homogeneous MOF-808-PBA-MV CT structure. We have also performed photocatalytic CO₂ reduction with (molecular/supramolecular) dye + MOF system in the presence of BNAH/TEA, which showed minimal activity as compared to MOF-808-PBA-MV. This poor activity can be attributed to the unstable CT complex in the solution state of MOF-808(PBA+MV)-Phy mix. which is clearly observed in UV-vis spectra (Supplementary Fig. 29). The absence of CT band implies that the CT complex formed by the physical mixture on the MOF-808 surface is not stable and disintegrate in the aqueous medium and thus could not act as light-harvesting agent to channelize the electron transfer the catalytic Zr-oxo cluster. But in case of MOF-808-PBA-MV, the electron transfer is not localized to the PBA-MV to the adjacent Zr-oxo cluster but is also possible to the other neighbouring clusters. This is feasible because of the periodic lattice of the MOF structure. Hence, consideration of a bandlike electronic structure is correct. This is discussed in the revised manuscript and supplementary information.

We understand the disagreement proposed by the reviewer, and to clarify this, we have stated both the arguments in the revised manuscript and supplementary information.

Supplementary Fig. 28 UV-vis spectra of MOF-808-PBA-MV, MOF-808-PBA, MOF-808-(PBA)-Phy mix., MOF-808-(PBA+MV)-Phy mix.

Table R1 photocatalytic CO₂ reduction with MOF-808-(PBA-MV)-Phy mix.

Catalyst	Reaction condition	Sacrificial Agent	Products		
			H ₂ (μmol g ⁻¹)	CO (μmol g ⁻¹)	CH ₄ (μmol g ⁻¹)
MOF-808(PBA+MV)-Phy mix.	300 W Xe lamp, >400 nm CO ₂ saturated water medium	BNAH, Triethylamine (TEA)	104	45	56

Supplementary Fig. 19 UV-vis spectra of MOF-808(PBA+MV)-Phy mix. in water.

Reviewer #3 (Remarks to the Author):
Now the manuscript is ready for acceptance.

Response: We are happy to note that the reviewer is satisfied with our revision and also thankful to the reviewer for accepting our manuscript.

We are thankful to all the reviewers for their appreciation, critical comments and suggestions, which have helped to improve the overall quality of the manuscript.